# Multi-omics characterization of autophagy-related molecular features for therapeutic targeting of autophagy

Mei Luo[1,2,3,7], Lin Ye[1,7], Ruimin Chang[1,7], Youqiong Ye[3,4], Zhao Zhang[3], Chunjie Liu[2,3], Shengli Li[3], Ying Jing[3], Hang Ruan[3], Guanxiong Zhang[1], Yi He[1], Yaoming Liu[3], Yu Xue[2], Xiang Chen[1]✉, An-Yuan Guo[2]✉, Hong Liu[1]✉ & Leng Han[3,5,6]✉

Autophagy is a major contributor to anti-cancer therapy resistance. Many efforts have been made to understand and overcome autophagy-mediated therapy resistance, but these efforts have been unsuccessful in clinical applications. In this study, we establish an autophagy signature to estimate tumor autophagy status. We then classify approximately 10,000 tumor samples across 33 cancer types from The Cancer Genome Atlas into autophagy score-high and autophagy score-low groups. We characterize the associations between multi-dimensional molecular features and tumor autophagy, and further analyse the effects of autophagy status on drug response. In contrast to the conventional view that the induction of autophagy serves as a key resistance mechanism during cancer therapy, our analysis reveals that autophagy induction may also sensitize cancer cells to anti-cancer drugs. We further experimentally validate this phenomenon for several anti-cancer drugs in vitro and in vivo, and reveal that autophagy inducers potentially sensitizes tumor cells to etoposide through downregulating the expression level of *DDIT4*. Our study provides a comprehensive landscape of molecular alterations associated with tumor autophagy and highlights an opportunity to leverage multi-omics analysis to utilize multiple drug sensitivity induced by autophagy.

Autophagy, also known as macroautophagy, is an evolutionarily conserved lysosomal degradation catabolic process, by which cells maintain cellular homeostasis via recycling nutrients from damaged organelles and proteins[1,2]. Autophagy, as a crosstalk of multiple biological processes, can affect cancers through multiple layers of molecular alterations, from genomics to transcriptomics and proteomics. For example, monoallelic deletion of the autophagy regulator *BECN1* contributes to tumorigenesis in many cancers, such as breast,

[1]Department of Dermatology, Hunan Engineering Research Center of Skin Health and Disease, Hunan Key Laboratory of Skin Cancer and Psoriasis, National Clinical Research Center for Geriatric Disorders, Xiangya Hospital, Central South University, Changsha, Hunan, China. [2]Center for Artificial Intelligence Biology, Hubei Bioinformatics & Molecular Imaging Key Laboratory, Key Laboratory of Molecular Biophysics of the Ministry of Education, College of Life Science and Technology, Huazhong University of Science and Technology, Wuhan 430074, China. [3]Department of Biochemistry and Molecular Biology, McGovern Medical School at The University of Texas Health Science Center at Houston, Houston, TX 77030, USA. [4]Shanghai Institute of Immunology, Department of Immunology and Microbiology, Shanghai Jiao Tong University School of Medicine, Shanghai 200025, China. [5]Center for Epigenetics and Disease Prevention, Institute of Biosciences and Technology, Texas A&M University, Houston, TX, USA. [6]Department of Translational Medical Sciences, College of Medicine, Texas A&M University, Houston, TX, USA. [7]These authors contributed equally: Mei Luo, Lin Ye, Ruimin Chang. ✉e-mail: chenxiangck@126.com; guoay@hust.edu.cn; hongliu1014@csu.edu.cn; leng.han@tamu.edu

prostate, and lung carcinomas[3,4]. Overexpression of *BECN1* suppresses cell proliferation in synovial sarcoma cells[5]. Downregulated expression of autophagy-related gene 5 (*ATG5*) promotes the carcinogenesis of early-stage cutaneous melanoma[6]. Proteomics analysis has shown that inhibition of both APE1 and autophagy can overcome resistance to cisplatin in the lung adenocarcinoma cell line A549 by enhancing apoptosis[7]. Moreover, a multi-dimensional molecular regulatory network constructed by miRNAs and transcription factors (TFs) plays a crucial role in the autophagy process. The miRNA-181a targets *ATG5* to regulate starvation- and rapamycin-induced autophagy[8], and miRNA-30a inhibits autophagy and sensitizes tumor cells to imatinib by downregulating *BECN1* expression in gastrointestinal stromal tumors[9]. Massive studies have demonstrated that FOXO members of TFs contribute to promoting the expression of genes involved in autophagy[10]. FOXO1, a member of FOXO TFs, can induce autophagic flux by interacting with ATG7, leading to increased apoptosis in tumors[11]. FOXO3 activation modulates multiple steps to promote autophagic flux, including increasing WIPI puncta formation and inducing ULK2 and LC3 colocalization[12]. FOXO3 can inhibit FOXO1 to negatively regulate autophagy in cancer cells[13]. These studies suggest a complicated interplay involved in the autophagy process. However, there is currently a comprehensive multi-omics analysis to elaborate on autophagy-related molecular alterations in cancer yet.

Recent studies have demonstrated robustness in estimating autophagy through LC3-based assays, SQSTM1/p62-based assays, and direct observation of autophagy-related structures and fate by electron microscopy[14]. However, none of these approaches are currently feasible under human physiological and pathological conditions. Despite a large number of autophagy-related genes that have been identified and collected in several databases, including MSigDB[15], Autophagy Database[16], THANATOS[17], HADB[18], HAMDB[19], and ncRDeathDB[20], there is no gene signature applied to cancer samples to estimate the autophagy status. In this study, we utilize single-sample gene set enrichment analysis (ssGSEA)[21], a widely used method[22–25], to estimate autophagy status in a large number of cancer samples, followed by a comprehensive analysis to understand molecular alterations in autophagy.

Furthermore, many studies have shown that autophagy serves as a key resistance mechanism to anticancer therapy, and that autophagy inhibition can improve drug sensitivity[26]. For example, tioconazole, targeting *ATG4*, can inhibit autophagy to enhance chemotherapeutic cytotoxicity in multiple cancer cell lines[27]. In vivo evidence has further demonstrated that autophagy inhibitors inhibit the growth of established tumors and improve the response to cancer therapy[1]. Based on this promising evidence, dozens of clinical trials involving autophagy inhibitors are ongoing, e.g., chloroquine or hydroxychloroquine, in combination with other drugs, such as the mTOR inhibitor temsirolimus[28], the proteasome inhibitor bortezomib[29], and the histone deacetylase inhibitor vorinostat[30]. The combination of the autophagy inhibitor pantoprazole with docetaxel has also been investigated in metastatic castration-resistant prostate cancer[31]. Most efforts focus on inhibiting autophagy to promote the efficacy of cancer therapy.

In this work, we estimate the tumor autophagy status for ~10,000 tumor samples across 33 cancer types from TCGA. We further characterize the associations between multi-dimensional molecular features and tumor autophagy, and demonstrate the effects of autophagy status on drug response. Through this comprehensive landscape of molecular alterations associated with tumor autophagy, our study provides biological insight into the therapeutic targeting of autophagy

## Results

### Identification of a gene signature to estimate autophagy status across cancer samples

To estimate the autophagy status, we first collected autophagy-related genes from six databases (MSigDB, HADB, HAMDB, ncRDeathDB,

THANATOS, and Autophagy)[15–20], and also collected six independent datasets with known autophagy status (GSE107600, GSE117189, GSE129204, GSE106175, GSE90444, and GSE31397). We then used the ssGSEA[21] algorithm to calculate the autophagy scores based on autophagy-related genes in each database throughout all six independent datasets. The ssGSEA scores based on autophagy-related genes from 4 databases (MSigDB, HADB, HAMDB, and ncRDeathDB) can accurately distinguish the cell lines in autophagy-high status vs. autophagy-low status in at least 4 independent datasets (marked in red in Fig. 1a). We integrated rigorous autophagy gene sets, using genes ($n$ = 37) overlapping in all four databases in this study (Fig. 1b, Supplementary Data 1). Thirty-two out of 37 genes are directly involved in the key steps of autophagy. For example, they are involved in the initiation (e.g., *ULK1*, *ULK2*, *ATG13*, *RB1CC1*, also known as *FIP200*), nucleation (e.g., *PIK3C3*, also known as *VPS34*, *BCL2*, *UVRAG*, and *ATG14*), elongation (e.g., *ATG3*, *ATG5*, *ATG7*, *ATG12*, *ATG16L1*, *GABARAPL1*, *MAP1LC3A*, and *MAP1LC3B*), cargo loading (e.g., *SQSTM1* and *CALCOCO2*), and fusion of AV and lysosomes (e.g., *LAMP1*; Supplementary Data 1)[1,16,32,33]. Notably, the 37-gene set was further consistently confirmed in 6 public datasets, including 5 different cancer types for which samples with high autophagy levels showed significantly higher autophagy scores than samples with low autophagy levels (Fig. 1c). Moreover, the autophagy score calculated for approximately 10,000 tumor samples across 33 cancers from The Cancer Genome Atlas (TCGA) based on this 37-gene signature is highly consistent with the score based on those genes from MSigDB, HADB, HAMDB, and ncRDeathDB (Fig. 1d), suggesting the robustness of the 37-gene signature to define autophagy status across cancer samples.

### Global landscape of multi-omics autophagy-associated alterations across multiple cancer types

To understand the global pattern of molecular alterations associated with the autophagy process, we classified these samples from TCGA into autophagy score-high, score-intermediate, and score-low groups using the distribution of score tertiles. In total, 24 cancer types with more than 30 samples in the autophagy score-high group and autophagy score-low group were used for subsequent analysis. To identify the molecular alterations associated with autophagy, we applied the propensity score matching (PSM) algorithm[22,34–38], which has been widely used in previous studies, to minimize the effects of clinical confounders (e.g., sex, age at initial pathologic diagnosis, tumor purity, pathologic stage, and histological type; Supplementary Fig. 1). We summarize the overall analysis in Fig. 2a.

In this study, we tested 5 different molecular features including 20,288 mRNAs, 218 proteins, 2435 miRNAs, 3785 mutations, and 450 somatic copy number alterations (SCNAs) in 24 cancer types (Fig. 2a). We identified significant numbers of molecular alterations associated with autophagy status (Fig. 2b) and the alterations significantly varied across different cancer types. For example, mRNA expression showed the largest number of variations (8531 mRNA variations), ranging from 58 genes in esophageal carcinoma to 3632 genes in thymoma (THYM; Fig. 2b). In contrast, somatic variation showed minimum variation, with only 51 mutations in eight cancer types, ranging from 1 mutation in lung squamous cell carcinoma (LUSC) to 32 mutations in stomach adenocarcinoma (STAD). The frequency of each molecular feature in different molecular profiles varied greatly across cancers. For example, several cancer types, including THYM, testicular germ cell tumor (TGCT), and skin cutaneous melanoma (SKCM), showed a larger number of variations between the autophagy score-high and autophagy score-low groups (Fig. 2b).

### Associations between autophagy and mRNA/protein expression and signaling pathways

To investigate the associations between autophagy and mRNA expression, we focused on the significantly altered mRNAs across 24

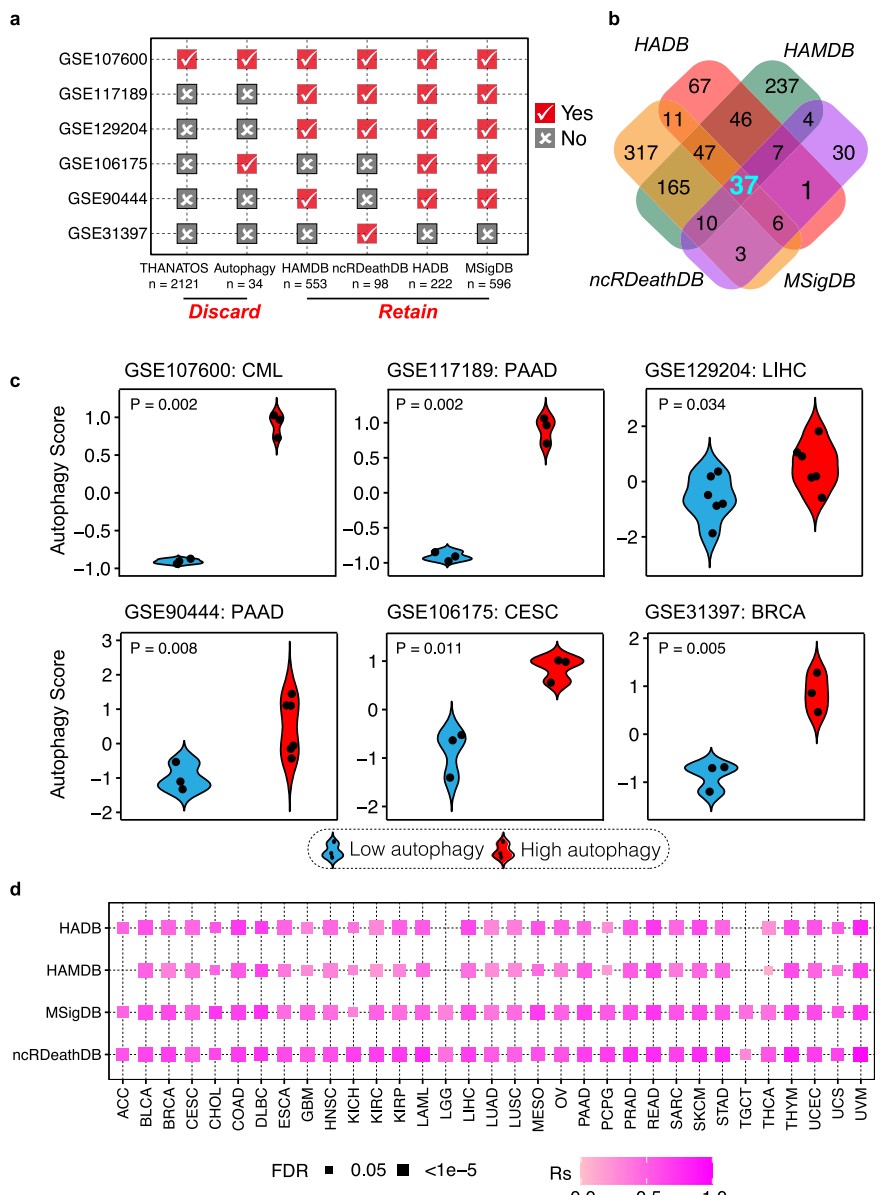

**Fig. 1 | Identification of a gene signature to estimate autophagy status across cancer samples. a** Autophagy status was evaluated by autophagy scores based on gene lists from 6 databases in the 6 public datasets. 'Yes' (check mark) indicates a gene list validated for autophagy status; 'No' (x) indicates a gene list invalidated. **b** Venn diagram of autophagy-related gene lists from the 4 better-performing databases. **c** Autophagy scores of 6 datasets based on 37-gene set signature in the autophagy score-high group (red) and autophagy score-low group (blue). Two-sided student's *t* test was used to assess the difference; $P < 0.05$. **d** Correlations between autophagy scores based on the 37-gene set and gene sets from MSigDB, HADB, HAMDB, and ncRDeathDB among TCGA tumor samples. Color intensity indicates Spearman correlation coefficient (Rs); rectangle size indicates FDR for Spearman correlation; FDR < 0.05; the absence of the rectangle means non-significance (FDR > 0.05). The Benjamini & Hochberg method was used for multiple hypothesis testing. Abbreviations: ACC Adrenocortical carcinoma, BLCA Bladder Urothelial Carcinoma, BRCA Breast invasive carcinoma, CESC Cervical squamous cell carcinoma and endocervical adenocarcinoma, CHOL Cholangiocarcinoma, COAD Colon adenocarcinoma, DLBC Lymphoid neoplasm diffuse large B-cell lymphoma, ESCA Esophageal carcinoma, GBM Glioblastoma, HNSC Head and Neck squamous cell carcinoma, KICH Kidney chromophobe, KIRC Kidney renal clear cell carcinoma, KIRP Kidney renal papillary cell carcinoma, LAML Acute Myeloid Leukemia, LGG Brain lower grade glioma, LIHC Liver hepatocellular carcinoma, LUAD Lung adenocarcinoma, LUSC Lung squamous cell carcinoma, MESO Mesothelioma, OV Ovarian serous cystadenocarcinoma, PAAD Pancreatic adenocarcinoma, PCPG Pheochromocytoma and paraganglioma, PRAD Prostate adenocarcinoma, READ Rectum adenocarcinoma, SARC Sarcoma, SKCM Skin Cutaneous Melanoma, STAD Stomach adenocarcinoma, THCA Thyroid carcinoma, TGCT Testicular germ cell tumors, THYM Thymoma, UCEC Uterine Corpus Endometrial Carcinoma, UCS Uterine carcinosarcoma, UVM Uveal melanoma, CML Chronic myelogenous leukemia.

cancers. These altered genes were enriched in 16 pathways from the Kyoto Encyclopedia of Genes and Genomes (KEGG), which were enriched with more than 10 genes in at least two cancer types (*p* value <0.05; Fig. 3a, Supplementary Data 2). These pathways include some autophagy-related pathways, such as the PI3K-AKT signaling pathway, MAPK signaling pathway, and Phagosome (Supplementary Data 2), which are frequently reported to modulate the autophagy process in multiple cancers[39,40].

Functional proteomics data of reverse-phase protein arrays cover key cancer-related total and phosphorylated proteins[22,37]. To further understand the potential effects of autophagy on protein expression, we compared the pathway scores between the autophagy score-high

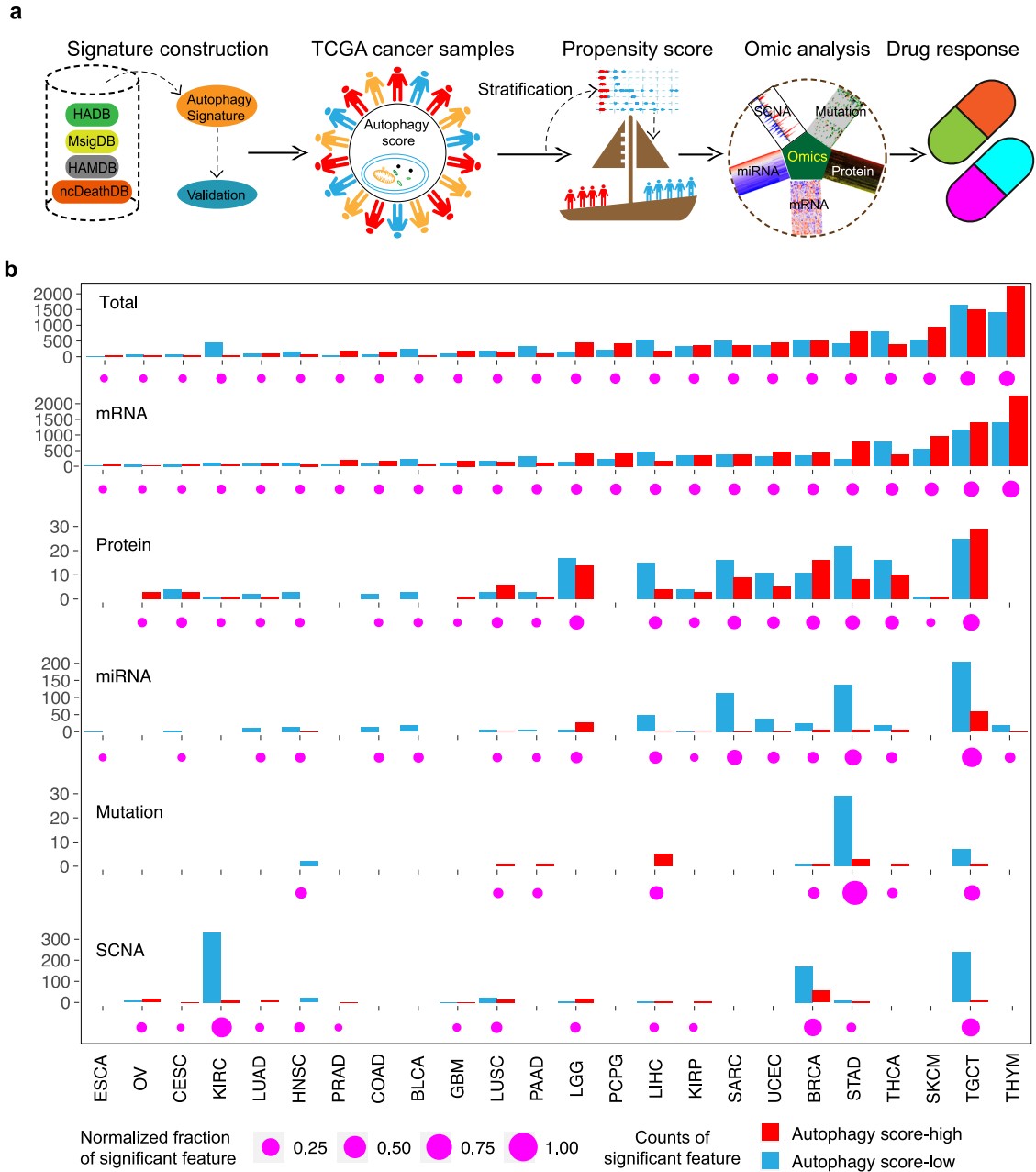

**Fig. 2 | Overview of five molecular features across cancer types. a** Integrated analysis of multi-omics and drug response across different cancer types. **b** Number of each altered molecular feature (mRNA, miRNA, mutation, protein, and SCNA) and total altered molecular features in autophagy score-high (red) and autophagy score-low (blue) groups from TCGA tumor samples. The Magenta point denotes the percentage of significant features over the total features in each cancer. SCNA, somatic copy number alterations.

group and the autophagy score-low group based on protein expression in 10 cancer signaling pathways, which are defined based on the reverse-phase protein array from TCGA[41–44]. Notably, we found that the mTOR pathway was significantly enriched in the autophagy score-low group in three cancers (LUSC: false discovery rate [FDR] = 0.032; SARC: FDR = 0.004, thyroid carcinoma (THCA): FDR = 0.009; Fig. 3b), which was consistent with previous observations that the mTOR pathway inhibits autophagy[1]. Moreover, we found that the RAS-MAPK pathway, as the activator of autophagy, was enriched in the autophagy score-high group in two cancers (glioblastoma multiforme: FDR = 0.017; PRAD: FDR = 0.041; Fig. 3b). The enrichment of epithelial-mesenchymal transition (EMT) in the autophagy score-high group of STAD and TGTC suggests associations between autophagy and EMT (Fig. 3b)[45].

To further explore the association between biological pathways and drug response, we calculated the correlations between the mRNA expression of genes in multiple cancer-related pathways and the area under the dose–response curve (AUC) of 252 anticancer drugs in Genomics of Drug Sensitivity in Cancer (GDSC)[46] across cancer cell lines. We found that 197 differentially expressed autophagy-associated genes in 16 cancer signaling pathways are highly associated with the drug response to 119 anticancer drugs in at least three cancer types ($|Rs|$ > 0.3, FDR < 0.05), and the majority of genes (91.1%; $n$ = 180) are associated with the sensitization of anticancer drugs (Fig. 3c). These pathways include some autophagy-associated pathways, such as the PI3K-AKT signaling pathway, MAPK signaling pathway, and mTOR signaling pathway (Fig. 3c). For example, the PI3K signaling pathway, which has been reported to be an inhibitor of the chemosensitivity[47], was

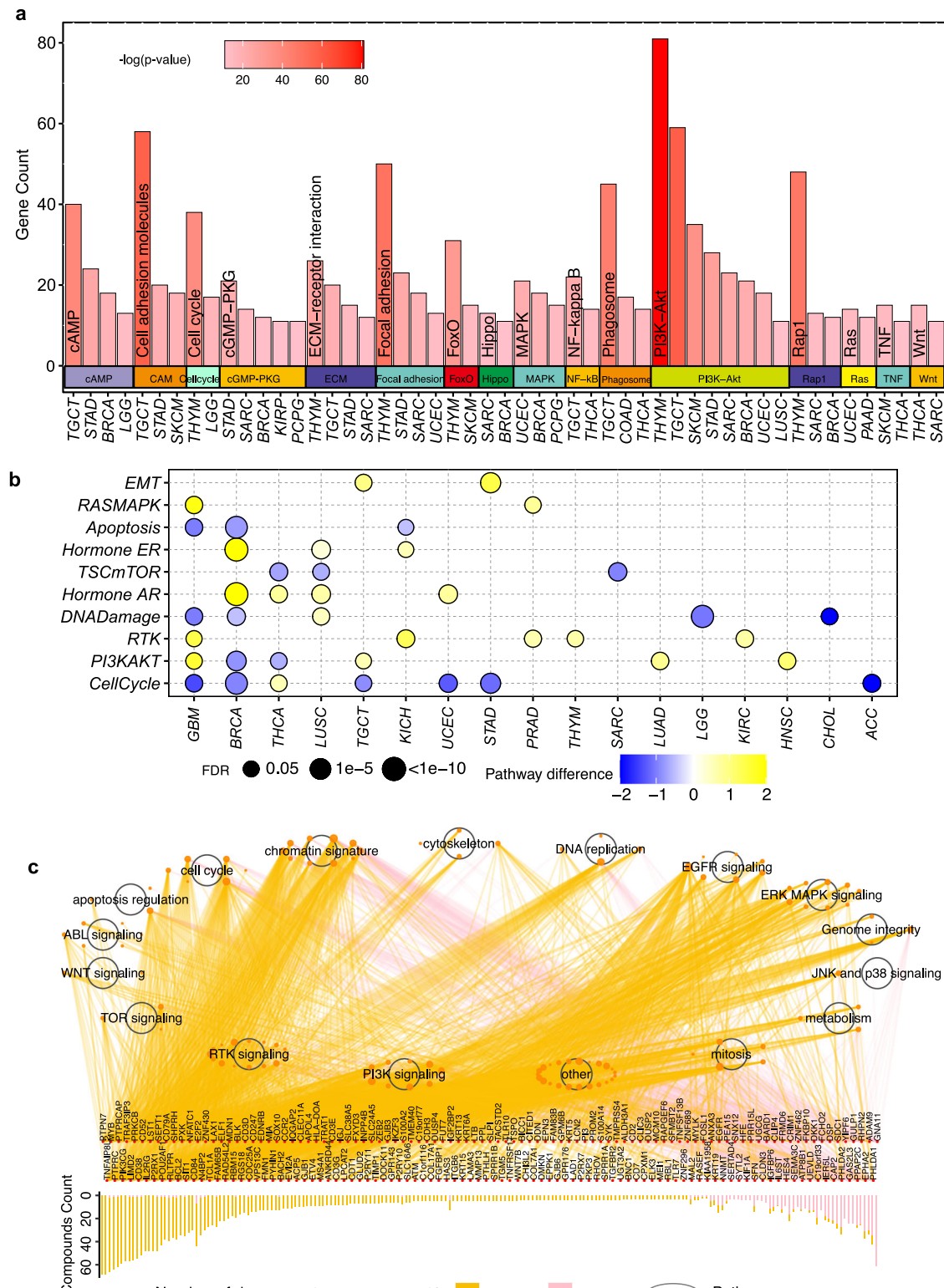

**Fig. 3 | Associations between autophagy and mRNA/ protein expression and signaling pathways. a** Enriched pathways with gene counts >10 in at least two cancers by significantly altered mRNAs in the comparison of the autophagy score-high group vs. the autophagy score-low group. Barplot indicates the enriched genes; barplot color intensity indicates the significance of the enriched pathways. Enrichment of pathways was evaluated with "clusterProfile" package. **b** Altered signaling pathways based on functional proteomics data of reverse-phase protein arrays in the autophagy score-high group vs. the autophagy score-low group for multiple cancers. Color indicates the difference in pathway score; point size indicates FDR for pathway score. **c** Spearman correlation between mRNA expression levels of autophagy-associated genes in different signaling pathways and area under the curve across 1074 cancer cell lines. Barplot denotes the number of drugs targeting each gene; the orange point shows the drug's targeted pathway; the point size indicates the number of genes correlated with drug sensitivity ($|Rs| > 0.3$, FDR < 0.05).

significantly varied in both mRNA and protein profiling. We found that 79 genes in the PI3K signaling pathway are associated with the sensitization of 11 anticancer drugs ($Rs < −0.3$, FDR < 0.05; Supplementary Fig. 2a), and 34 genes in the PI3K signaling pathway are associated with the resistance of nine anticancer drugs ($Rs > 0.3$, FDR < 0.05; Supplementary Fig. 2a). Regulation of MAPK signaling pathway enhances the drug sensitivity of cancer cells in multiple cancer types[48,49]. We found that 36 genes in the MAPK signaling pathway are associated with the sensitization of 11 anticancer drugs ($Rs < −0.3$, FDR < 0.05; Supplementary Fig. 2b), and four genes in the MAPK signaling pathway are associated with the resistance of nine anticancer drugs ($Rs > 0.3$, FDR < 0.05; Supplementary Fig. 2b). Previous studies have also indicated that regulating the mTOR signaling pathway can sensitize cancer cells to anticancer drugs[50,51]. We found that 26 genes in the mTOR signaling pathway are associated with the sensitization of two anticancer drugs ($Rs < −0.3$, FDR < 0.05; Supplementary Fig. 2c), and 10 genes in the mTOR signaling pathway are associated with the resistance of nine anticancer drugs ($Rs > 0.3$, FDR < 0.05; Supplementary Fig. 2c). These results provide a comprehensive view for exploring therapeutic strategies via targeting genes in signaling pathways.

### Regulatory network among autophagy-related miRNAs, target genes, and TFs

To further understand the regulatory network in autophagy, we identified 36 autophagy-related miRNAs in at least four cancers and constructed a miRNA-target regulatory network to explore the effects of miRNAs on autophagy. We identified 4062 edges between 36 miRNAs and 1994 genes with significant expression alterations in at least three cancers (Fig. 4a). These genes targeted by miRNAs were enriched in multiple autophagy-related pathways, including the PI3K-AKT and MAPK signaling pathways (Fig. 4a, FDR < 0.05; Supplementary Fig. 3a, Supplementary Data 3). For example, in the PI3K-AKT signaling pathway, miR-106a, miR-17-5p, and miR-93-5p have lower expression in the autophagy score-high group in at least four cancers (Fig. 4a; Supplementary Fig. 3b). Previous studies have shown that miR-106a, miR-17-5p, and miR-93-5p, as members of the miR-17 family, can inhibit autophagy by targeting autophagy-related genes[52–54]. These varied miRNAs and interactions provide crucial insights into the regulatory mechanisms of autophagy involved in cancer.

TFs are crucial regulators of gene expression. We found that 94 TFs were significantly altered in at least nine cancers, and this TFs targeted 1759 significantly varied mRNAs (Fig. 4b). The mRNAs targeted by TFs were enriched in 9 signaling pathways, including the autophagy-related PI3K-AKT signaling pathway (Fig. 4b, Supplementary Fig. 3c; Supplementary Data 4). The genes of the PI3K-AKT signaling pathway were targeted by 66 TFs, which included FOXA3, FOXJ1, FOXL1, and FOXH1 (Supplementary Fig. 3c). These TFs, as members of the forkhead TF (FOX) family, were significantly downregulated in the autophagy score-high group of multiple cancer types (Fig. 4b), which is consistent with previous knowledge that knockdown of FOXA3 enhances the autophagy process[55]. Interestingly, we also observed alterations in several other FOX family members under autophagy status. For example, FOXP2 was upregulated in the autophagy score-high group of 7 cancer types (Fig. 4b), and the nonsense mutant of FOXP1, as the same subclass of FOXP2, can induce autophagy[56]. FOXA2 was downregulated in the autophagy score-high group of 4 cancer types, and FOXA2 was regulated by autophagy activity in ovarian cancer stem cells[57]. Taken together, our results suggest that TFs play a significant regulatory role in autophagy, which provides biological insights for further investigations.

### Association between autophagy and somatic copy number alterations

SCNAs can affect the expression level of genes in multiple processes of autophagy across various cancer types. For example, 5q copy number gain leads to the overexpression of *SQSTM1*, an autophagy cargo, in kidney cancer[58]. Copy number amplification at 2q37 leads to elevated expression levels of *ATG16L1*, a key component of a large autophagy-related protein complex in prostate cancer[59]. To explore the roles of SCNAs across multiple cancers, we evaluated and identified 82 SCNAs significantly altered in 14 cancer types (Fig. 5a), ranging from 1 SCNA in prostate adenocarcinoma (PRAD) to 33 SCNAs in BRCA. These regions covered 850 genes, ranging from 1 gene in PRAD to 341 genes in kidney renal clear cell carcinoma (KIRC; Fig. 2b). For example, we identified a total of 69 deletions in 13 cancer types (Fig. 5a), with coverage of 588 genes in these regions. We identified only 36 amplifications in 10 cancer types (Fig. 5a), with 267 genes located in these regions.

Furthermore, 16 clinically actionable genes were harbored in these altered SCNAs (Fig. 5b), and they were targeted by 37 anticancer drugs of four categories, including target therapy and immunotherapy (Fig. 5c). For example, 2q37.3 deletion is the most frequent SCNA in the autophagy score-high group of four cancers (OV, BRCA, LGG, and KIRP; Fig. 5a). The programmed cell death protein 1 (*PDCD1*), as the target of pembrolizumab and nivolumab for cancer immunotherapy[60], is located in the 2q37.3 region in BRCA and LGG (Fig. 5b). *MTOR*, as one of the most well-known components targeting the autophagy process in drug development, is located in the 1p36.13 deletion region in autophagy score-high samples of KIRC and BRCA (Fig. 5b). Notably, *MTOR* was targeted by the mTOR inhibitors temsirolimus and everolimus (RAD001; Fig. 5c), which have been reported in multiple studies as promising drugs to improve the sensitivity of combination therapy by mediating autophagy status[28,61]. These results suggest that autophagy-associated SCNAs can affect the sensitivity of anticancer drugs, including drugs for both target therapy and immunotherapy.

### Functional effects of autophagy status on drug response

Massive studies have demonstrated that autophagy can promote tumorigenesis, and that autophagy inhibition can sensitize resistant tumor cells to chemotherapy, radiation therapy, and targeted therapy[62,63]. We comprehensively depicted the associations between autophagy status and drug response (Supplementary Fig. 4). We first calculated the correlation between the autophagy score and imputed drug data of 138 drugs in TCGA[64]. We found that the number of drugs associated with autophagy varied from 7 in ACC to 73 in TGCT (Supplementary Fig. 4a; FDR < 0.05). Among these drugs, a total of 41 drugs targeted 32 clinically actionable genes (CAGs), which are targeted by 13 Food and Drug Administration (FDA)-approved drugs. These CAGs were altered in four different molecular layers (mRNA expression, protein expression, SCNA, and somatic mutation) of 20 cancer types (Fig. 6). For example, the clinically actionable gene AR was significantly changed at the mRNA and protein levels in 10 cancer types, including PRAD. The AR targeted by the anti-androgen bicalutamide (Fig. 6) has consistently been reported to promote autophagy in prostate cancer cells[65,66]. Furthermore, we observed that paclitaxel was more resistant in patients with LUAD and autophagy score-high (Supplementary Fig. 4a), which is consistent with the previous observation that 3-methyladenine (3-MA), an autophagy inhibitor, will lead A549 cells to be more sensitive to paclitaxel[67]. Surprisingly, autophagy score-high cancer samples are more sensitive to many drugs, including erlotinib in seven cancer types (Fig. 6 and Supplementary Fig. 4a). This finding contrasts with the traditional view that autophagy will lead to drug resistance, suggesting complicated effects of autophagy on the drug response.

### Autophagy sensitizing drug response in vitro and in vivo

Melanoma is one of the top cancer types (THYM, TGCT, SKCM) with the largest number of variations affected by autophagy status (Fig. 2b), and it is more common than the other two cancer types. We thus examined the drug response sensitized by autophagy in melanoma. We selected five drugs (BMS708163, CMK, BMS536924, DMOG,

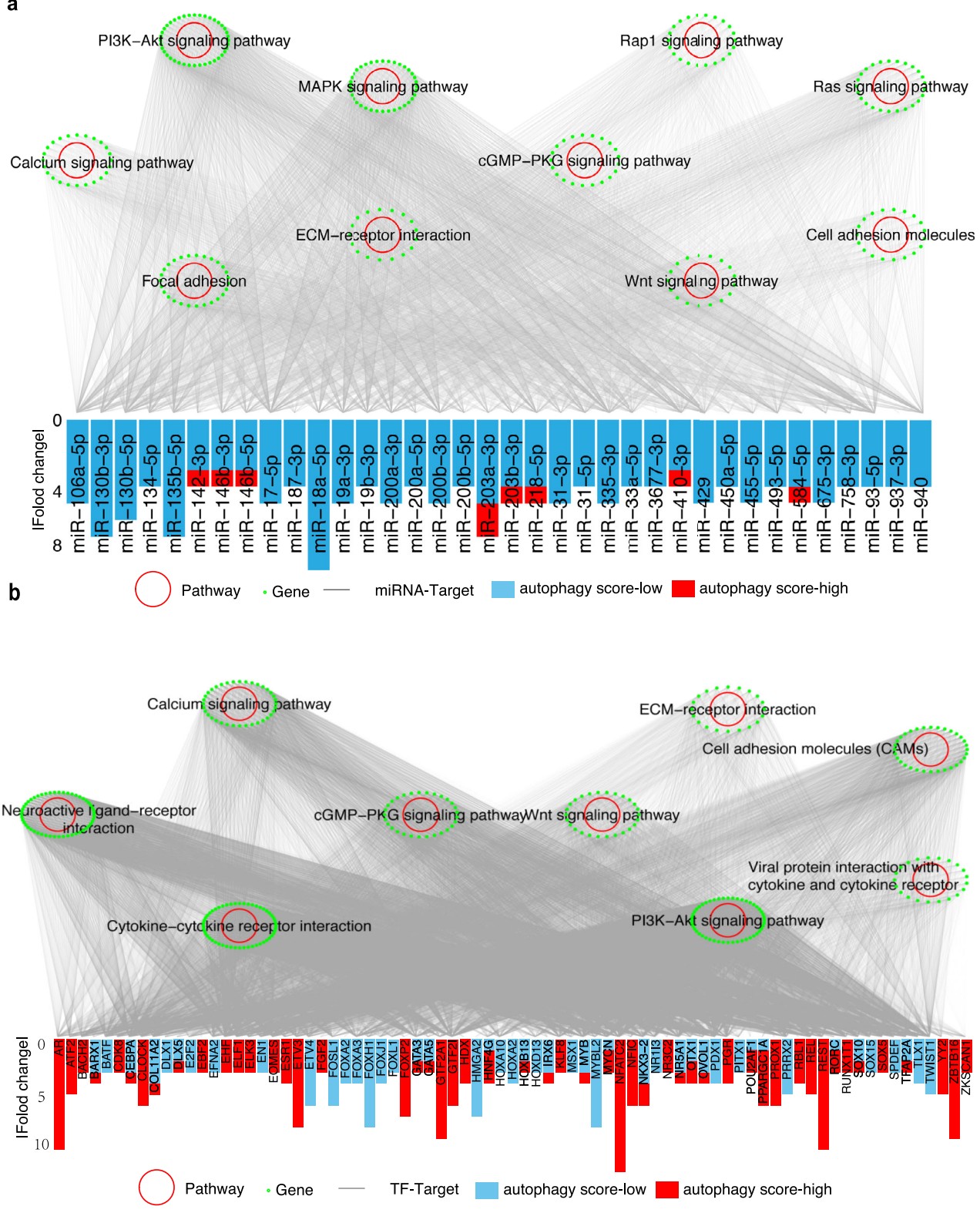

**Fig. 4 | miRNA-target and TF-target regulatory networks in autophagy.**
**a** Significant KEGG pathways enriched by mRNAs in the miRNA-target regulatory network. Green dots: genes; red circles: enriched KEGG pathways with FDR < 0.05; barplot denotes the number of cancer types with altered miRNAs. Y-axis denotes | fold change|. **b** Significant KEGG pathways enriched by mRNAs in the TF-target regulatory network. Green points: genes; red circle: enriched KEGG pathways with FDR < 0.05; barplot denotes the number of cancer types with altered TFs. The *y* axis denotes |fold change|.

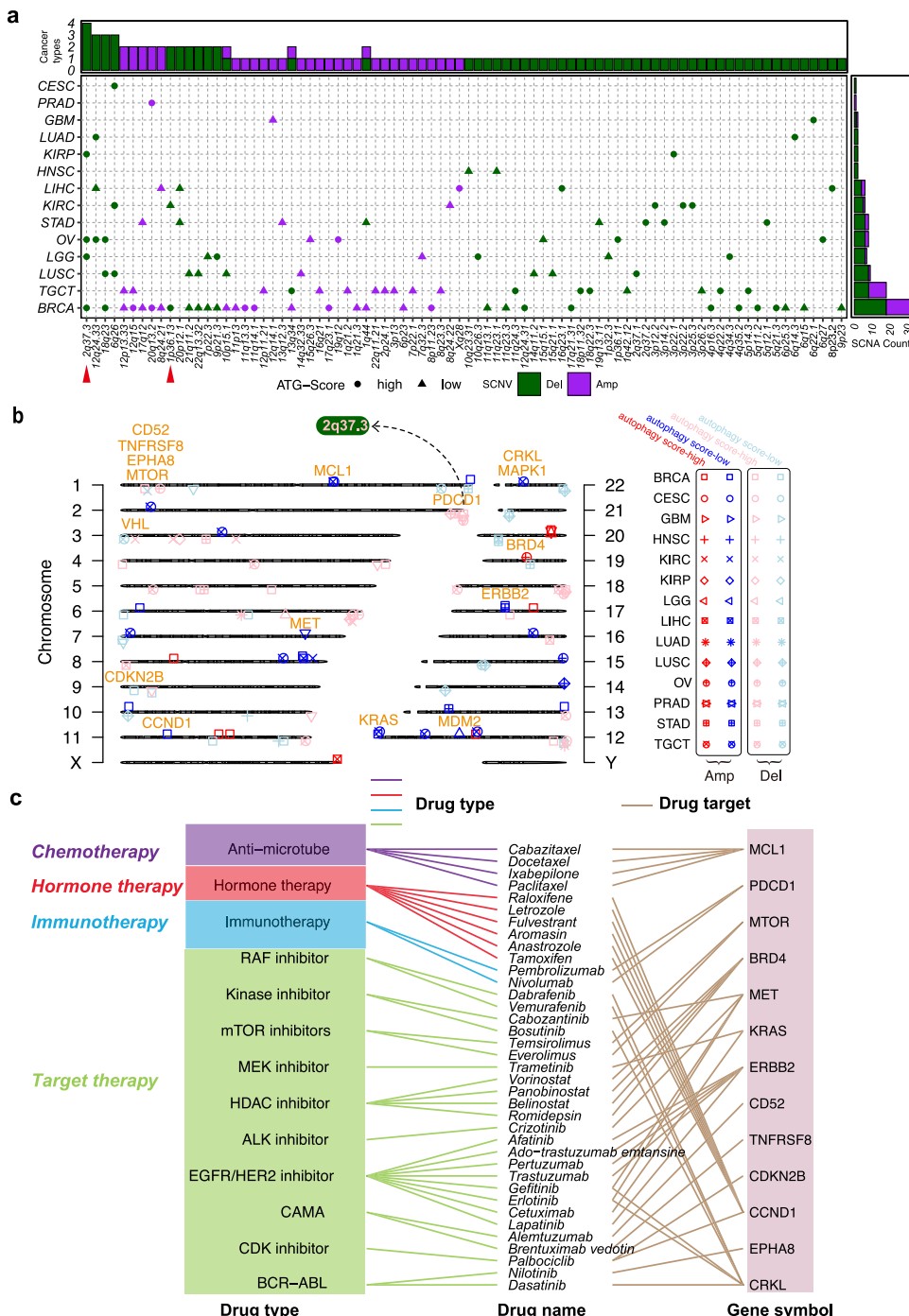

**Fig. 5 | Association between SCNAs and autophagy across multiple cancers. a** Summary characteristics of SCNAs across multiple cancers. The upper bar denotes the number of cancer types with amplifications (purple) or deletions (dark green); the right bar denotes the number of amplifications (purple) or deletions (dark green) in each cancer. Circles indicate the autophagy score-high; triangles indicate the autophagy score-low. **b** Chromosome plot displays locations of autophagy-associated SCNAs with significant alterations or harbored genes in the autophagy score-high group vs. the autophagy score-low group. Red indicates amplification; blue indicates deletion in the autophagy score-high group. Pink indicates amplification; light blue indicates deletion in the autophagy score-low group. The orange label shows CAGs. **c** FDA-approved drugs targeting CAGs in autophagy-associated SCNAs.

etoposide) predicted to be sensitive to SKCM on high autophagy status (Fig. 7a), excluding CGP.082996 due to unavailability. Among these drugs, both A375 and SK-MEL-28 cell lines became significantly sensitive to etoposide and BMS536924 under the rapamycin-induced autophagy conditions (Fig. 7b). We examined the autophagy status of melanoma cells on different conditions by western blot of LC3A/B and p62[68], confirming the activation of autophagy by rapamycin treatment (Fig. 7c). Therefore, etoposide and BMS536924 were chosen

for further experiments in this study. We conducted drug sensitivity assays on selected drugs in three different melanoma cell lines (A375, SK-MEL-28, and SK-MEL-5) under a rapamycin-induced autophagy condition versus a control condition. Our experiments showed that three melanoma cell lines are more sensitive to etoposide and BMS536924 under rapamycin-induced autophagy conditions (Fig. 7b and Supplementary Fig. 5a). Considering other potential anticancer effects of rapamycin on melanoma cell lines, we further utilized

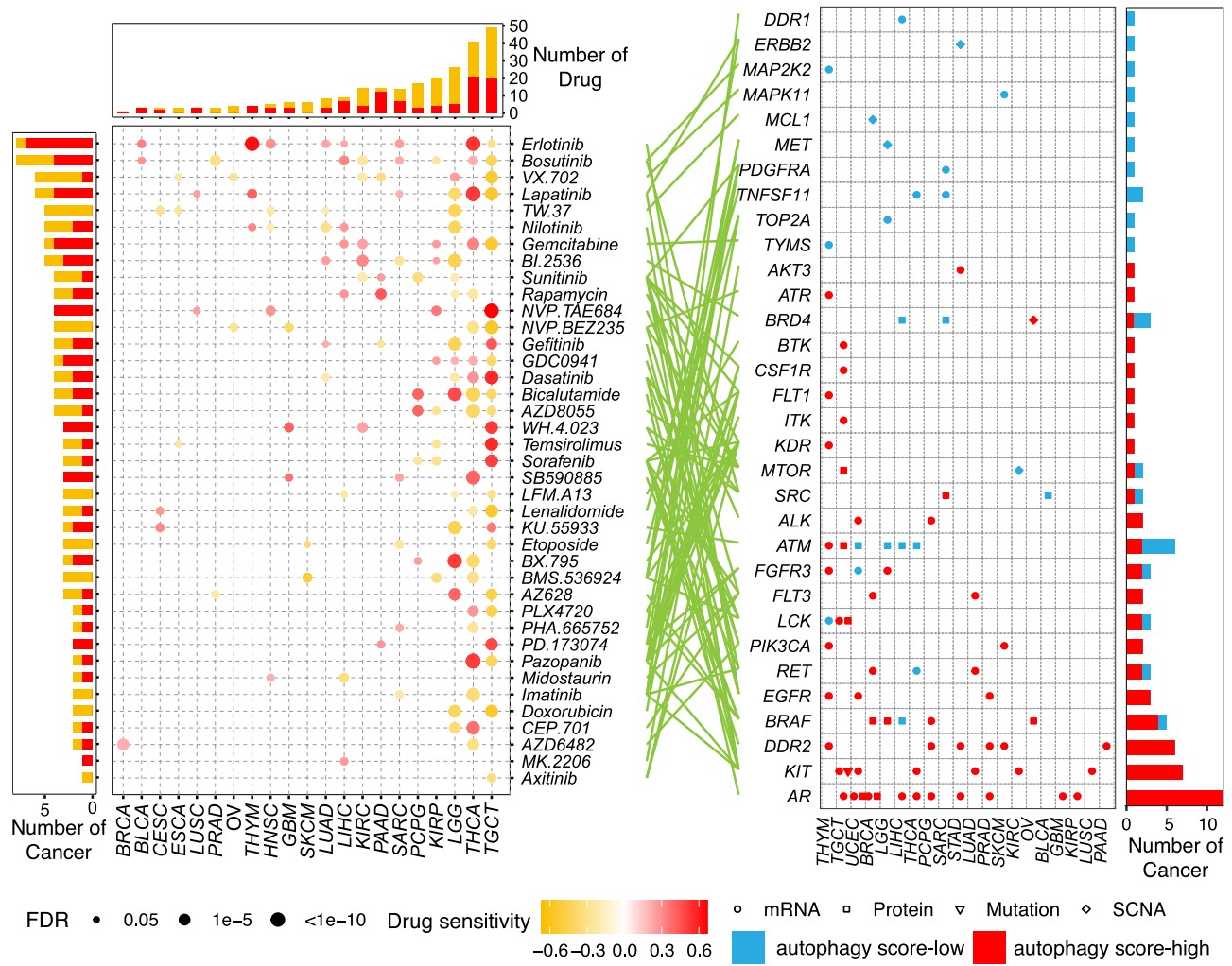

**Fig. 6 | Effects of molecular alterations in clinically actionable genes on drug response.** Sensitivity of drugs targeting clinically actionable genes with any alterations at the mRNA, protein, mutation, and SCNA levels. The point in the left panel indicates the Spearman correlation between imputed drug data and autophagy scores in multiple cancers (red: positive, drug-resistant; orange: negative, drug-sensitive). Different shapes in the right panel represent different types of molecular signatures. Red point: autophagy score-high group; blue point: autophagy score-low group.

starvation (instead of rapamycin) to induce autophagy, and observed that A375, SK-MEL-28, and SK-MEL-5 cells are more sensitive to etoposide and BMS536924 (Fig. 7d and Supplementary Fig. 5b). Moreover, we performed the cell death assays and observed that etoposide and BMS536924 killed more melanoma cells in rapamycin-induced/starvation-induced autophagy conditions (Supplementary Fig. 5c–g). These results suggest that autophagy induction is likely to cause drug sensitivity.

To further confirm the effects of autophagy on the drug response in vivo, we chose etoposide, an anti-tumor drug used in the clinic[69], to create a xenograft model with the A375 cell line, which has a highly similar pattern to SK-MEL-28 in drug sensitivity (Fig. 7b, d). We utilized etoposide alone or in combination with the autophagy inducer rapamycin to treat BALB/C-nu/nu mice inoculated with A375 cells every three days (Fig. 7e). All mice were sacrificed, and the tumor was excised on the 18th day (Fig. 7f). Western blot analysis of autophagy markers, including both LC3A/B and p62, showed that the autophagy status of tumor tissues was elevated by rapamycin (Fig. 7g). We observed that the tumor growth was significantly inhibited by etoposide with the autophagy inducer rapamycin (Fig. 7f), as evaluated by tumor volume at different time points (Fig. 7h), and tumor volume (Supplementary Fig. 5h) and tumor weight (Supplementary Fig. 5i) at the time point when mice were sacrificed.

## The potential mechanism of autophagy inducer sensitizing drug response

To expound how autophagy inducer sensitized tumor cells to etoposide, we performed the RNA sequencing (RNA-seq) analysis for A375 and SK-MEL-28 cells incubated with etoposide versus etoposide + rapamycin. PCA analysis showed that the samples were well clustered (Supplementary Fig. 6a). We identified 946 differentially expressed genes between etoposide versus etoposide + rapamycin in A375 cells, and 782 differentially expressed genes in SK-MEL-28 cells (Supplementary Fig. 6b). Among these genes, there were 290 overlapping differentially expressed genes between etoposide versus etoposide + rapamycin in both A375 and SK-MEL-28 with |fold change| >1.5 and adjusted *p* value <0.05 (Supplementary Fig. 6c, Supplementary Data 6). Based on these 290 overlapping differentially expressed genes (Supplementary Fig. 6b), we further performed KEGG pathway enrichment analysis, and noticed that several downregulated genes (*EGF*, *FGF1*, *BINP3*, *DDIT4*, *PPP2R2B*, *SFN*) were frequently observed in pathways related to cancer or autophagy, such as the PI3K-AKT signaling pathway and mTOR signaling pathway (Supplementary Data 7). We further examined these function-related genes by RT-PCR, and we found that the relative expression of the *DDIT4* was the lowest in the rapamycin + etoposide group compared to rapamycin or etoposide alone (Supplementary Fig. 6d), which was consistent with tumor

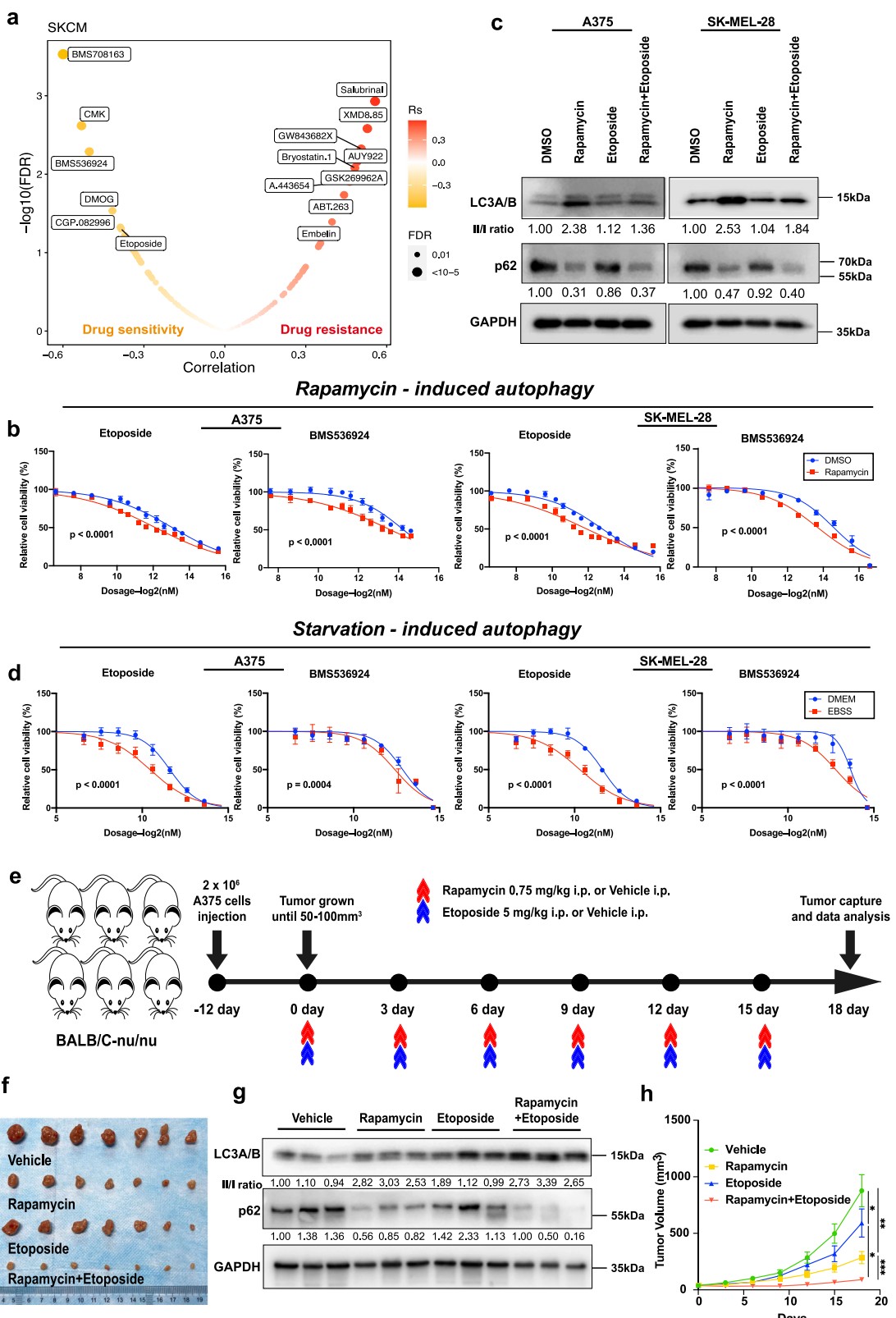

volumes of four groups in a xenograft model (Fig. 7h). RNA-seq showed that *DDIT4* was significantly downregulated in A375 (fold change = −1.60, adjusted *p* value = $1.32 \times 10^{-13}$) and SK-MEL-28 (fold change = −2.34, adjusted *p* value = $2.60 \times 10^{-39}$) when treated with etoposide and rapamycin compared to etoposide alone (Fig. 8a), which was confirmed in A375 and SK-MEL-28 with RT-PCR (Fig. 8b). Moreover, we also observed that the expression of *DDIT4* was

decreased by rapamycin in a public data from GSE27784[70] (Supplementary Fig. 7a).

To examine the functional role of *DDIT4* in autophagy inducer sensitizing melanoma cells to etoposide, we transfected three *DDIT4* siRNAs (si-DDIT4) into A375 and SK-MEL-28 cells treated with etoposide, respectively. The DDIT4 expression was decreased in the etoposide + si-DDIT4-2 or si-DDIT4-3 and the etoposide + rapamycin cells, as

**Fig. 7 | Characterization of the drug response associated with autophagy in vitro *and* in vivo. a** Drug response associated with the autophagy status in SKCM. The orange point indicates sensitivity; the red point indicates resistance. **b, d** Dose-response curves for the mean value of cell viability of etoposide and BMS536924 in rapamycin-induced (**b**) or starvation-induced (**d**) and non-induced conditions in the melanoma cell line A375 and SK-MEL-28. Cell viability was normalized to the level of cells treated with DMSO (**b**) or DMEM (**d**). DMEM: Dulbecco's Modified Eagle Medium for control. EBSS: Earle's Balanced Salt Solution for starvation. Error bars indicate the mean ± SD. The drug screen data of different groups ($n = 4$) were fitted and compared by sigmoidal dose-response curves. **c** Western blot of autophagy markers in A375 and SK-MEL-28 cell lines treated with DMSO, rapamycin, etoposide, and rapamycin + etoposide. The ratios of the LC3-II/LC3-I and p62 band intensities normalized to DMSO are displayed below the blots. Similar results were observed in at least three independent experiments. **e** The design of xenograft experiments. Nude mice were injected with $2 \times 10^6$ A375 cells. When the tumor size

reached 50–100 mm³, mice were treated with vehicle, 0.75 mg/kg rapamycin intraperitoneally (i.p.), 5 mg/kg etoposide (i.p.), or the combination of rapamycin and etoposide every 3 days ($n = 7$). Mice were depicted using "mouse-animal-rodent-mammal-little" (https://pixabay.com/zh/vectors/mouse-animal-rodent-mammal-little-310756/), available under the Simplified Pixabay License (https://pixabay.com/zh/service/license/). **f** Excised tumors to represent the tumor size on day 18. **g** Western blot of autophagy markers in mouse tissues with the treatment of vehicle, rapamycin, etoposide, and rapamycin + etoposide. The ratios of the LC3-II/LC3-I and p62 band intensities normalized to vehicle are displayed below the blots ($n = 3$). **h** Quantitative analysis of tumor volume at different time points with the treatment ($n = 7$). Error bars indicate the mean ± SD. The difference in multiple groups was estimated by one-way ANOVA analysis. *P* values are as follows: $p = 0.011$ (Vehicle vs. Etoposide), $p = 0.0012$ (Vehicle vs. Rapamycin), $p = 0.026$ (Etoposide vs. Rapamycin+Etoposide), and $p = 0.00058$ (Rapamycin vs. Rapamycin+Etoposide). **c, g** See Supplementary Fig. 8 for uncropped data.

shown by western blot (Fig. 8c). The relative cell viability of A375 and SK-MEL-28 was significantly inhibited with etoposide and si-DDIT4-2 or si-DDIT4-3 (Fig. 8d). In addition, A375 and SK-MEL-28 cells treated with etoposide and rapamycin were infected with DDIT4-OE lentivirus or negative control lentivirus, and the relative expression of DDIT4 was increased in etoposide + rapamycin + DDIT4-OE lentivirus-treated cells compared to etoposide + rapamycin + negative control lentivirus-treated cells, as shown by western blot (Fig. 8e). The cell viability was restored when DDIT4 was overexpressed (Fig. 8f). The efficiency and specificity of *DDIT4* siRNAs were examined in A375 and SK-MEL-28 (Supplementary Fig. 7b), and the autophagy status of cells when *DDIT4* was knocked down was also examined (Supplementary Fig. 7c). The efficiency of DDIT4-OE lentivirus was examined in A375 and SK-MEL-28 (Supplementary Fig. 7d), and the autophagy status was also examined (Supplementary Fig. 7e).

Moreover, we inhibited autophagy by transfecting three *ATG5* siRNAs (si-ATG5) into A375 and SK-MEL-28 cells treated with etoposide, respectively, and the efficiency of ATG5 siRNAs was examined by western blot (Supplementary Fig. 7f). The relative cell viability of A375 and SK-MEL-28 was significantly increased with etoposide and si-ATG5-1 (Fig. 8g). The expression of DDIT4 was also significantly increased when si-ATG5-1 was transfected into A375 and SK-MEL-28 cells treated with etoposide, respectively (Fig. 8h). Taken together, our results suggest that melanoma cells may be more sensitive to etoposide when present with an autophagy inducer, and this pattern may be mediated through *DDIT4* (Fig. 8i).

## Discussion

Autophagy can promote tumorigenesis and is associated with the cell response to immunotherapy, hormone therapy, and target therapy. A comprehensive analysis of multi-omics and therapeutic responses associated with autophagy will significantly contribute to cancer therapy. Despite the critical roles of autophagy, there are no applicable methods for estimating autophagy status in a large number of patients. In this study, we proposed to define the autophagy status of patients based on an autophagy gene signature, which facilitates the investigation of the functional roles of autophagy in patients. We then classified all TCGA cancer samples into autophagy score-high groups and autophagy score-low groups based on the autophagy signature. To explore the effects of autophagy on biological molecules in cancers, we applied a propensity score algorithm to minimize the disturbance of various confounding factors: tumor purity, sex, ethnicity, age at diagnosis, and smoking status. We identified multiple autophagy-related molecular features across 24 cancer types, including mRNA, miRNA and protein expression, somatic mutations, and SCNAs. In addition, we explored the influence of the regulatory network among different molecules on the autophagy process, which is the most comprehensive landscape thus far. This comprehensive map will provide strong biological insights for future investigations, which has

been demonstrated by previous rigorous analysis of large-scale data, including TCGA and ICGC[22,34,37,38,41,71–77]. Our integrative analysis suggests that autophagy status can impact the alterations of molecular features in multi-omics layers in diverse tumors.

Notably, we found a total of 32 CAGs were significantly altered across 20 cancer types, and multiple cancer types can be resistant or sensitive to the drugs targeting these CAGs upon autophagy induction (Fig. 6). In contrast to the conventional view that autophagy inhibition confers drug sensitivity, our comprehensive analysis and in vitro/in vivo experiments highlight an opportunity to leverage personalized molecular feature analysis to overcome multiple drug resistance induced by autophagy. It is important to note, however, that only with single experiments (e.g., single drug or cell line), it will be very challenging to conclude that autophagy may drive the drug sensitivity for an appreciable number of anticancer drugs. Our analysis further suggested that *DDIT4* is a potential target to mediate the autophagy induction to sensitize tumor cells to etoposide. Recent studies have shown that the immune system is weakened by autophagy inhibition due to decreased degradation of immune metabolites[78,79], which further highlights the need for caution when utilizing autophagy inhibitors, especially in immunotherapy. Therefore, the combination of autophagy inducers and immunotherapy may provide an alternative direction for cancer therapy.

Challenges remain in the study of autophagy that requires further investigation. We estimated the relative autophagy status across tumor samples, which may be limited in its ability to reflect the real status of autophagy in patients. Autophagy is a dynamic biological process, which involves the regulation of various related molecules. Therefore, it is difficult to evaluate it accurately by the evaluation of only a small number of autophagy-related markers, such as LC3 or P62, etc. Although autophagy flux can be used to monitor autophagy status, this approach is challenging even in cultured cell lines and model organisms[80], not to mention in large-scale (~10,000) patient samples. Similar to this, there is no direct approach to assess the hypoxia status in patients, while we can only indirectly access the hypoxia status through a 15-gene signature[22,25,81,82], which can still lead to significant biological discoveries. Our analysis is based on bulk RNA-seq across different cell types within a sample. Further efforts should take tumor heterogeneity into consideration, particularly with advancements in single-cell profiling technology. Finally, most clinical trials do not have information on the autophagy status of patients' cancer samples due to the technical challenge of monitoring autophagy status in vivo. Currently, most studies have focused on autophagy inhibition, while the importance of autophagy induction has been ignored, which leads to a limited understanding of how molecular signatures are affected by the autophagy microenvironment and a reasonable interpretation of unexpectedly adverse drug outcomes in anticancer therapy. Nonetheless, our study emphasizes the significance of monitoring tumor autophagy status in future clinical studies.

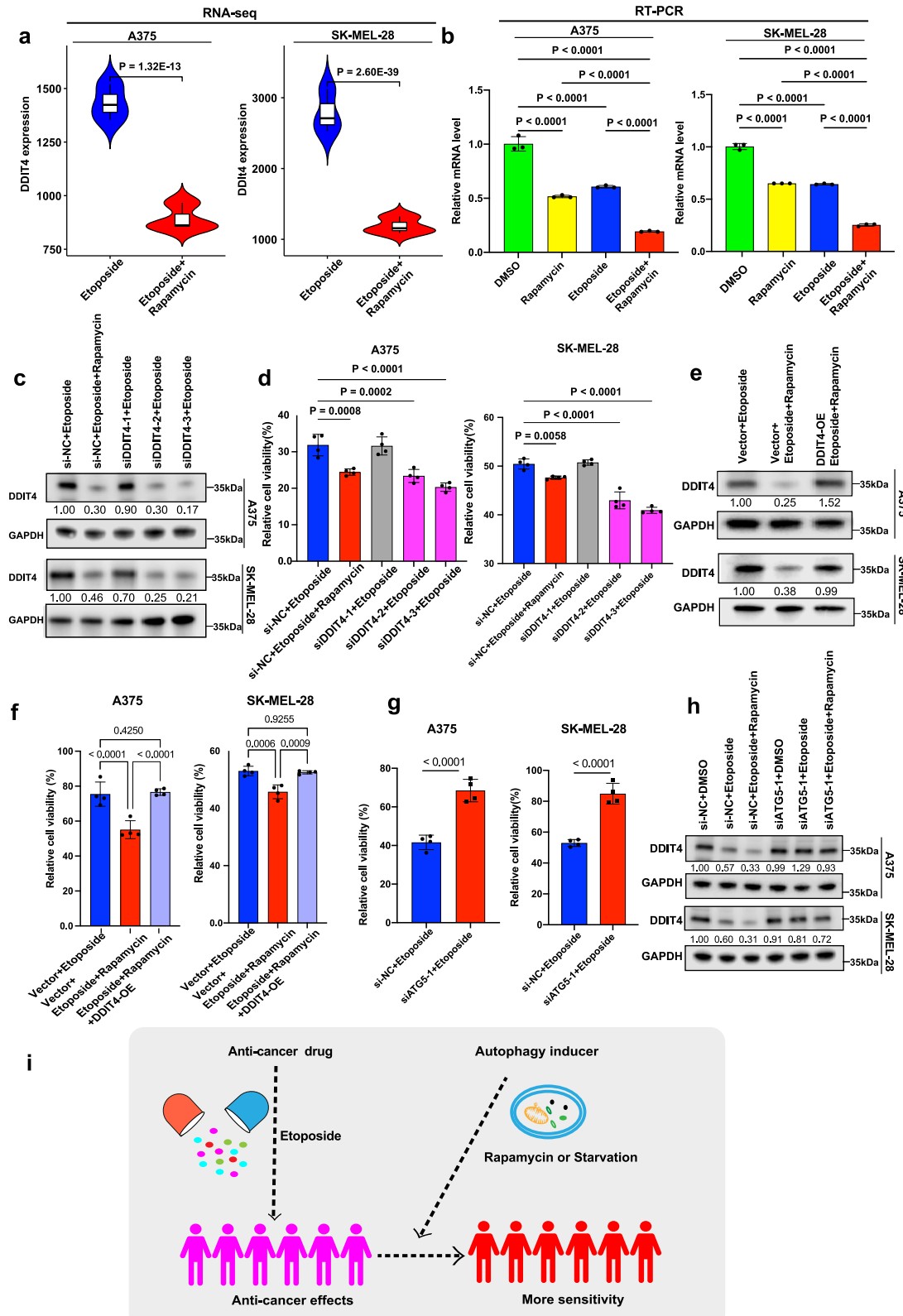

## Methods

### Identification of a gene signature for estimating autophagy status across cancer samples

We collected data from six currently widely used autophagy-related databases, in which autophagy-associated genes were manually collected from the literature related to autophagy. We downloaded six independent datasets with known autophagy status from the Gene Expression Omnibus (GEO): GSE107600[83], GSE117189[84], GSE129204[85], GSE106175[86], GSE90444[87], and GSE31397[88]. For each human gene list from the autophagy-related database, we calculated the ssGSEA score of samples from six datasets using the GSVA package[89]. We retained gene lists for which the score level was consistent with the autophagy status of samples in at least four validation datasets. We further obtained the intersection of these four gene lists, which included 37

**Fig. 8 | Potential mechanism through the functional characterization of *DDIT4* in vitro. a** The mRNA expression of *DDIT4* in A375 and SK-MEL-28 by RNA-seq ($n = 3$). The boxes show the median ±1 quartile, with whiskers extending to the most extreme data point within 1.5 interquartile range from the box boundaries. The significance ($p$ value) of differentially expressed genes was evaluated with 'DESeq2' package. **b** Relative mRNA expression of *DDIT4* in A375 and SK-MEL-28 by RT-PCR ($n = 3$). **c** Western blot of DDIT4 in A375 or SK-MEL-28 cells treated with si-NC or *DDIT4* siRNAs in combination with etoposide. DDIT4 band intensities normalized to si-NC + etoposide are displayed below the blots. **d** The cell viability of A375 or SK-MEL-28 cells treated with si-NC or *DDIT4* siRNAs in combination with etoposide ($n = 4$). Pink bars denote the successful knockdown of si-*DDIT4* (si-DDIT4-2 and si-DDIT4-3), while gray bar denotes the unsuccessful knockdown of si-*DDIT4* (si-DDIT4-1). **e** Western blot of DDIT4 in A375 or SK-MEL-28 cells transfected with DDIT4-OE lentivirus or vector in combination with etoposide and rapamycin.

DDIT4 band intensities normalized to Vector+Etoposide are displayed below the blots. **f** The cell viability of A375 or SK-MEL-28 cells transfected with DDIT4-OE lentivirus or vector in combination with etoposide and rapamycin ($n = 4$). **g** The cell viability of A375 or SK-MEL-28 cells treated with si-NC or *ATG5* siRNA in combination with etoposide ($n = 4$). **h** Western blot of DDIT4 in A375 or SK-MEL-28 cells treated with si-NC or *ATG5* siRNAs in combination with etoposide. DDIT4 band intensities normalized to si-NC + DMSO are displayed below the blots. **i** The illustration for the potential mechanism of autophagy inducer sensitizing drug response. **b, d, f, g** Data were presented as means ± SD. **b, d, f** The difference in multiple groups was estimated by one-way ANOVA analysis. **g** Two-sided student's t test was used for the estimation of the difference in two groups. *$P < 0.05$; **$P < 0.01$; ***$P < 0.001$; ****$P < 0.0001$; ns is not significant. **c, e, h** See Supplementary Fig. 8 for uncropped data.

autophagy-associated genes (Supplementary Data 1). We confirmed that the autophagy scores based on the 37-gene list are consistent with the autophagy status in six datasets from GEO. The autophagy score for each tumor sample or cancer cell line was defined by the ssGSEA score based on the 37-gene list.

## Classification and multi-omics analysis of tumor samples from TCGA

The multi-omics datasets (mRNA expression, miRNA expression, protein expression, somatic mutations, and SCNA) and clinical data (age, gender, smoking, etc.) from 32 tumor types were downloaded from the TCGA data portal (https://portal.gdc.cancer.gov/). The tumor purity of TCGA samples was downloaded from Tumor IMmune Estimation Resource (http://cistrome.org/TIMER/download.html)[90] and https://doi.org/10.5281/zenodo.253193[91] and integrated. The autophagy scores of tumor samples from 32 tumors were calculated based on the 37-gene autophagy signature. We classified tumor samples based on the distribution of tertiles, defining the top and bottom samples as autophagy score-high and autophagy score-low samples, respectively. We retained a total of 24 cancer types with ≥30 samples in both autophagy score-high and autophagy score-low groups for further analysis.

We used the matching weights (MW) method of the PSM algorithm to balance the effects of potential confounders, including age at diagnosis, gender, tumor purity, race, histological type, and tumor stage, and examined the balance by comparing the standardized difference before and after PSM (standardized difference <0.1). The Benjamini & Hochberg method was used to calculate the FDR in each cancer type. Subsequently, we compared the molecular difference of multi-omics data between the autophagy score-high and autophagy score-low in 24 cancer types with every group's sample number >30. To decrease the random noise in feature identification, we repeated the permutation test 100 times, randomly selecting the autophagy score-high or autophagy score-low samples. Significant features for the five molecular types in each cancer type were identified by the following criteria: mRNA expression |fold change| >2, FDR < 0.05; miRNA expression |fold change| >1.5, FDR < 0.05; total protein and DNA methylation: |difference| >0.2, FDR < 0.05; somatic mutation and SCNA: FDR < 0.05.

## Pathway enrichment and miRNA-target and TF-target regulatory networks

We performed KEGG pathway enrichment analysis by using the "clusterProfiler" package[92] with default parameters. The gene list of human TFs was downloaded from the AnimalTFDB 3.0[93], TF-target pairs were downloaded from the hTFtarget database[94], and miRNA-target pairs were downloaded from the FFLtool[95]. We matched the TFs from the altered mRNAs (as targets) and identified the TF-target regulatory pairs based on the downloaded TF-target pairs. The significantly altered miRNAs and mRNAs (as targets) were used to identify

the miRNA-target relationships. Based on the TF-target pairs and miRNA-target pairs, we constructed TF-target regulatory networks and miRNA-target regulatory networks, where the nodes are the TFs, miRNAs, or genes, and the edges are the regulatory pairs.

## Analysis of drug response in autophagy status

The AUC data and the gene expression matrix for cancer cell lines were downloaded from GDSC (http://www.cancerrxgene.org/downloads). The imputed drug response of 138 anticancer drugs in TCGA cancer patients was downloaded from a previous study[64]. The information on clinically actionable genes targeted by FDA-approved drugs was downloaded from a previous study[96]. The drug repurposing information with drug target was downloaded from The Drug Repurposing Hub (https://clue.io/repurposing-app)[97]. To assess the drug response in cancer cell lines, we calculated the Spearman correlation between the AUC and gene expression of cancer cell lines from GDSC for drug responsiveness ($|Rs| > 0.3$; FDR < 0.05). As in our previous study[22], a positive Spearman correlation was defined as drug-resistant, while a negative Spearman correlation was defined as drug-sensitive. To assess the drug response in TCGA tumor samples, we calculated the Spearman correlation between the imputed drug data and autophagy score to assess the drug response of TCGA cancer samples according to autophagy status ($|Rs| > 0.2$; FDR < 0.05).

## Cell Culture

The human malignant melanoma cell lines, A375, SK-MEL-28, and SK-MEL-5 cell lines were obtained from American Type Culture Collection (ATCC) and were cultured in Dulbecco's Modified Eagle Medium (DMEM; Biological Industries) supplemented with 10% fetal bovine serum (FBS; Biological Industries) and 1% penicillin and streptomycin (Biological Industries) at 37 °C in 5% $CO_2$ (v/v). To induce autophagy, cells were incubated with the starvation medium Earle's Balanced Salt Solution (EBSS; E2888, Sigma-Aldrich) instead of the normal medium for 24 h.

## Cell proliferation assay for drug sensitivity

We purchased CMK (HY-52101) from MedChemExpress, and etoposide (S1225), BMS536924 (S1012), BMS708163 (S1262), DMOG (S7483) and rapamycin (S1039) from Selleckchem. The effects of the drugs on cell proliferation were determined using a CellTiter 96® AQ$_{ueous}$ One Solution Cell Proliferation Assay kit (Promega) according to the manufacturer's instructions. We plated 3000 cells in each assay of 96-well plates. One day later, we treated cells with a range of drug concentrations prepared by serial dilution plus 200 uM rapamycin or dimethyl sulfoxide (DMSO; four replicates per condition). The plates were incubated at 37 °C and 5% $CO_2$ (v/v). After 2 days, 20 μl of the CellTiter 96® AQ$_{ueous}$ One Solution Reagent was directly added to the culture wells, and incubated for 1 h, and then the absorbance was recorded at 490 nm with Epoch (Biotek). The relative growth was normalized to the untreated samples in each group.

## Cell death and flow cytometry

Cell death was quantified by the percentage of propidium iodide (PI; P4170, Sigma-Aldrich) positive staining cells detected by flow cytometric analysis. $2 \times 10^5$ cells per well were seeded in 6-well plates and treated with indicated drugs. After harvesting the cells, they were washed with phosphate-buffered saline (PBS) twice and then incubated with antibodies on ice for 30 min in the dark. Flow cytometric analysis was conducted on a FACS LSR II Fortessa (BD Biosciences) and the FACS data were analyzed with FlowJo software (Tree Star).

## RNA isolation and quantitative real-time PCR assay

Total RNA was extracted from cultured cells and tumor tissues using the MagZol reagent (Magen) following the manufacturer's instructions. RNA (1 μg) was used to synthesize cDNA using HiScript II Q RT SuperMix for qPCR (Vazyme), and then cDNA was amplified by RT-PCR with specific primers using Quant Studio 3 (Thermo Scientific). All mRNA expression levels were normalized to GAPDH and calculated using the $2^{-\triangle\triangle CT}$ method. Human *DDIT4* primer forward sequence: 5′-TGAGGATGAACACTTGTGTGC-3′, reverse sequence: 5′-CCAACTGGC TAGGCATCAGC-3′. Human *GAPDH* primer forward sequence: 5′-GGAGCGAGATCCCTCCAAAAT-3′, reverse sequence: 5′-GGCTGTTGTC ATACTTCTCATGG-3′. Human *PIK3R3* primer forward sequence: 5′-TACAATACGGTGTGGAGTATGGA-3′, reverse sequence: 5′-TCATTG GCTTAGGTGGCTTTG-3′.

Human *EGF* primer forward sequence: 5′-TGTCCACGCAA TGTGTCTGAA, −3′ reverse sequence: 5′-CATTATCGGGTGAGGAAC AACC-3′.

Human *FGF1* primer forward sequence: 5′-ACACCGACGGGCTTTT ATACG-3′, reverse sequence: 5′-CCCATTCTTCTTGAGGCCAAC-3′.

Human *BNIP3* primer forward sequence: 5′-CAGGGCTCCTGGGT AGAACT-3′, reverse sequence: 5′-CTACTCCGTCCAGACTCATGC-3′.

Human *PPP2R2B* primer forward sequence: 5′-CCATGAACCCG AGTTCGATTAC-3′, reverse sequence: 5′-GGCCCTCCTCATCTTTCAG ATT-3′.

Human *SFN* primer forward sequence: 5′-TGACGACAAG AAGCGCATCAT-3′, reverse sequence: 5′-GTAGTGGAAGACGGAAAAG TTCA-3′.

## Western blot analysis

Cells were washed twice with PBS and lysed in RIPA buffer (P0013C, Beyotime) containing 1× Protease Inhibitor Cocktail (B14002, Bimake) and 1× Phosphatase Inhibitor Cocktail (B15002, Bimake) on ice for 30 min. Then, the cells were centrifuged for 10 min at 14,000 rpm. Supernatants were collected and proteins were quantified by a BCA assay kit (P0010S, Beyotime). Equal amounts of proteins were loaded onto SDS-PAGE gels. The following antibodies were used: anti-DDIT4 (10638-1-AP, Proteintech, 1:1000), anti-ATG5 (12994 T, Cell Signaling Technology, 1:1000), LC3A/B (12741 S, Cell Signaling Technology, 1:1000), SQSTM1/P62 (8025 S, Cell Signaling Technology, 1:1000), anti-GAPDH (60004-1-Ig, Proteintech, 1:50,000), HRP goat anti-mouse IgG (H + L) (AS003, ABclonal, 1:50,000) and HRP goat anti-rabbit IgG (H + L) (AS014, ABclonal, 1:50,000). Protein bands were evaluated by Image J. Protein levels were quantified relative to GAPDH in the same sample, and the relative protein expression was normalized to the respective control group, which was set to 1.

## RNA interference and lentivirus transfection

Human *DDIT4* siRNA-1 (5′GAUGAACACUUGUGUGCCATTGGCACA CAAGUGUUCAUCUU-3′), Human *DDIT4* siRNA-2 (5′GGAAUAGU GUUUCCCAGGATTCCUGGGAAACACUAUUCCUU-3′), Human *DDIT4* siRNA-3 (5′GUUUGUGUAUCUUACUGGUUACCAGUAAGAUACACAA ACTT-3′) and control empty siRNA were purchased from GenePharma. Human *ATG5* siRNA-1 (5′GGACGAAUUCCAACUUGUUUAACAA GUUGGAAUUCGUCCUU-3′), Human *ATG5* siRNA-2 (5′GGAAGCA GAACCAUACUAUUUAUAGUAGGUUCUGCUUCCUU-3′), Human *ATG5*

siRNA-3 (5′CCAUCAAUCGGAAACUCAUUUAUGAGUUUCCGAUUGAU GGTT-3′) and control empty siRNA were purchased from GenePharma. These siRNAs were transfected into cells using TurboFect Transfection Reagent (R0531, Thermo Scientific) according to the manufacturer's instructions, respectively. The human *DDIT4* overexpression lentivirus vector and control empty lentivirus vector was obtained from Genechem. The stable overexpression cells were selected by adding puromycin.

## Xenograft experiments

Animals were housed under specific pathogen-free conditions. The housing conditions were strictly following the ethical regulations (ambient temperature of 22–25 °C; relative humidity of 50–60%; 12 h/ 12 h light/dark cycle; ad libitum access to food and water). For the cell line xenograft models, 4-week-old nude male mice (from the SLAC, Shanghai, China) were subcutaneously implanted with $2 \times 10^6$ A375 cells on the right of the dorsal midline. Once the tumors reached 50–100 mm³, the mice were pooled and randomly allocated into four groups ($n = 7$, the vehicle group, the rapamycin group, the etoposide group, and the etoposide + rapamycin group). Mice were treated with rapamycin (0.75 mg/kg, intraperitoneal injection [i.p.]), etoposide (5 mg/ kg, i.p.), combination therapy, or a single drug only every 3 days. Tumors were measured by calipers every 3 days, and volumes were calculated using the formula length × width² × 1/2. All animal experiments were approved by the Animal Care and Use Committee of Central South University (Changsha, Hunan, China). The maximal tumor size permitted by the ethics committee was 2000 mm³, and it was not exceeded in our animal experiment. The procedures for all animal experiments were approved by the Ethical Committee for Animal Research of Xiangya Hospital, Central South University (Code: 2022020466).

## RNA sequencing and differential expression analysis

RNA was extracted from 12 samples from two melanoma cell lines using MagZol reagent with a standard protocol. The mRNA polyA-based enrichment method was performed to establish the RNA library. RNA sequencing was performed by Illumina NovaSeq 6000 at Novogene-Tianjin, China (Supplementary Data 5). Clean reads were mapped to the grch38 genome by HISAT (version 2-2.2.0); the Samtools (version 1.3.1) was used to convert the SAM to BAM format; the StringTie (version 2.1.4.Linux_x86_64) was used to estimate the gene expression by the annotation file "Homo_sapiens.GRCh38.90.gtf". The differentially expressed genes were detected by the R package "DESeq2" with | fold change| >1.5 and adjusted $p$ value < 0.05. All RNA-seq data have been deposited in the GEO (accession: GSE185814). The enrichment analysis of differentially expressed genes was performed by the "clusterProfiler" package[92] with default parameters.

## Statistics and reproducibility

The statistical analysis of experimental data was performed by GraphPad Prism version 9.0 (GraphPad software). The drug screen data of different groups were fitted and compared by sigmoidal dose-response curves. Data were presented as the means ±SD. The difference in multiple groups was estimated by one-way ANOVA analysis and Student's $t$ test was used for the estimation of the difference in only two groups. (*) $P < 0.05$, (**) $P < 0.01$, (***) $P < 0.001$, (****) $P < 0.0001$; ns is not significant. All data were obtained from at least three independent biological replicates.

## Reporting summary

Further information on research design is available in the Nature Research Reporting Summary linked to this article.

## Data availability

The raw data of RNA-seq in this study were deposited in Sequence Read Archive [SRA, https://www.ncbi.nlm.nih.gov/sra/] under

accession ID SRP341179. The processed RNA-seq dataset was deposited in Gene Expression Omnibus [GEO, https://www.ncbi.nlm.nih.gov/geo/] under accession ID GSE185814. The public expression datasets were obtained from GEO, including GSE27784, GSE107600, GSE117189, GSE129204, GSE106175, GSE90444, and GSE31397. The multi-omics datasets (mRNA expression, miRNA expression, protein expression, somatic mutations, and SCNA) and clinical data (age, gender, smoking, etc.) from 32 tumor types were downloaded from the TCGA data portal [https://portal.gdc.cancer.gov/]. The tumor purity of TCGA samples was downloaded from Tumor IMmune Estimation Resource [http://cistrome.org/TIMER/download.html] and [https://doi.org/10.5281/zenodo.253193] and integrated. The drug repurposing information with drug target was downloaded from The Drug Repurposing Hub [https://clue.io/repurposing-app].Uncropped scans of all blots underlying Figs. 7c, g, 8c, e, h and Supplementary Fig. 7b–f are provided as Supplementary Fig. 8. Intensity values estimated by Image J are provided in western blots of main figures (Fig. 7c, g, Fig. 8c, e, h) and supplement figures (Supplementary Fig. 7b–f) Source data are provided with this paper.

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

## Acknowledgements

This work was supported by the National Key Research and Development Program of China (no. 2019YFE0120800 and no. 2019YFA0111600 to H. L., no. 2021YFF0703704 to A.G.), the Natural Science Foundation of China for outstanding Young Scholars (no. 82022060) to H.L., the Science Foundation for Distinguished Young Scholars of Hubei Province of China (2020CFA070) to A.G., the Natural Science Foundation of Hunan Province for outstanding Young Scholars (no. 2019JJ30040) to H.L., Talent Young Scholars of Hunan Province (no. 2019RS2009) to H.L., the China Postdoctoral Science Foundation (2022M713541 to M.L., 2020M682587 to G.Z.), the National Natural Science Foundation of China (no. 62102455) to G.Z., the Hunan Outstanding Postdoctoral Innovative Talents Program (2021RC2035) to G.Z. We gratefully acknowledge contributions from TCGA Research Network. We thank LeeAnn Chastain for editorial assistance.

## Author contributions

L.H., H.L., A.G., and X.C. conceived and supervised the project. M.L. and L.H. designed and performed the research. M.L., Y.Y., Z.Z., C.L, S.L., Y.J., H.R., G.Z., Y.H., Y.L, and Y.X. performed the data analysis. L.Y., and R.C. performed the experimental assays. M.L. and L.H. wrote the manuscript with input from all other authors.

## Competing interests

The authors declare no competing interests.
