## [Peer review file · Nature Communications]

REVIEWER COMMENTS

Reviewer #1 (Remarks to the Author): Expert in bioinformatics, drug response prediction, and multi-omics

In this paper, Luo et al. estimated the activity of autophagy in 33 cancer types from TCGA using a 37-gene signature constructed on the basis of six independent expression datasets with known autophagy status. They have shown that in many cancers ssGSEA-based autophagy scores vary widely and subsets of tumors with high and low autophagy scores are enriched with a number of molecular alterations at the level of the gene, protein, and miRNA expressions, methylation, and somatic mutations.

The main finding of this work is an unexpectedly high correlation between autophagy scores and predicted tumor sensitivities to some drugs. Since this contradicts previous knowledge on autophagy as a drug resistance mechanism, Luo et al. decided to go beyond in silico analysis and performed additional experiments in vitro and in vivo. They have shown that sensitivities of two cell lines to two drugs, etoposide and BMS536924, can be modified by (rapamycin-induced) autophagy and proposed a mechanistic explanation of this effect. They compared dose-response curves for cell lines treated with etoposide and BMS536924 alone and in combination with rapamycin, which is an inducer of autophagy. Analysis of RNA-seq profiles from these cell lines treated with etoposide and etoposide + rapamycin revealed 84 genes significantly differentially expressed between two groups. One of these 84 genes, DDIT4, was shown to be a mediator of rapamycin-induced autophagy effect on etoposide response. Its knockdown and overexpression decreased and increased cell viability, respectively. Finally, the authors created mice xenograft models and confirmed that the combination with rapamycin reinforces the effect of etoposide in vivo too.

The manuscript reports an interesting finding with a potential to advance anticancer treatment approaches and therefore is of interest to the community. I think that methods description, quality of figures, structure, and overall readability of the manuscript could be improved.

Structure of the manuscript

I would recommend extending the experimental validation part and making it more central in the paper. Figure 7 contains too many details.

The sections describing the associations between omics data could be reduced and/or moved to supplementary. As far as I understood, the proposed mechanism of action is explained with the use of expression data only. If it is not the case, the role of other omics layers in this mechanism should be explained more clearly.

I would add a section discussing the limitations of this work

Methods

Unclear, which statistical tests were used to detect associations of autophagy scores with features and how FDR was computed.

How many features were tested in total?

Method descriptions in "Pathway enrichment and miRNA-target and TF-target regulatory network" are missing necessary details, tool versions, parameters. Were the tools run with default parameters? What kind of problem did each tool solve?

Why etoposide and BMS536924 were chosen for further experiments among all drugs demonstrating a high correlation of predicted response with autophagy score?

How did the authors choose DDIT4 among 84 differentially expressed genes?

Why the functional role of DDIT4 was examined in SK-MEL-28, but A375 was used to create xenografts?

Figures

In Fig. 1C: "Student's t-test was used to assess the difference;" - isn't Mann-Whitney U test more appropriate for such small groups?

Fig. 1D:

- unclear, what does the absence of rectangle mean, e.g. for HAMDB and ACC?
- presenting R as a number and $-\log(\text{adj. p-value})$ as color or size would greatly simplify the figure

Fig. 2B:

Interesting, that for miRNA, mutations, and sCNA the number of features associated with low autophagy score was much higher than the number of features associated with high autophagy.

Fig. 3A: legend label should be $-\log(\text{adj. p-value})$, not just p-value.

Fig. 3B: FDR is not on a log-scale.

Figures 4A and B: y-axis label for barplots is missing

Figure 5B: figure legend contains unnecessary repetitions

Figure S4: the number of drugS

Other comments and corrections

Please correct the database name, it should be ncRDeathDB instead of ncDeathDB.

"We validated the 37-gene list in six independent datasets." - I would not call this validation because the same datasets were used to build this gene list. I would reformulate it e.g. "we confirmed that the 37-gene list is consistent with autophagy status in six datasets from GEO."

page 6: "For example, several cancer types, ...showed a larger number of variations between the autophagy score-high and autophagy score-low groups (Figure 2B), suggesting a greater impact of autophagy in these cancer types". I do not think that drawing the conclusion about autophagy impact just from the number of associated alterations is correct.

What if these cancer types are more heterogeneous than the others?

Please define in the main text "genes enriched in cancer types" (2nd sentence of Associations between autophagy and mRNA/ protein expression and signaling pathways, page 6)? I could not find any explanation in main text, it appears only in Fig. 3 caption.

page 7:

- Why did the authors use drug responses from GDSC with omics data from CCLE? GDSC provides several versions of multi-omics profiles for tested cell lines.

- The second paragraph does not agree with the content of "Analysis of drug response in autophagy status" on page 15. According to Methods, expression profiles of cell lines were obtained from GDSC, not CCLE. The authors should clearly write which source of expression data they used.

- CCLE and GDSC datasets are not referenced

page 7: The authors write about "genes ... sensitive to anti-cancer drugs (Figure 3C)". How do they define genes sensitive or resistant to drugs?

page 8: "These genes targeted by miRNAs were enriched in multiple autophagy-related pathways, including the PI3K-AKT and MAPK signaling pathways (Figure 4A, FDR < 0.05; Figure S3C, Table S3)." - the details on the statistical test used are missing. Shouldn't it be p-value reported for this single test instead of FDR?

page 12: "sensitizing etoposide to autophagy inducer"

page 13: "32 CAGs to be significantly altered across 20 cancer types, which were resistant or sensitive in multiple cancer types" on page 13.

The last paragraph in "The potential mechanism of autophagy inducer sensitizing drug response" section is hard to understand.

page 14: "from 32 TCGA tumor samples" - the authors probably mean 32 tumor types.

page 18: adjust*ed* p-value

Reviewer #2 (Remarks to the Author): Expert in melanoma therapy, autophagy, functional assays, and in vivo models

The authors established an autophagy signature and a deep multi-omics analysis to define the autophagy status of a number of cancer types from TCGA. They also associated the autophagy status with the response to several anti-cancer drugs, and found that autophagy induction can sensitize some cancer cells to anti-cancer therapy, despite the common view that autophagy confers therapy resistance. They also experimentally validated this finding for a couple of anti-cancer drugs and in a cancer cell type.

Luo and co-authors had a very appealing essential idea: using the massive amount of RNA-seq data available for various cancer types to assess how autophagy is regulated based on the transcriptional levels of selected autophagic genes. Despite the attractive aim, the manuscript is mostly fragmented, multiple analyses have been shown (gene expression, GSEA and pathway analysis, miRNA and TFs analysis) without deeply elaborating them, and in some cases, wrong assumptions have been reported (e.g., PI3K-AKT cascade does promote the mTORC1 activation). Undoubtedly the autophagy role in regulating cancer development and progression and drug resistance is highly debated probably due to the heterogeneity of tumors and their multi-steps of progression. To this aim, it is essential to use appropriate lists of genes that can clearly describe the autophagy cascade. In most of the cases and, in particular, for autophagy and mTORC1 the GO gene lists contain genes that can have the opposite effects on autophagy regulation. The authors selected some autophagic genes based on the assumption that are present in all the datasets used, without specifying the role of these genes and if these are up- or down-regulated upon autophagy induction. Moreover, the authors did not assess if the identified TFs or miRNA may positively or negatively regulate these autophagic markers.

Then the authors attempted to experimentally validate their findings by characterising the response to specific drugs (etoposide and partly BMS592364) in relation to autophagy status in two melanoma cell lines in vitro and in xenografts.

Major concerns:

- The rationale behind their choice of skin melanoma as target cancer where to validate their findings should be better elucidated. Also, some reference in connection to the first sentence of the paragraph could be added.
- A more comprehensive analysis of the effects of the two selected drugs + autophagy inducer should be performed, including several melanoma cell lines and at least another autophagy inducer.

- Autophagy markers should be analysed in these cells (as well as in tumors from xenografts) to show that autophagy is effectively increasing upon these treatments.
- In the xenografts experiment, they should also try the other drug (namely BMS592364) in combination with the autophagy inducer, which seems to be more effective from Fig.7A, B. Or at least explain why they have selected only the etoposide.
- A more careful analysis of autophagy activation in these tumors should be performed. Rapamycin blocks the mTOR pathway and thus may interfere with different aspects of the tumor besides autophagy (cell growth, cell metabolism). Before claiming that autophagy induction is sensitizing the tumor to etoposide treatment, a more extensive analysis should be done. Moreover, the tumor growth kinetics in Fig. 7E (and tumor volume/weight in Fig. S5) shows that rapamycin treatment alone is more effective than etoposide alone. This aspect should be further elucidated.
- The section regarding the potential mechanism underlying the effect of rapamycin+etoposide treatment is very poor. It is unclear why they selected the DDIT4 as target gene among those downregulated upon combination treatment. Looking at the volcano plots (Fig. S6), several genes are downregulated and DDIT4 should be shown in the plot. The authors claimed that 84 genes are differentially expressed in etoposide vs etoposide+rapamycin in both cell lines and no figures have been reported to show that. Also, why did they look at the differences between etoposide vs etoposide+rapamycin only? This part should be carefully revised.
- They validated the downregulation of DDIT4 and its effects only in SK-Mel28. They should also check these in A375, the melanoma cells used for the in vivo studies. Also, other analyses should be included to check autophagy and DNA damage status upon silencing of DDIT4.
- The difference in terms of cell viability shown in Fig. 7I, J is very low. Moreover, control conditions are omitted (siCTR, CTR virus, untreated cells). A better explanation of these results (also in Fig. legend) is requested. And other assays should be included to prove this result.
- In general, figure legends should be more accurate and detailed. In suppl. Figure S5 there are many mistakes and Fig. S6 is even not indicated.
- The scheme in Fig. 7K has to be improved: from their results etoposide alone also affects DDIT4 (Fig. 7H).

Reviewer #3 (Remarks to the Author): Expert in autophagy in cancer and functional assays

In this manuscript (NCOMMS-21-25603A-Z), Luo M at colleagues report a study in which they have generated a gene signature to infer the autophagy status in tumors. According to the authors, this gene signature (n=37 genes) has been validated through a series of 10.000 tumors from the TCGA. In a next

step, they used this gene profile to classify tumors in autophagy high/low tumors to define the molecular alterations associated to autophagy alterations and potential therapeutic approaches to treat these tumors. To do this they focus on multi-omic analysis based on mRNA, miRNA and protein analysis plus DNA alterations (somatic mutations and copy-number changes). The authors finally report that rapamycin, a suspected autophagy inducer in their experimental model, sensitizes melanoma cancer cells to etoposide and BMS536924, an IGF1R inhibitor.

This is a non-incremental work that might hold great potential, especially considering that a proper computational tool to effectively assess autophagy status in a tumor is not available. The first impression is that the manuscript is technically sound, despite its biological relevance is demonstrated by scarce in vitro/in vivo experimentation. However, after initial excitement, this reader finds the work presented does not fully meet the expectations generated. The first observation is that, in opinion of this reviewer, the authors use in excess a panoply of computational tools to infer potential molecular mechanisms that sometimes are not properly explained or directly seem to be used for biased purposes. For instance, during the in silico analysis multiple selection criteria are not detailed, conclusions are drawn without being fully supported by the data collected, and are carelessly presented (the manuscript standard English is not optimal and presented in a way the audience loses interest). Although this can be easily addressed, other more integral concerns embody the main criticisms to this manuscript (explained below). It also presents some overstatements in the text and in some parts data appears inconsistent. Hence, while the decision on whether this study deserves to be published in Nature Communications is at the discretion of the editorial board, this referee does not consider this study to be worth published in this journal unless extensively edited and technically/experimentally improved.

In the following lines I present my comments that, hopefully, can help to improve the quality of the manuscript together with the insights from other referees:

Major points:

1) the main criticism to this paper is that the authors never experimentally assessed autophagy levels. One of the main reasons explaining the unavailability of reliable “predictors” of autophagy status is that autophagy is a dynamic biological process and, as such, cannot be fully addressed by analyzing static datasets but by monitoring flux experimentally. In opinion of this reviewer, the scoring of tumors into autophagy high/low is not supported and cannot be considered as such.

2) While a big part of the manuscript presented encompasses a commendable in silico computational work, the results that derive from it are merely correlative associations without experimental validations. The main conclusions that can be drawn from >3/4 of the manuscript are, at best, descriptive and lack any biological insight (e.g., miRNA and transcriptional factor analysis, etc). This further links to several inconsistencies detected throughout the manuscript (knowledge and bibliographic based) with regards to autophagy inhibition or activation and its interrelation with multiple signalling pathways. For instance, in the manuscript (page6) mTOR is both mentioned as an autophagy inducer or inhibitor depending on the interest of the authors. This becomes the norm throughout the

manuscript and other examples can be found when the authors describe the PI3K/AKT pathways as an inhibitor of mTOR (page6) or the RAS-MAPK pathway as an activator of autophagy. Additional unsubstantiated correlations without proper verification are also exposed when hypothesizing a potential crosstalk between EMT and autophagy activation (page7), a result for which we could expect an insightful discussion once highlighted in the text.

3) in multiple sections data seems counterintuitive. E.g., how come a deletion in ATG4B (2q37.3), a crucial gene to initiate autophagy, is associated by the authors to autophagy-high activity?

4) There are too many critical steps in the computational filtering that this reviewer could not find. Could the authors clarify what is a "consistent pattern of reflecting autophagy" (page 4) when they describe that four datasets present an autophagy validated status? how is it calculated/validated? can they provide a numerical value of this consistency?

5) Fig1D: where is the correlation based on the 37-gene set signature? this figure should be better described in the text.

6) page6: this reviewer does not find THYM and SKCM as tumor types exhibiting a large number of variations, as the authors suggest based on Fig2B. TGCT, STAD and BRCA seem to present, overall, higher variations if taking into account mRNA, protein, miRNA, mutations and SCNA data.

7) Fig2B: how many genes within the 37-gene set signature are altered in the five molecular types of alterations?

8) why do the authors analyse only 10 cancer-associated signalling pathways "to study the potential effects of autophagy on protein expression"? could the authors better explain the purpose of this analysis?

9) Fig7B-D. Inhibition of TORC1 by rapamycin is a well described procedure to increase autophagy. However, rapamycin has been demonstrated to induce a weak autophagic response in certain cell types (e.g., neurons, human neuroblastoma SH-SY5Y cells, etc). Moreover, TORC1 functions as an essential rheostat that integrates multiple signalling cues (e.g., insulin, EGF, aminoacid intracellular abundance, etc). Hence, it is likely that rapamycin ultimately also modulates other signal pathways other than autophagy. Therefore, how do the authors determine that the increased sensitization of A375 and SKMEL28 melanoma cells to etoposide and BMS536924 combined with rapamycin is derived from an actual increase in the autophagy levels? Did the authors assess the autophagy status after rapamycin exposure? Could the authors inhibit autophagy (e.g., ATG5 knockdown or knockout) to restore

chemoresistance? A complementary method such as nutrient deprivation should be used to strengthen this data and avoid rapamycin side-effects.

10) why did not the authors analyze the 37-gene signature status in their RNA sequenced/rapamycin-treated samples? Does their gene signature provide an autophagy-high validation?

11) Fig7F-G: data for all conditions should be reported.

12) Fig7H: qRT-PCR data for A375 cells should also be provided.

13) are the effects of rapamycin on DDIT4 (DNA damage inducible transcript 4) autophagy-related? Can this decrease be abrogated when autophagy is inhibited?

14) the discussion section does not contain any reference to the DNA damage response and autophagy, which is surprising considering the authors have focused on DDTI4 (a DNA damage-related protein) to explain the effects of rapamycin. In this regard, the authors should provide evidence that the decrease of DDIT4 by rapamycin is, indeed, causing DNA damage in SKMEL28 cells, as depicted in Fig7K.

Minor points:

1) the reference used to report ssGSEA may be wrong. It looks to this referee as if the reference included in the manuscript is used to report GSVA, not ssGSEA. To report the latter, Subramanian A PNAS. 2005;102(43):15545-15550 seems a better fit, according to <https://www.genepattern.org/modules/docs/ssGSEAProjection/4>.

2) page14: "significant features for the six molecular types" should be corrected for "five molecular types".

3) page7: genes are not sensitive to anticancer drugs. They can, on the other hand, be associated to sensitization/resistance.

4) page7: the PI3K/AKT pathway is a canonical inhibitor of autophagy (Klionsky DJ, et al Autophagy 2021). Since the authors suggest that this pathway may regulate autophagy in different directions, they should better clarify the foundation of this statement.

5) page7: the PI3K signalling pathway is a well-recognised inhibitor of chemosensitivity, not the opposite.

6) page11: figures S5C-F are, in fact, figures S6A-D

Re: NCOMMS-21-25603A-Z

REVIEWER COMMENTS

Reviewer #1 (Remarks to the Author): Expert in bioinformatics, drug response prediction, and multi-omics

In this paper, Luo et al. estimated the activity of autophagy in 33 cancer types from TCGA using a 37-gene signature constructed on the basis of six independent expression datasets with known autophagy status. They have shown that in many cancers ssGSEA-based autophagy scores vary widely and subsets of tumors with high and low autophagy scores are enriched with a number of molecular alterations at the level of the gene, protein, and miRNA expressions, methylation, and somatic mutations.

The main finding of this work is an unexpectedly high correlation between autophagy scores and predicted tumor sensitivities to some drugs. Since this contradicts previous knowledge on autophagy as a drug resistance mechanism, Luo et al. decided to go beyond in silico analysis and performed additional experiments in vitro and in vivo. They have shown that sensitivities of two cell lines to two drugs, etoposide and BMS536924, can be modified by (rapamycin-induced) autophagy and proposed a mechanistic explanation of this effect. They compared dose-response curves for cell lines treated with etoposide and BMS536924 alone and in combination with rapamycin, which is an inducer of autophagy. Analysis of RNA-seq profiles from these cell lines treated with etoposide and etoposide + rapamycin revealed 84 genes significantly differentially expressed between two groups. One of these 84 genes, DDIT4, was shown to be a mediator of rapamycin-induced autophagy effect on etoposide response. Its knockdown and overexpression decreased and increased cell viability, respectively. Finally, the authors created mice xenograft models and confirmed that the combination with rapamycin reinforces the effect of etoposide in vivo too.

The manuscript reports an interesting finding with a potential to advance anticancer treatment approaches and therefore is of interest to the community. I think that methods description, quality of figures, structure, and overall readability of the

manuscript could be improved.

Response: We thank the very nice summary and overall positive comments from the reviewer.

Structure of the manuscript

I would recommend extending the experimental validation part and making it more central in the paper. Figure 7 contains too many details.

Response: We thank the reviewer for this suggestion. We would like to take this opportunity to clarify that our manuscript is more computational-centralized. Recent studies demonstrated the power of rigorous analysis from large-scale data. For example, the recent TCGA/ICGC Pan-cancer project published a series of high-profile papers in Nature and Cell series journals¹⁻⁸. These comprehensive analyses provide insights for the research community for further investigations. Our group also significantly contributed to the cancer genomics field⁹⁻¹³.

In this manuscript, we first developed a method based on gene signature to characterize the autophagy status in cancer patients, which is still very challenging to evaluate nowadays. We then characterized the landscape of alterations at different molecular levels (e.g., mRNA expression, miRNA expression, protein expression, somatic mutations, and SCNVs), which is the most comprehensive landscape for autophagy-related alterations thus far. This comprehensive map will provide strong biological insights for future investigations. Furthermore, some key findings can only be identified through comprehensive analysis, including the conclusion summarized by the reviewer “**The main finding of this work is an unexpectedly high correlation between autophagy scores and predicted tumor sensitivities to some drugs.**” Please note, only with individual experiments (e.g., individual drug/cell line), it will be very challenging to make the conclusion that autophagy may drive the drug sensitivity for an appreciable number of anti-cancer drugs. We clarified this in the discussion section of the revised manuscript (page 14). We also hope the reviewer could appreciate the significance of the computational sections of this manuscript.

Nevertheless, we also agree with the reviewer to extend the experimental validation part. We performed significant number of experiments and re-organized the figure 7 as figure 7 (Autophagy sensitizing drug response *in vitro* and *in vivo*) and figure 8 (The potential mechanism of autophagy inducer sensitizing drug response), as well as supplementary figures S5-S7. In figure 7, panel B, D and G are newly added experimental results. In figure 8, panel B-F are newly added experimental results in A375 cell line, and the panel G-H are completely new. We also added other experiments in figure S5A-S5B, figure S6D and figure S7B-7F.

In detail, additional experimental results were listed below:

(1) We assessed the autophagy levels in both A375 and SK-MEL-28 (Figure 7B), and in xenograft model (Figure 7G), which confirmed that the rapamycin can induce autophagy.

Figure 7

Figure 7: Characterization of drug response associated with autophagy *in vitro* and *in vivo*.

(B) The autophagy status in A375 and SK-MEL-28 cell lines with DMSO, treatment of Rapamycin, Etoposide, and Rapamycin + Etoposide. (G) The autophagy status in mouse tissues with treatment of Vehicle, Rapamycin, Etoposide, and Rapamycin + Etoposide.

(2) With another autophagy-inducer (starvation-induced) and an additional cell line (SK-MEL-5), we validated the effects of autophagy-induction sensitizing melanoma cells to etoposide (Figure 7D, S5A-S5B).

The rapamycin-induced autophagy can sensitize additional cell line (SK-MEL-5) to etoposide (Figure S5A).

Figure S5

Figure S5: Characterization of autophagy sensitizing drug response *in vitro* and *in vivo*.

(A) Dose-response curves for the mean value of cell viability of etoposide and BMS536924 in rapamycin-induced and non-induced conditions in the melanoma cell line SK-MEL-5. Cell viability was normalized to level treated with DMSO. Error bars indicate the mean \pm SD. Sigmoidal dose-response curves were used to fit the data.

Meanwhile, starvation-induced autophagy can also sensitize melanoma cells (A375, SK-MEL-28, SK-MEL-5) to etoposide (Figure S5B, 7D).

Figure S5

Figure S5: Characterization of autophagy sensitizing drug response *in vitro*. (B) Dose-response curves for the mean value of cell viability of etoposide and BMS536924 in starvation-

induced and non-induced conditions in the melanoma cell line SK-MEL-5. Cell viability was normalized to level treated with DMSO. DMEM: Dulbecco's Modified Eagle Medium for control. EBSS: Earle's Balanced Salt Solution for starvation. Error bars indicate the mean \pm SD. Sigmoidal dose-response curves were used to fit the data.

Figure 7

Figure 7: Characterization of drug response associated with autophagy *in vitro* and *in vivo*.

(D) Dose-response curves for the mean value of cell viability of etoposide and BMS536924 in starvation-induced and non-induced conditions in the melanoma cell line A375 and SK-MEL-28. Cell viability was normalized to level treated with DMSO. DMEM: Dulbecco's Modified Eagle Medium for control. EBSS: Earle's Balanced Salt Solution for starvation. Error bars indicate the mean \pm SD. Sigmoidal dose-response curves were used to fit the data.

(3) We examined the expression level of function-related genes (EGF, FGF1, BINP3,

PPP2R2B, DDIT4 and SFN) by RT-PCR in A375 or SK-MEL-28 cells (Figure S6D).

Figure S6

Figure S6: Analysis of autophagy sensitizing drug response. (D) Relative expression of EGF, FGF1, BNIP3, PPP2R2B, SFN and DDIT4 in four groups of A375 and -SK-MEL-28 by RT-PCR.

(4) We evaluated the DDIT4 function and downregulation in both A375 and SK-MEL-28 by a series of additional experiments (Figure 8B-8F). Furthermore, we also inhibited autophagy by knocking down ATG5, and found that the sensitivity of melanoma to etoposide was decreased (Figure 8G). Meanwhile, the expression of DDIT4 was also increased (Figure 8H).

Figure 8

Figure 8: Potential mechanism through the functional characterization of DDIT4 *in vitro*.

(B) Relative mRNA expression of DDIT4 in A375 and SK-MEL-28 by RT-PCR. $n = 3$ (C) Western blot of DDIT4 in A375 or SK-MEL-28 cells treated with si-NC or DDIT4 siRNAs in combination with etoposide. (D) The cell viability of A375 or SK-MEL-28 cells treated with si-NC or DDIT4 siRNAs in combination with etoposide. Pink bars denote successful knockdown of si-DDIT4 (si-DDIT4-2 and siDDIT4-3), while grey bar denotes unsuccessful knockdown of si-DDIT4 (si-DDIT4-

1). (E) Western blot of DDIT4 in A375 or SK-MEL-28 cells transfected with DDIT4-OE lentivirus or vector in combination with etoposide and rapamycin. (F) The cell viability of A375 or SK-MEL-28 cells transfected with DDIT4-OE lentivirus or vector in combination with etoposide and rapamycin. (G) The cell viability of A375 or SK-MEL-28 cells treated with si-NC or ATG5 siRNA in combination with etoposide. (H) Western blot of DDIT4 in A375 or SK-MEL-28 cells treated with treated with si-NC or ATG5 siRNAs in combination with etoposide.

(5) The efficiency and specificity of DDIT4 siRNAs were examined in A375 and SK-MEL-28 (Figure S7B), and the autophagy status of cells when DDIT4 knockdown was also examined (Figure S7C). The efficiency of DDIT4-OE lentivirus was examined in A375 and SK-MEL-28 (Figure S7D), and the autophagy status was also examined (Figure S7E). The efficiency of ATG5 siRNAs was examined by western blot (Figure S7F).

Figure S7

Figure S7: Expression of DDIT4 and the efficiency and specificity of DDIT4 siRNAs or DDIT4-OE lentivirus and ATG5 siRNAs *in vitro*. (A) The expression difference of DDIT4 between the DMSO group and the rapamycin group based on public dataset, GSE27784. (B) The western blot of DDIT4 in A375 and SK-MEL-28 cells transfected with si-NC or DDIT4 siRNAs. (C) The autophagy status of A375 and SK-MEL-28 cells transfected with si-NC or DDIT4 siRNAs. (D) The western blot of DDIT4 in A375 and SK-MEL-28 cells transfected with vector or DDIT4-OE lentivirus. (E) The autophagy status of A375 and SK-MEL-28 cells transfected with vector or DDIT4-OE lentivirus. (F) The western blot of ATG5 in A375 and SK-MEL-28 cells transfected with si-NC or ATG5 siRNAs.

The sections describing the associations between omics data could be reduced and/or moved to supplementary. As far as I understood, the proposed mechanism of action is explained with the use of expression data only. If it is not the case, the role of other omics layers in this mechanism should be explained more clearly.

Response: We thank the reviewer for this suggestion. In this manuscript, we first defined the autophagy score to assess autophagy status across TCGA tumor samples. We then depicted the global pattern of multiple omics data in 24 cancer types, which will serve as a great resource for the research community for further investigations. We feel that the omics analysis is necessary, otherwise, we will not have the global view that the autophagy may drive the drug sensitive for an appreciable number of anti-cancer drugs. Please refer to our above response, and we prefer to keep the omics data analysis, while adding additional experimental validation.

I would add a section discussing the limitations of this work

Response: We thank the reviewer for this great suggestion. We further discussed the limitations of this work in the revised manuscript, *“Challenges remain in the study of autophagy that require further investigation. We estimated the relative autophagy status across tumor samples, which may be limited in its ability to reflect the real status of autophagy in patients. Autophagy is a dynamic biological process, which involves the regulation of various related molecules. Therefore, it is difficult to evaluate it accurately by the evaluation of only a small number of autophagy-related markers, such as LC3 or P62, etc. Although autophagy flux can monitor autophagy status, this approach is challenging even in cultured cell lines and model organisms⁸⁰, not to mention in large-scale (~10,000) patient samples. Similar to this, there is no direct approach to access the hypoxia status in patients, while we can only indirectly access the hypoxia status through a 15-gene signature^{81–84}, which can still lead to significant biological discoveries. Our analysis is based on bulk RNA-seq across different cell types within a sample. Further efforts should take tumor heterogeneity*

into consideration, i.e., with advancements in single-cell profiling technology. Finally, most clinical trials do not have information on the autophagy status of cancer samples due to the technical challenge of monitoring autophagy status in vivo. Currently main studies focused on the autophagy inhibition, while the importance of autophagy induction have been ignored, which lead to the limited understanding of how molecular signatures are affected by the autophagy microenvironment and reasonable interpretation of unexpectedly adverse drug outcome in anticancer therapy. Nonetheless, our study emphasizes the significance of monitoring tumor autophagy status in future clinical studies.” on page 14-15.

Methods

Unclear, which statistical tests were used to detect associations of autophagy scores with features and how FDR was computed.

Response: In this study, we classified tumor samples into the autophagy score-high group and autophagy score-low group using the autophagy scores of tumor samples in TCGA, and identified the associated features (mRNA, protein, miRNA, mutation, SCNv) by comparing the two classified groups. To minimize the disturbance of various confounding factors (tumor purity, sex, ethnicity, age at diagnosis, and smoking status), the Propensity Score Matching algorithm was used. The similar approach was used in our previous studies^{12,13}, as well as other studies^{14,15}. *“The Benjamini & Hochberg method was used to calculate the FDR (False Discovery Rate) in each cancer type”*. In the revised manuscript, we clarified this in the method section on page 16.

How many features were tested in total?

Response: In this study, we tested 5 different omics data including 20,288 mRNAs, 218 proteins, 2,435 miRNAs, 3,785 mutations, and 450 SCNvs in 24 cancer types. We clarified this in the revised manuscript (page 6). In the top row of revised figure 2B (please see below), we added the total number of all significant features in each cancer type.

Figure 2

B

Figure 2: Overview of five molecular features across cancer types. (B) Number of each altered molecular feature (mRNA, miRNA, mutation, protein and SCNA) and total altered molecular features in autophagy score-high (red) and autophagy score-low (blue) groups from TCGA tumor samples. Magenta points denote percentage of significant features over total features in each cancer.

Method descriptions in "Pathway enrichment and miRNA-target and TF-target regulatory network" are missing necessary details, tool versions, parameters. Were the tools run with default parameters? What kind of problem did each tool solve?

Response: We thank this valuable comment. Briefly, the clusterProfiler package was used for KEGG pathway enrichment analysis with default parameters. We obtained the human TF gene list from the AnimalTFDB 3.0 database¹⁶, the TF-Target pairs

from the hTFtarget database¹⁷, and the miRNA-Target pairs from the FFLtool database¹⁸. In the revised manuscript, we described this section in detail: *“KEGG pathway enrichment analysis was performed using the ‘clusterProfiler’ package⁹⁴ with default parameters. The gene list of human TFs was downloaded from AnimalTFDB 3.0⁹⁵, TF-target pairs downloaded from hTFtarget database⁹⁶, and miRNA-target pairs downloaded from the FFLtool⁹⁷. We matched the TFs from the altered mRNAs (as targets) and identified the TF-target regulatory pairs based on the TF-target pairs downloaded. The significantly altered miRNAs and mRNAs (as targets) were used to identify the miRNA-target relationships. Based on the TF-target pairs and miRNA-target pairs, we constructed TF-target regulatory networks and miRNA-target regulatory networks, where the nodes are the TFs or miRNAs and the genes. and the edges are the regulatory pairs.”* on page 16-17.

Why etoposide and BMS536924 were chosen for further experiments among all drugs demonstrating a high correlation of predicted response with autophagy score?

Response: In preliminary cellular experiments, we selected top six drugs (BMS708163, CMK, DMOG, BMS536924 and Etoposide) predicted to be sensitive on high autophagy status in SKCM (Figure 7A), excluding CGP.082996 due to the unavailability. Among these, etoposide and BMS536924 became significantly sensitive under rapamycin-induced autophagy conditions in both A375 and SK-MEL-28 cell lines (Figure 7C). Therefore, etoposide and BMS536924 were chosen for further experiments in this study. We clarified this in the revised manuscript (page 10-11).

How did the authors choose DDIT4 among 84 differentially expressed genes?

Response: In our manuscript, we used the cutoff of $|\text{Fold change}| > 1.5$ and adjusted p-value < 0.05 to identify differentially expressed genes, and the intersection number of differentially expressed genes in A375 and SK-MEL-28 was 290 (Table S6; Figure S6B). We apologize for misrepresenting the 84 differentially expressed genes here. We have clarified it in the revised manuscript (page 12).

Based on these overlapped differentially expressed genes (Figure S6C), we further performed KEGG pathway enrichment analysis, and noticed that several genes (EGF, FGF1, BINP3, DDIT4, PPP2R2B, SFN) was frequently observed in these pathways related to tumor or autophagy, such as PI3K-Akt signaling pathway and mTOR signaling pathway (Table S7).

Figure S6: Analysis of autophagy sensitizing drug response. (B) Volcano plot of differentially expressed genes for etoposide + rapamycin vs. etoposide in A375 and SK-MEL-28. (C) The venn plot of differentially expressed genes of etoposide + rapamycin vs. etoposide in A375 and -SK-MEL-28.

We then further examined these function-related genes by RT-PCR, and we found the relative expression of the DDIT4 was the lowest in the rapamycin + etoposide group compared to rapamycin or etoposide alone (Figure S6D), which is consistent with tumor volumes of four groups in xenograft model (Figure 7H). We therefore focused on DDIT4, and we clarified this in the revised manuscript (page 12).

Figure S6

Figure S6: Analysis of autophagy sensitizing drug response. (D) Relative expression of EGF, FGF1, BNIP3, PPP2R2B, SFN and DDIT4 in four groups of A375 and -SK-MEL-28 by RT-PCR.

Why the functional role of DDIT4 was examined in SK-MEL-28, but A375 was used to create xenografts?

Response: "Characterization of drug response associated with autophagy *in vitro*" showed that A375 and SK-MEL-28 have highly similar pattern (Figure 7B-7D). We therefore chose A375 to create xenograft model. Moreover, we added results for the functional examination of DDIT4 in A375 (Figure 8B-8F) as suggested by the reviewer. We clarified this in the revised manuscript (page 11).

Figures

In Fig. 1C:

"Student's t-test was used to assess the difference;" - isn't Mann-Whitney U test more appropriate for such small groups?

Response: A sample size of at least 16 is required for using the Mann-Whitney U

test¹⁹, and it is not suitable for cases with very limited samples. For example, we assess the difference between group A (1,2,3) and group B (11,12,13), and the p-value is 0.00026 according to Student's t-test (Figure R1A) and 0.1 on the Mann-Whitney U test (Figure R1B). Although the difference between the A group and the B group is obvious, the Mann-Whitney U test cannot test the difference. Moreover, Student's t-test was used to assess the difference between two small groups in previous studies^{20–22}.

Figure R1

Figure R1: RNA-seq analysis of autophagy sensitizing drug response. (A) The boxplot of group A (1,2,3) and group B (11,12,13), the difference was assessed by Student's t-test. (B) The boxplot of group A (1,2,3) and group B (11,12,13), the difference was assessed by Mann-Whitney U test.

Fig. 1D:

- unclear, what does the absence of rectangle mean, e.g. for HAMDB and ACC?

Response: We displayed the significant Spearman correlation ($FDR < 0.05$) in the figure 1D, and the absence of the rectangle means non-significance ($FDR \geq 0.05$). We clarified this in the revised figure legend.

- presenting R as a number and $-\log(\text{adj. p-value})$ as color or size would greatly simplify the figure.

Response: We displayed the original figure 1D and re-plotted the figure according to the reviewer's suggestions (Figure R2). We feel that the original figure might be

better, and we prefer to keep the original figure 1D. We will be honored to revise accordingly if the reviewer has any remaining concerns.

Figure 1

Figure R2

Figure R2: (A) Correlations between autophagy scores based on the 37-gene set and gene sets from MSigDB, HADB, HAMDB, and ncRDeathDB among TCGA tumor samples. Rectangle size indicates FDR for Spearman correlation; FDR < 0.05. The number displayed was Spearman correlation coefficient (Rs). (B) Correlations between autophagy scores based on the 37-gene set and gene sets from MSigDB, HADB, HAMDB, and ncRDeathDB among TCGA tumor samples. Color intensity indicates FDR for Spearman correlation; FDR < 0.05. The number displayed was Spearman correlation coefficient (Rs).

Fig. 2B: Interesting, that for miRNA, mutations, and sCNA the number of features associated with low autophagy score was much higher than the number of features associated with high autophagy.

Response: We agreed with the reviewer that it is interesting that for miRNA, mutations, and SCNA the number of features associated with low autophagy scores was much higher than the number of features associated with high autophagy. This also demonstrated that the large-scale analysis may reveal some interesting patterns as well as novel biological insight. Future investigations are necessary to further understand the pattern and underlying mechanism.

Fig. 3A: legend label should be $-\log(\text{adj. p-value})$, not just p-value.

Response: We revised the legend label in Figure. 3A.

Fig. 3B: FDR is not on a log-scale.

Response: We revised the legend label in Figure. 3B.

Figures 4A and B: y-axis label for barplots is missing

Response: We added the y-axis label of barplots in Figures 4A-4B.

Figure 5B: figure legend contains unnecessary repetitions

Response: We revised the figure legend in Figure 5B.

Figure S4: the number of drugs

Response: We revised the legend label in Figure S4.

Other comments and corrections

Please correct the database name, it should be ncRDeathDB instead of ncDeathDB.

Response: We corrected the name as ncRDeathDB in the revised manuscript.

"We validated the 37-gene list in six independent datasets." - I would not call this validation because the same datasets were used to build this gene list. I would reformulate it e.g. "we confirmed that the 37-gene list is consistent with autophagy status in six datasets from GEO."

Response: According to reviewer's suggestions, we have revised the sentence to "*We confirmed that the autophagy scores based on the 37-gene list are consistent with autophagy status in six datasets from GEO*" (page 15).

page 6: "For example, several cancer types, ...showed a larger number of variations between the autophagy score-high and autophagy score-low groups (Figure 2B), suggesting a greater impact of autophagy in these cancer types". I do not think that drawing the conclusion about autophagy impact just from the number of associated alterations is correct. What if these cancer types are more heterogeneous than the others?

Response: We thank this valuable comment and agreed with reviewer's comments. We removed "*suggesting a greater impact of autophagy in these cancer types*" in the revised manuscript.

Please define in the main text "genes enriched in cancer types" (2nd sentence of Associations between autophagy and mRNA / protein expression and signaling pathways, page 6)? I could not find any explanation in main text, it appears only in Fig. 3 caption.

Response: We thank this valuable comment. In the revised manuscript, we clarified it, "*These altered genes were enriched in 16 pathways from the Kyoto Encyclopedia of Genes and Genomes (KEGG), which were enriched with more than 10 genes in at least two cancer types (p -value < 0.05; Figure 3A, Table S2).*" on page 6.

page 7:

- Why did the authors use drug responses from GDSC with omics data from CCLE?

GDSC provides several versions of multi-omics profiles for tested cell lines.

- The second paragraph does not agree with the content of "Analysis of drug response in autophagy status" on page 15. According to Methods, expression profiles of cell lines were obtained from GDSC, not CCLE. The authors should clearly write which source of expression data they used.

- CCLE and GDSC datasets are not referenced

Response: We are sorry for the confusion, and we only used the data from GDSC, which has 252 drugs. We have corrected the source and cited it in the second paragraph on page 7: *"we first calculated the correlations between mRNA expression of multiple cancer-related pathways in 1,074 cancer cell lines and the area under the dose-response curve (AUC) of 252 anti-cancer drugs in Genomics of Drug Sensitivity in Cancer (GDSC)⁴⁴".*

page 7: The authors write about "genes ... sensitive to anti-cancer drugs (Figure 3C)".

How do they define genes sensitive or resistant to drugs?

Response: We thank this comment. In the section "Analysis of drug response in autophagy status" of Methods, we have defined it, *"As in our previous study⁸¹, the positive Spearman correlation was defined as drug resistant, while the negative Spearman correlation was defined as drug sensitive."* on page 17.

page 8: "These genes targeted by miRNAs were enriched in multiple autophagy-related pathways, including the PI3K-AKT and MAPK signaling pathways (Figure 4A, FDR < 0.05; Figure S3C, Table S3)." - the details on the statistical test used are missing. Shouldn't it be p-value reported for this single test instead of FDR?

Response: In this study, all enrichment analysis were performed by 'ClusterProfiler'²³ package with default parameters. The enrichment analysis of these genes targeted by miRNAs is not a single test. In the revised manuscript, we have made it clear in the section "Pathway enrichment and miRNA-target and TF-target regulatory network" of Methods on page 16-17, *"KEGG pathway enrichment analysis was performed using the 'clusterProfiler' package⁹⁴ with default parameters. The*

gene list of human TFs was downloaded from AnimalTFDB 3.0⁹⁵, TF-target pairs downloaded from hTFtarget database⁹⁶, and miRNA-target pairs downloaded from the FFLtool⁹⁷. We matched the TFs from the altered mRNAs (as targets) and identified the TF-target regulatory pairs based on the TF-target pairs downloaded. The significantly altered miRNAs and mRNAs (as targets) were used to identify the miRNA-target relationships. Based on the TF-target pairs and miRNA-target pairs, we constructed TF-target regulatory networks and miRNA-target regulatory networks, where the nodes are the TFs or miRNAs and the genes. and the edges are the regulatory pairs.”

page 12: “sensitizing etoposide to autophagy inducer”

Response: We corrected it as “*sensitizing etoposide by autophagy inducer*”.

page 13: "32 CAGs to be significantly altered across 20 cancer types, which were resistant or sensitive in multiple cancer types" on page 13.

Response: In the revised manuscript, we revised it as “*32 CAGs were significantly altered across 20 cancer types, and drugs targeting these CAGs were resistant or sensitive to autophagy in multiple cancer types*”.

The last paragraph in “The potential mechanism of autophagy inducer sensitizing drug response” section is hard to understand.

Response: In the revised manuscript, we carefully revised the entire section “The potential mechanism of autophagy inducer sensitizing drug response” on page 11-13.

page 14: "from 32 TCGA tumor samples" - the authors probably mean 32 tumor types.

Response: We have corrected it in the revised manuscript.

page 18: adjust*ed* p-value

Response: We have revised the “adjust p-value” to “adjusted p-value”.

Reviewer #2 (Remarks to the Author): Expert in melanoma therapy, autophagy, functional assays, and in vivo models

The authors established an autophagy signature and a deep multi-omics analysis to define the autophagy status of a number of cancer types from TCGA. They also associated the autophagy status with the response to several anti-cancer drugs, and found that autophagy induction can sensitize some cancer cells to anti-cancer therapy, despite the common view that autophagy confers therapy resistance. They also experimentally validated this finding for a couple of anti-cancer drugs and in a cancer cell type.

Luo and co-authors had a very appealing essential idea: using the massive amount of RNA-seq data available for various cancer types to assess how autophagy is regulated based on the transcriptional levels of selected autophagic genes. Despite the attractive aim, the manuscript is mostly fragmented, multiple analyses have been shown (gene expression, GSEA and pathway analysis, miRNA and TFs analysis) without deeply elaborating them, and in some cases, wrong assumptions have been reported (e.g., PI3K-AKT cascade does promote the mTORC1 activation).

Response: We thank the reviewer's valuable comments. We would like to take this opportunity to clarify that our manuscript is more computational-centralized. Recent studies demonstrated the power of rigorous analysis from large-scale data. For example, the recent TCGA/ICGC Pan-cancer project published a series of high-profile papers in Nature and Cell series journals¹⁻⁸. These comprehensive analyses provide insights for the research community for further investigations. Our group also contributed significantly to the cancer genomics field^{9-12,24}.

In this manuscript, we first developed a method based on gene signature to characterize the autophagy status in cancer patients, which is still very challenging to evaluate nowadays. We then characterized the landscape of alterations at different molecular levels (e.g., mRNA expression, miRNA expression, protein expression, somatic mutations, and SCNVs), which is the most comprehensive landscape for autophagy-related alterations thus far. This comprehensive map will provide strong biological insight for future investigations. Furthermore, some key findings can only

be identified through comprehensive analysis. Please note, with individual experiments (e.g., individual drug/cell line), it will be very challenging to make the conclusion that autophagy may drive the drug sensitivity for an appreciable number of anti-cancer drugs. We clarified this in the revised manuscript (page 14). We also hope the reviewer could appreciate the significance of the computational sections of this manuscript.

We carefully checked our statements to avoid some controversial assumptions. For example, we removed *“For example, we found that RPS6KA6, a member of the RPS6K kinase family targeted by autophagy inducer mTOR⁴⁰, was highly expressed in autophagy score-high groups across multiple cancer types (Figure S2A). The PI3K-AKT signaling pathway inactivates the mTORC1 pathway, inhibition of which will activate the autophagy process¹. LAMA3 in PI3K-AKT, a member of the LAMP family that includes LAMP-1 and LAMP-2 as regulators in the fusion of autophagic vesicles and lysosomes, has higher expression in the high-ATG group of TGCT (Figure S2A)³³.”* on page 7;

We removed *“The PI3K-AKT pathway score is significantly higher in the autophagy score-high group of three cancers (GBM, head and neck squamous cell carcinoma, and lung adenocarcinoma [LUAD]), while it is significantly lower in the autophagy score-low group of breast invasive carcinoma (BRCA) and THCA. This is consistent with previous reports that PI3K-AKT may regulate autophagy in different directions in different cancer types^{1,40}, suggesting complex associations between the PI3KAKT and autophagic machinery.”* on page 7;

We also removed *“The autophagy-associated gene ATG4B, which is crucial for processing LC3B in autophagosome formation, is located in the 2q37.3 deletion of BRCA and LGG (Figure 5B). A previous study showed that ATG4B knockdown can improve trastuzumab sensitivity to reduce cell viability in trastuzumab-resistant HER2+ breast cancer cells⁵⁴”* on page 10.

Undoubtedly the autophagy role in regulating cancer development and progression and drug resistance is highly debated probably due to the heterogeneity of tumors

and their multi-steps of progression. To this aim, it is essential to use appropriate lists of genes that can clearly describe the autophagy cascade. In most of the cases and, in particular, for autophagy and mTORC1 the GO gene lists contain genes that can have the opposite effects on autophagy regulation. The authors selected some autophagic genes based on the assumption that are present in all the datasets used, without specifying the role of these genes and if these are up- or down-regulated upon autophagy induction.

Response: The autophagy cascade was regulated by a series of genes, and thus dysregulation of some genes could be compensated by other genes. We did not define autophagy status based on up- and down-regulation of any individual genes. Instead, based on a series of autophagy-related genes, we defined an autophagy score by the ssGSEA algorithm, which can estimate the overall autophagy status. The autophagy score is based on a 37-gene signature (37 genes as a whole) to estimate autophagy status regardless of the individual up- and down-regulation of the expression of these genes. The ssGSEA algorithm establishes a baseline and estimates the distance between baseline and each gene in signatures. This algorithm was widely used to estimate status of samples in the previous studies^{13,25–29}.

Moreover, the authors did not assess if the identified TFs or miRNA may positively or negatively regulate these autophagic markers.

Response: The regulatory network constructed by the identified TFs or miRNA and targeted genes, and the regulatory pairs were downloaded from hTFtarget¹⁷ (a comprehensive database for regulations of human transcription factors and their targets) and FFLtools¹⁸ (a database with miRNA and their targets). These regulatory pairs were predicted by bioinformatical tools based on biological regulation and sequence pairing. Here, we provide an overview of TF or miRNA regulatory networks in 24 cancer types, which will provide important molecular regulatory resources for studying the effects of autophagy in cancer in further investigations.

Then the authors attempted to experimentally validate their findings by characterising

the response to specific drugs (etoposide and partly BMS592364) in relation to autophagy status in two melanoma cell lines in vitro and in xenografts.

Response: We thank the reviewer's comments.

Major concerns:

- The rationale behind their choice of skin melanoma as target cancer where to validate their findings should be better elucidated. Also, some reference in connection to the first sentence of the paragraph could be added.

Response: We thank this valuable comment. One reason is that melanoma is one of the top cancer types with the largest number of variations affected by autophagy status (Figure 2B). Furthermore, our experimental team has been committing to the melanoma research^{30,31}, and has a more comprehensive understanding of melanoma than other cancer types. We have various melanoma cell lines, which will provide sufficient conditions for subsequent experiments, including drug response and animal model construction. In the revised manuscript, we have stated, "*Melanoma is one of the top cancer types (THYM, TGCT, SKCM) with the largest number of variations affected by autophagy status (Figure 2B), and it is more common than the other two cancer types. We thus examined the drug response sensitized by autophagy in melanoma.*".

- A more comprehensive analysis of the effects of the two selected drugs + autophagy inducer should be performed, including several melanoma cell lines and at least another autophagy inducer.

Response: Again, we hope the reviewer could appreciate the significance of the computational sections of this manuscript and our manuscript is more computational-centralized as we responded above.

Meanwhile, we also agreed with the reviewer's great suggestions to use another autophagy inducer to strengthen our experimental sections. In the revised manuscript, we added an additional autophagy inducer (starvation-induction) and an additional cell line (SK-MEL-5) to validate our results (S5A-S5B, Figure 7D).

In detail, the rapamycin-induced autophagy can sensitize an additional cell line (SK-MEL-5) to etoposide and BMS536924 (Figure S5A).

Figure S5

Figure S5: Characterization of autophagy sensitizing drug response *in vitro* and *in vivo*..

(A) Dose-response curves for the mean value of cell viability of etoposide and BMS536924 in rapamycin-induced and non-induced conditions in the melanoma cell line SK-MEL-5. Cell viability was normalized to level treated with DMSO. Error bars indicate the mean \pm SD. Sigmoidal dose-response curves were used to fit the data.

Meanwhile, starvation-induced autophagy can sensitize melanoma cells (A375, SK-MEL-28, SK-MEL-5) to etoposide and BMS536924 (Figure S5B, 7D).

Figure S5

Figure S5: Characterization of autophagy sensitizing drug response *in vitro* and *in vivo*..

(B) Dose-response curves for the mean value of cell viability of etoposide and BMS536924 in

starvation-induced and non-induced conditions in the melanoma cell line SK-MEL-5. DMEM: Dulbecco's Modified Eagle Medium for control. EBSS: Earle's Balanced Salt Solution for starvation. Cell viability was normalized to level treated with DMSO. Error bars indicate the mean \pm SD. Sigmoidal dose-response curves were used to fit the data.

Figure 7

Figure 7: Characterization of drug response associated with autophagy *in vitro* and *in vivo*.

(D) Dose-response curves for the mean value of cell viability of etoposide and BMS536924 in starvation-induced and non-induced conditions in the melanoma cell line A375 and SK-MEL-28. Cell viability was normalized to level treated with DMSO. DMEM: Dulbecco's Modified Eagle Medium for control. EBSS: Earle's Balanced Salt Solution for starvation. Error bars indicate the mean \pm SD. Sigmoidal dose-response curves were used to fit the data.

Taken together, we showed two autophagy inducers (rapamycin and starvation

induction) can make three melanoma cells (A375, SK-MEL-28, SK-MEL-5) sensitive to two drugs (etoposide and BMS536924), respectively.

- Autophagy markers should be analysed in these cells (as well as in tumors from xenografts) to show that autophagy is effectively increasing upon these treatments.

Response: We thank this valuable comment. We have checked the autophagy markers (LC3A/B, p62) in A375, SK-MEL-28 (Figure 7B) and tumors from xenograft model (Figure 7F), which showed that autophagy was increased by rapamycin.

Figure 7

Figure 7: Characterization of drug response associated with autophagy *in vitro* and *in vivo*.

(B) The autophagy status in A375 and SK-MEL-28 cell lines with DMSO, treatment of Rapamycin, Etoposide, and Rapamycin + Etoposide. (G) The autophagy status in mouse tissues with treatment of Vehicle, Rapamycin, Etoposide, and Rapamycin + Etoposide.

- In the xenografts experiment, they should also try the other drug (namely BMS592364) in combination with the autophagy inducer, which seems to be more effective from Fig.7A, B. Or at least explain why they have selected only the etoposide.

Response: We thank this valuable comment. Etoposide is an anti-cancer drug used in the clinic, so the xenografts for etoposide may enlarge the clinical impact. We clarified this reason in the revised manuscript.

- A more careful analysis of autophagy activation in these tumors should be performed. Rapamycin blocks the mTOR pathway and thus may interfere with different aspects of the tumor besides autophagy (cell growth, cell metabolism). Before claiming that autophagy induction is sensitizing the tumor to etoposide treatment, a more extensive analysis should be done. Moreover, the tumor growth kinetics in Fig. 7E (and tumor volume/weight in Fig. S5) shows that rapamycin treatment alone is more effective than etoposide alone. This aspect should be further elucidated.

Response: We thank this valuable comment. We did notice that autophagy inducer rapamycin does have anti-tumor effects in melanoma³². This may also explain the result that tumor growth kinetics in Fig. 7E (revised Figure 7H). To further confirm the effect of autophagy induction on the sensitivity of cancer cells to etoposide, we added another autophagy inducer starvation (instead of rapamycin) to induce autophagy, which showed that etoposide can be still sensitized in A375 and SK-MEL-28 (Figure 7D) and additional SK-MEL-5 (Figure S5B) as mentioned above, suggesting that autophagy activation is likely to cause this sensitivity.

Furthermore, we also used three siATG5, and successfully inhibited autophagy by knocking down ATG5 through siATG5-1 (Figure S7F). We found that the sensitivity of melanoma to etoposide was also decreased in A375 and SK-MEL-28 (Figure 8G).

Figure S7

Figure S7: Expression of DDIT4 and the efficiency and specificity of DDIT4 siRNAs or DDIT4-OE lentivirus and ATG5 siRNAs *in vitro*. (F) The western blot of ATG5 in A375 and SK-MEL-28 cells transfected with si-NC or ATG5 siRNAs.

Figure 8

Figure 8: Potential mechanism through the functional characterization of DDIT4 *in vitro*. (G) The cell viability of A375 or SK-MEL-28 cells treated with si-NC or ATG5 siRNA in combination with etoposide.

- The section regarding the potential mechanism underlying the effect of rapamycin+etoposide treatment is very poor. It is unclear why they selected the DDIT4 as target gene among those downregulated upon combination treatment. Looking at the volcano plots (Fig. S6), several genes are downregulated and DDIT4 should be shown in the plot. The authors claimed that 84 genes are differentially expressed in etoposide vs etoposide+rapamycin in both cell lines and no figures have been reported to show that. Also, why did they look at the differences between etoposide vs etoposide+rapamycin only? This part should be carefully revised.

Response: We thank this valuable comment, and we revised this section very carefully. In our manuscript, we used $|\text{fold change}| > 1.5$ and adjusted $p\text{-value} < 0.05$ to identify differentially expressed genes (Figure S6B), and the intersection number

of differentially expressed genes in A375 and SK-MEL-28 was 290 (Figure S6C). We apologize for misrepresenting the 84 differentially expressed genes here. We have clarified it in the revised manuscript (page 12). We also labeled the DDIT4 and several other genes in the volcano plots (Figure S6B) and provided the list of the differentially expressed genes in Table S6 in the revised manuscript.

Figure S6: Analysis of autophagy sensitizing drug response. (B) Volcano plot of differentially expressed genes for etoposide + rapamycin vs. etoposide in A375 and SK-MEL-28. (C) The venn plot of differentially expressed genes of etoposide + rapamycin vs. etoposide in A375 and -SK-MEL-28.

Based on these overlapped differentially expressed genes, we further performed KEGG pathway enrichment analysis, and noticed that several genes (EGF, FGF1, BINP3, PPP2R2B, DDIT4, SFN) was frequently observed in these pathways associated to tumor or autophagy, such as PI3K-AKT signaling pathway and HIF-1 signaling pathway (Table S7). We then further examined these function-related genes by RT-PCR, and we found the expression of the DDIT4 was the lowest in the rapamycin + etoposide group compared to rapamycin or etoposide alone (Figure S6D), which is consistent with our observation in tumor volumes of four groups in xenograft model (Figure 7F & 7H). We therefore focused on DDIT4, and we clarified this in the revised manuscript.

Figure S6

Figure S6: Analysis of autophagy sensitizing drug response. (D) Relative expression of EGF, FGF1, BINP3, PPP2R2B, SFN and DDIT4 in four groups of A375 and -SK-MEL-28 by RT-PCR.

About “why did they look at the differences between etoposide vs etoposide+rapamycin only?”, in this manuscript, we aim to understand the effect of autophagy activation (induced by rapamycin) on the etoposide, so we only compared the difference in the transcriptome of melanoma cell lines with the treatment of etoposide+rapamycin (autophagy-inducer) and etoposide+DMSO. Moreover, rapamycin is widely used to induce autophagy, and there are many public data released (e.g., GSE27784). Therefore, we can use these public data to further validate our results. For example, a re-analysis of GSE27784 showed that rapamycin can significantly decrease the expression of DDIT4 (Figure S7A). In the revised manuscript, we clarified it on page 12, “RNA-seq showed that DDIT4 was highly expressed and significantly downregulated in A375 (fold change = -1.60, adjusted p-value = 1.32×10^{-13}) and SK-MEL-28 (fold change = -2.34, adjusted p-value = 2.60×10^{-39}) when treated with etoposide and rapamycin compared to etoposide alone

(Figure 8A), which was confirmed in A375 and SK-MEL-28 with real-time PCR when treated with etoposide + rapamycin (Figure 8B). Moreover, we also observed that the expression of DDIT4 could be decreased by rapamycin in the public data (Figure S7A).”.

Figure S7

Figure S7: Expression of DDIT4 and the efficiency and specificity of DDIT4 siRNAs or DDIT4-OE lentivirus and ATG5 siRNAs in vitro. (A) The expression difference of DDIT4 between the DMSO group and the rapamycin group based on public dataset, GSE27784.

- They validated the downregulation of DDT4 and its effects only in SK-Mel28. They should also check these in A375, the melanoma cells used for the in vivo studies. Also, other analyses should be included to check autophagy and DNA damage status upon silencing of DDT4.

Response: We have checked the downregulation of DDIT4 and its effects in A375, which showed the consistent results with SK-MEL-28 (Figure 8B-8H).

Figure 8

Figure 8: Potential mechanism through the functional characterization of DDIT4 *in vitro*.

(B) Relative mRNA expression of DDIT4 in A375 and SK-MEL-28 by RT-PCR. n = 3 (C) Western blot of DDIT4 in A375 or SK-MEL-28 cells treated with si-NC or DDIT4 siRNAs in combination with etoposide. (D) The cell viability of A375 or SK-MEL-28 cells treated with si-NC or DDIT4 siRNAs in combination with etoposide. Pink bars denote successful knockdown of si-DDIT4 (si-DDIT4-2 and siDDIT4-3), while grey bar denotes unsuccessful knockdown of si-DDIT4 (si-DDIT4-

1). (E) Western blot of DDIT4 in A375 or SK-MEL-28 cells transfected with DDIT4-OE lentivirus or vector in combination with etoposide and rapamycin. (F) The cell viability of A375 or SK-MEL-28 cells transfected with DDIT4-OE lentivirus or vector in combination with etoposide and rapamycin. (G) The cell viability of A375 or SK-MEL-28 cells treated with si-NC or ATG5 siRNA in combination with etoposide. (H) Western blot of DDIT4 in A375 or SK-MEL-28 cells treated with treated with si-NC or ATG5 siRNAs in combination with etoposide.

We also checked the autophagy status upon silencing of DDIT4 in A375, SK-MEL-28 (Figure 7SC).

Figure S7

Figure S7: Expression of DDIT4 and the efficiency and specificity of DDIT4 siRNAs or DDIT4-OE lentivirus and ATG5 siRNAs *in vitro*. (C) The autophagy status of A375 and SK-MEL-28 cells transfected with si-NC or DDIT4 siRNAs.

Finally, we agree with the reviewer that our evidence about DNA damage status is not strong enough. We only propose the potential mechanisms in our original manuscript. We removed related sections in the revised manuscript. Please note, removing the DNA damage part will not affect the major conclusion of our study.

- The difference in terms of cell viability shown in Fig. 7I, J is very low. Moreover, control conditions are omitted (siCTR, CTR virus, untreated cells). A better explanation of these results (also in Fig. legend) is requested. And other assays should be included to prove this result.

Response: The differences of cell viability were tested to be statistically significant (p -value < 0.05). The previous studies also reported the significant difference, although the differences in cell viability look low³³⁻³⁶. In the revised manuscript, we have labeled the control condition of siDDIT4 (si-NC) and control condition of OE-DDIT4 (Vector) (Figure 8C-8F).

Figure 8

Figure 8: Potential mechanism through the functional characterization of DDIT4 *in vitro*.

(C) Western blot of DDIT4 in A375 or SK-MEL-28 cells treated with si-NC or DDIT4 siRNAs in combination with etoposide. (D) The cell viability of A375 or SK-MEL-28 cells treated with si-NC or DDIT4 siRNAs in combination with etoposide. Pink bars denote successful knockdown of si-

DDIT4 (si-DDIT4-2 and siDDIT4-3), while grey bar denotes unsuccessful knockdown of si-DDIT4 (si-DDIT4-1). (E) Western blot of DDIT4 in A375 or SK-MEL-28 cells transfected with DDIT4-OE lentivirus or vector in combination with etoposide and rapamycin. (F) The cell viability of A375 or SK-MEL-28 cells transfected with DDIT4-OE lentivirus or vector in combination with etoposide and rapamycin.

- In general, figure legends should be more accurate and detailed. In suppl. Figure S5 there are many mistakes and Fig. S6 is even not indicated.

Response: We thank this valuable comment. We have revised the figure legend to be more accurate and detailed.

- The scheme in Fig. 7K has to be improved: from their results etoposide alone also affects DDT4 (Fig. 7H).

Response: In the illustration, we aim to emphasize that autophagy may drive drug sensitive, potentially through impact on DDIT4, while the effects of etoposide on DDIT4 is not our focus. We revised the figure accordingly to avoid confusion as the figure below. We will be honored to further revise if the reviewer has any remaining concerns.

Figure 8

Reviewer #3 (Remarks to the Author): Expert in autophagy in cancer and functional assays

In this manuscript (NCOMMS-21-25603A-Z), Luo M at colleagues report a study in which they have generated a gene signature to infer the autophagy status in tumors. According to the authors, this gene signature (n=37 genes) has been validated through a series of 10.000 tumors from the TCGA. In a next step, they used this gene profile to classify tumors in autophagy high/low tumors to define the molecular alterations associated to autophagy alterations and potential therapeutic approaches to treat these tumors. To do this they focus on multi-omic analysis based on mRNA, miRNA and protein analysis plus DNA alterations (somatic mutations and copy-number changes). The authors finally report that rapamycin, a suspected autophagy inducer in their experimental model, sensitizes melanoma cancer cells to etoposide and BMS536924, an IGF1R inhibitor.

This is a non-incremental work that might hold great potential, especially considering that a proper computational tool to effectively assess autophagy status in a tumor is not available. The first impression is that the manuscript is technically sound, despite its biological relevance is demonstrated by scarce in vitro/in vivo experimentation. However, after initial excitement, this reader finds the work presented does not fully meet the expectations generated. The first observation is that, in opinion of this reviewer, the authors use in excess a panoply of computational tools to infer potential molecular mechanisms that sometimes are not properly explained or directly seem to be used for biased purposes. For instance, during the in silico analysis multiple selection criteria are not detailed, conclusions are drawn without being fully supported by the data collected, and are carelessly presented (the manuscript standard English is not optimal and presented in a way the audience loses interest). Although this can be easily addressed, other more integral concerns embody the main criticisms to this manuscript (explained below). It also presents some overstatements in the text and in some parts data appears inconsistent. Hence, while the decision on whether this study deserves to be published in Nature Communications is at the discretion of the editorial board, this referee does not consider this study to be worth published in this

journal unless extensively edited and technically/experimentally improved.

In the following lines I present my comments that, hopefully, can help to improve the quality of the manuscript together with the insights from other referees:

Response: We appreciate those valuable comments, especially with the comment “This is a non-incremental work that might hold great potential, especially considering that a proper computational tool to effectively assess autophagy status in a tumor is not available.” We would like to take this opportunity to clarify that our manuscript is more computational-centralized. Recent studies demonstrated the power of rigorous analysis from large-scale data. For example, the recent TCGA/ICGC Pan-cancer project published a series of high-profile papers in Nature and Cell sister journals¹⁻⁸. These comprehensive analyses provide insights for the research community for further investigations. Our group also contributed significantly to the cancer genomics field^{9-12,24}.

In this manuscript, we first developed a method based on gene signature to characterize the autophagy status in cancer patients, which is still very challenging to evaluate nowadays. We then characterized the landscape of alterations at different molecular levels (e.g., mRNA expression, miRNA expression, protein expression, somatic mutations, and SCNVs), which is the most comprehensive landscape for autophagy-related alterations thus far. This comprehensive map will provide strong biological insight for future investigations. Furthermore, some key findings can only be identified through comprehensive analysis. For example, with individual experiments (e.g., individual drug/cell line), it will be very challenging to make the conclusion that autophagy may drive the drug sensitivity for an appreciable number of anti-cancer drugs. We further clarified this in the revised manuscript (page 14). We also hope the reviewer could appreciate the significance of the computational sections of this manuscript.

For reviewer’s comment “its biological relevance is demonstrated by scarce in vitro/in vivo experimentation”, we have added significant number of experiments to demonstrate biological relevance, and these experiments included:

(1) We assessed the autophagy levels of A375 and SK-MEL-28 (Figure 7B), and

xenograft model (Figure 7G), which confirmed that the rapamycin can induce autophagy.

Figure 7

Figure 7: Characterization of drug response associated with autophagy *in vitro* and *in vivo*.

(B) The autophagy status in A375 and SK-MEL-28 cell lines with DMSO, treatment of Rapamycin, Etoposide, and Rapamycin + Etoposide. (G) The autophagy status in mouse tissues with treatment of Vehicle, Rapamycin, Etoposide, and Rapamycin + Etoposide.

(2) With another autophagy-inducer (starvation-induced) and an additional cell line (SK-MEL-5), we validated the effects of autophagy-induction sensitizing melanoma cells to etoposide (Figure 7D, S5A-S5B).

In detail, the rapamycin-induced autophagy can sensitize additional cell line SK-MEL-5 cells to etoposide (Figure S5A).

Figure S5

Figure S5: Characterization of autophagy sensitizing drug response *in vitro* and *in vivo*.

(A) Dose-response curves for the mean value of cell viability of etoposide and BMS536924 in rapamycin-induced and non-induced conditions in the melanoma cell line SK-MEL-5. Cell viability was normalized to level treated with DMSO. Error bars indicate the mean \pm SD. Sigmoidal dose-response curves were used to fit the data.

Meanwhile, starvation-induced autophagy can sensitize melanoma cells (A375, SK-MEL-28, SK-MEL-5) to etoposide (Figure 7D, S5B).

Figure S5

Figure S5: Characterization of autophagy sensitizing drug response *in vitro* and *in vivo*.

(B) Dose-response curves for the mean value of cell viability of etoposide and BMS536924 in starvation-induced and non-induced conditions in the melanoma cell line SK-MEL-5. DMEM:

Dulbecco's Modified Eagle Medium for control. EBSS: Earle's Balanced Salt Solution for starvation. Cell viability was normalized to level treated with DMSO. Error bars indicate the mean \pm SD. Sigmoidal dose-response curves were used to fit the data.

Figure 7

Figure 7: Characterization of drug response associated with autophagy *in vitro* and *in vivo*.

(D) Dose-response curves for the mean value of cell viability of etoposide and BMS536924 in starvation-induced and non-induced conditions in the melanoma cell line A375 and SK-MEL-28. Cell viability was normalized to level treated with DMSO. DMEM: Dulbecco's Modified Eagle Medium for control. EBSS: Earle's Balanced Salt Solution for starvation. Error bars indicate the mean \pm SD. Sigmoidal dose-response curves were used to fit the data.

(3) We examined function-related genes (EGF, FGF1, BINP3, PPP2R2B, DDIT4 and SFN) by RT-PCR in A375 or SK-MEL-28 cells (Figure S6D).

Figure S6

Figure S6: Analysis of autophagy sensitizing drug response. (D) Relative expression of EGF, FGF1, BNIP3, PPP2R2B, SFN and DDIT4 in four groups of A375 and -SK-MEL-28 by RT-PCR.

(4) We evaluated the DDIT4 function and downregulation consistently in A375 and SK-MEL-28 by a series of additional experiments (Figure 8B-8F). Furthermore, we also inhibited autophagy by knocking down ATG5, and found that the sensitivity of melanoma to etoposide was decreased (Figure 8G). Meanwhile, the expression of DDIT4 was also increased (Figure 8H).

Figure 8

Figure 8: Potential mechanism through the functional characterization of DDIT4 *in vitro*.

(B) Relative mRNA expression of DDIT4 in A375 and SK-MEL-28 by RT-PCR. n = 3 (C) Western blot of DDIT4 in A375 or SK-MEL-28 cells treated with si-NC or DDIT4 siRNAs in combination with etoposide. (D) The cell viability of A375 or SK-MEL-28 cells treated with si-NC or DDIT4 siRNAs in combination with etoposide. Pink bars denote successful knockdown of si-DDIT4 (si-DDIT4-2 and siDDIT4-3), while grey bar denotes unsuccessful knockdown of si-DDIT4 (si-DDIT4-

1). (E) Western blot of DDIT4 in A375 or SK-MEL-28 cells transfected with DDIT4-OE lentivirus or vector in combination with etoposide and rapamycin. (F) The cell viability of A375 or SK-MEL-28 cells transfected with DDIT4-OE lentivirus or vector in combination with etoposide and rapamycin. (G) The cell viability of A375 or SK-MEL-28 cells treated with si-NC or ATG5 siRNA in combination with etoposide. (H) Western blot of DDIT4 in A375 or SK-MEL-28 cells treated with treated with si-NC or ATG5 siRNAs in combination with etoposide.

(5) The efficiency and specificity of DDIT4 siRNAs were examined in A375 and SK-MEL-28 (Figure S7B), and the autophagy status of cells when DDIT4 knockdown was also examined (Figure S7C). The efficiency of DDIT4-OE lentivirus was also examined in A375 and SK-MEL-28 (Figure S7D), and the autophagy status was also examined (Figure S7E). The efficiency of ATG5 siRNAs was examined by western blot (Figure S7F).

Figure S7

Figure S7: Expression of DDIT4 and the efficiency and specificity of DDIT4 siRNAs or DDIT4-OE lentivirus and ATG5 siRNAs *in vitro*. (A) The expression difference of DDIT4 between the DMSO group and the rapamycin group based on public dataset, GSE27784. (B) The western blot of DDIT4 in A375 and SK-MEL-28 cells transfected with si-NC or DDIT4 siRNAs. (C) The autophagy status of A375 and SK-MEL-28 cells transfected with si-NC or DDIT4 siRNAs. (D) The western blot of DDIT4 in A375 and SK-MEL-28 cells transfected with vector or DDIT4-OE lentivirus. (E) The autophagy status of A375 and SK-MEL-28 cells transfected with vector or DDIT4-OE lentivirus. (F) The western blot of ATG5 in A375 and SK-MEL-28 cells transfected with si-NC or ATG5 siRNAs.

Furthermore, we extensively revised the manuscript:

(1) We added extensive results of experiments in the results section of “Autophagy sensitizing drug response *in vitro* and *in vivo*” and “The potential mechanism of autophagy inducer sensitizing drug response”.

(2) We clarified the details of methods and selection criteria, such as “KEGG pathway enrichment analysis was performed using the ‘clusterProfiler’ package⁹⁴ with default parameters. The gene list of human TFs was downloaded from AnimalTFDB 3.0⁹⁵, TF-target pairs downloaded from hTFtarget database⁹⁶, and miRNA-target pairs downloaded from the FFLtool⁹⁷. We matched the TFs from the altered mRNAs (as targets) and identified the TF-target regulatory pairs based on the TF-target pairs downloaded. The significantly altered miRNAs and mRNAs (as targets) were used to identify the miRNA-target relationships. Based on the TF-target pairs and miRNA-target pairs, we constructed TF-target regulatory networks and miRNA-target regulatory networks, where the nodes are the TFs or miRNAs and the genes. and the edges are the regulatory pairs.”.

(3) We adjusted the statement accordingly to avoid overstatement. For example, we removed the section related to DNA damage, “Previous studies reported the potential mechanism of etoposide to inhibit tumor is related to DNA replication and damage⁶⁵”.

(4) We carefully deleted some inconsistent sentences, “For example, we found that RPS6KA6, a member of the RPS6K kinase family targeted by autophagy inducer mTOR⁴⁰, was highly expressed in autophagy score-high groups across multiple cancer types (Figure S2A). The PI3K-AKT signaling pathway inactivates the mTORC1 pathway, inhibition of which will activate the autophagy process¹. LAMA3 in PI3K-AKT, a member of the LAMP family that includes LAMP-1 and LAMP-2 as regulators in the fusion of autophagic vesicles and lysosomes, has higher expression in the high-ATG group of TGCT (Figure S2A)³³.” on page 7; “The PI3K-AKT pathway score is significantly higher in the autophagy score-high group of three cancers (GBM, head and neck squamous cell carcinoma, and lung adenocarcinoma [LUAD]), while it is significantly lower in the autophagy score-low group of breast invasive carcinoma

(BRCA) and THCA. This is consistent with previous reports that PI3K-AKT may regulate autophagy in different directions in different cancer types^{1,40}, suggesting complex associations between the PI3KAKT and autophagic machinery.” on page 7; “The autophagy-associated gene ATG4B, which is crucial for processing LC3B in autophagosome formation, is located in the 2q37.3 deletion of BRCA and LGG (Figure 5B). A previous study showed that ATG4B knockdown can improve trastuzumab sensitivity to reduce cell viability in trastuzumab-resistant HER2+ breast cancer cells⁵⁴” on page 10.

(5) We carefully edited our language.

We hope the reviewer will satisfy our revision, and we will be honored to further revise if the reviewer has any remaining concerns.

Major points:

1) the main criticism to this paper is that the authors never experimentally assessed autophagy levels. One of the main reasons explaining the unavailability of reliable “predictors” of autophagy status is that autophagy is a dynamic biological process and, as such, cannot be fully addressed by analyzing static datasets but by monitoring flux experimentally. In opinion of this reviewer, the scoring of tumors into autophagy high/low is not supported and cannot be considered as such.

Response: We strongly agree that “autophagy is a dynamic biological process and, as such, cannot be fully addressed by analyzing static datasets”. Autophagy is a dynamic biological process, which involves the regulation of various related molecules. Therefore, it is difficult to evaluate it accurately by the evaluation of only a small number of autophagy-related markers, such as LC3 or P62, etc. Although autophagy flux can monitor autophagy status, this approach is challenging even in cultured cell lines and model organisms³⁷, not to mention in large-scale (~10,000) patient samples. Similar to this, there is no direct approach to access the hypoxia status in patients, while we can only indirectly access the hypoxia status through a 15-gene signature^{13,25,38,39}, which can still lead to significant biological discoveries. In this study, we confirmed the robustness of the autophagy status based on a 37-gene

signature in six independent datasets from GEO (Figure 1C).

To further address the reviewer's concern, we added additional experimental assessments of autophagy levels whenever we can, including in the cell lines (A375, SK-MEL-28; Figure 7B) and tumors from xenograft model (Figure 7G) as mentioned above.

We also discussed the limitations of our approach in the discussion section of the revised manuscript, *“Challenges remain in the study of autophagy that require further investigation. We estimated the relative autophagy status across tumor samples, which may be limited in its ability to reflect the real status of autophagy in patients. Autophagy is a dynamic biological process, which involves the regulation of various related molecules. Therefore, it is difficult to evaluate it accurately by the evaluation of only a small number of autophagy-related markers, such as LC3 or P62, etc. Although autophagy flux can monitor autophagy status, this approach is challenging even in cultured cell lines and model organisms⁸⁰, not to mention in large-scale (~10,000) patient samples. Similar to this, there is no direct approach to access the hypoxia status in patients, while we can only indirectly access the hypoxia status through a 15-gene signature^{81–84}, which can still lead to significant biological discoveries. Our analysis is based on bulk RNA-seq across different cell types within a sample. Further efforts should take tumor heterogeneity into consideration, i.e., with advancements in single-cell profiling technology. Finally, most clinical trials do not have information on the autophagy status of cancer samples due to the technical challenge of monitoring autophagy status in vivo. Currently main studies focused on the autophagy inhibition, while the importance of autophagy induction have been ignored, which lead to the limited understanding of how molecular signatures are affected by the autophagy microenvironment and reasonable interpretation of unexpectedly adverse drug outcome in anticancer therapy. Nonetheless, our study emphasizes the significance of monitoring tumor autophagy status in future clinical studies”* page 14-15. Please note, it will be extremely challenged to better evaluate the autophagy status in large-scale patient samples.

2) While a big part of the manuscript presented encompasses a commendable in silico computational work, the results that derive from it are merely correlative associations without experimental validations. The main conclusions that can be drawn from >3/4 of the manuscript are, at best, descriptive and lack any biological insight (e.g., miRNA and transcriptional factor analysis, etc

Response: We thank to the reviewers for their appreciation of our computational work “a big part of the manuscript presented encompasses a commendable in silico computational work” and agree with the reviewer that the in silico computational work are merely correlative associations. However, recent studies demonstrated the power of rigorous analysis from large-scale data. For example, the recent TCGA/ICGC Pan-cancer project published a series of high-profile papers in Nature and Cell series journals¹⁻⁸. These comprehensive analyses provide insights for the research community for further investigations.

For the majority of the manuscript based on computational analysis, we first developed a method based on gene signature to characterize the autophagy status in cancer patients, which is still very challenging to evaluate nowadays. We then characterized the landscape of alterations at different molecular levels (e.g., mRNA expression, miRNA expression, protein expression, somatic mutations, and SCNVs), which is the most comprehensive landscape for autophagy-related alterations thus far. This comprehensive map will provide strong biological insight for future investigations. We hope the reviewer could appreciate the significance of the computational sections of this manuscript. Please also refer to our response above.

Nevertheless, we also added extensive experiments mentioned above focusing on the key points presented as shown above in the revised manuscript (Figure 7-8, Figure S5-S7).

This further links to several inconsistencies detected throughout the manuscript (knowledge and bibliographic based) with regards to autophagy inhibition or activation and its interrelation with multiple signaling pathways. For instance, in the manuscript (page6) mTOR is both mentioned as an autophagy inducer or inhibitor

depending on the interest of the authors. This becomes the norm throughout the manuscript and other examples can be found when the authors describe the PI3K/AKT pathways as an inhibitor of mTOR (page6) or the RAS-MAPK pathway as an activator of autophagy. Additional unsubstantiated correlations without proper verification are also exposed when hypothesizing a potential crosstalk between EMT and autophagy activation (page7), a result for which we could expect an insightful discussion once highlighted in the text.

Response: We thank the reviewer for this valuable comment. We then carefully went throughout the manuscript and revised them accordingly. For example, we deleted *“For example, we found that RPS6KA6, a member of the RPS6K kinase family targeted by autophagy inducer mTOR⁴⁰, was highly expressed in autophagy score-high groups across multiple cancer types (Figure S2A). The PI3K-AKT signaling pathway inactivates the mTORC1 pathway, inhibition of which will activate the autophagy process¹. LAMA3 in PI3K-AKT, a member of the LAMP family that includes LAMP-1 and LAMP-2 as regulators in the fusion of autophagic vesicles and lysosomes, has higher expression in the high-ATG group of TGCT (Figure S2A)³³.”* on page 7; *“The PI3K-AKT pathway score is significantly higher in the autophagy score-high group of three cancers (GBM, head and neck squamous cell carcinoma, and lung adenocarcinoma [LUAD]), while it is significantly lower in the autophagy score-low group of breast invasive carcinoma (BRCA) and THCA. This is consistent with previous reports that PI3K-AKT may regulate autophagy in different directions in different cancer types^{1,40}, suggesting complex associations between the PI3K/AKT and autophagic machinery.”* on page 7; *“The autophagy-associated gene ATG4B, which is crucial for processing LC3B in autophagosome formation, is located in the 2q37.3 deletion of BRCA and LGG (Figure 5B). A previous study showed that ATG4B knockdown can improve trastuzumab sensitivity to reduce cell viability in trastuzumab-resistant HER2+ breast cancer cells⁵⁴”* on page 10 to avoid controversy assumptions in the revised manuscript. We also cited a relevant reference for *“a potential crosstalk between EMT and autophagy activation”*, which is reported in a

previous study⁴⁰ (page7). We will be honored to further revise accordingly if the reviewer has any remaining concerns.

3) in multiple sections data seems counterintuitive. E.g., how come a deletion in ATG4B (2q37.3), a crucial gene to initiate autophagy, is associated by the authors to autophagy-high activity?

Response: We thank the reviewer for this valuable comment. The ATG4B belongs to the ATG4 family, which included several other genes, including ATG4A, ATG4C, and ATG4D. These ATG4 genes have similar functions (autophagy initiation) and/or shared targets^{41,42}, such as 34 common targets of ATG4A and ATG4B, and 11 common targets of ATG4B and ATG4D⁴¹. Therefore, although the ATG4B was deleted, other ATG4 genes may compensate for the loss of function. To avoid confusion, we deleted the related sentence, "*The autophagy-associated gene ATG4B, which is crucial for processing LC3B in autophagosome formation, is located in the 2q37.3 deletion of BRCA and LGG (Figure 5B). A previous study showed that ATG4B knockdown can improve trastuzumab sensitivity to reduce cell viability in trastuzumab-resistant HER2+ breast cancer cells*" on page 10.

4) There are too many critical steps in the computational filtering that this reviewer could not find. Could the authors clarify what is a "consistent pattern of reflecting autophagy" (page 4) when they describe that four datasets present an autophagy validated status? how is it calculated/validated? can they provide a numerical value of this consistency?

Response: To further clarify, we first collected the autophagy-related genes from six databases (MSigDB, HADB, HAMDB, and ncRDeathDB, THANATOS, Autophagy), and also collected six independent datasets with known autophagy status (GSE107600, GSE117189, GSE129204, GSE106175, GSE90444, and GSE31397). We then used the ssGSEA algorithm to calculate the autophagy scores based on autophagy-related genes in each database throughout all six independent datasets (Figure 1A). As you can see, the ssGSEA scores based on autophagy-related genes

from 4 databases (MSigDB, HADB, HAMDB, and ncRDeathDB) can accurately distinguish the cell lines in autophagy-high status vs. autophagy-low status in at least four independent datasets (marked in red in figure 1A). We then identified the overlapped autophagy-related genes across these four databases and defined the overlapped genes (n = 37) as a 37-gene signature for the following analysis. The ssGSEA scores based on the 37-gene signature can accurately distinguish the autophagy-high status vs. autophagy-low status in all six independent datasets from GEO (Figure 1C). We clarified this in the revised manuscript (page 4-5).

5) Fig1D: where is the correlation based on the 37-gene set signature? this figure should be better described in the text.

Response: The correlation indicated the Spearman correlations between autophagy scores based on 37-gene set and autophagy scores based on gene sets from MSigDB, HADB, HAMDB, or ncRDeathDB, respectively. In the revised Fig. 1D legend, we have revised it, "*Correlation between autophagy scores based on 37-gene set and gene sets from MSigDB, HADB, HAMDB, or ncRDeathDB among TCGA tumor samples. Color intensity indicates Spearman correlation coefficient (Rs); rectangle size indicates FDR for Spearman correlation; FDR < 0.05. The absence of the rectangle means non-significance (FDR > 0.05).*".

6) page6: this reviewer does not find THYM and SKCM as tumor types exhibiting a large number of variations, as the authors suggest based on Fig2B. TGCT, STAD and BRCA seem to present, overall, higher variations if taking into account mRNA, protein, miRNA, mutations and SCNA data.

Response: Please note, that the numbers on the Y-axis are quite different for different molecular types. To avoid confusion, we added a total number of associated features in the revised figure 2B (the top row), and THYM, TGCT, and SKCM exhibited the largest number of variations.

Figure 2

B

Figure 2: Overview of five molecular features across cancer types. (B) Number of each altered molecular feature (mRNA, miRNA, mutation, protein and SCNA) and total altered molecular feature in autophagy score-high (red) and autophagy score-low (blue) groups from TCGA tumor samples. Magenta point denotes percentage of significant features over total features in each cancer.

7) Fig2B: how many genes within the 37-gene set signature are altered in the five molecular types of alterations?

Response: We thank this valuable comment. We have checked five molecular types and found that there are 13 altered genes of 37-gene set signature across 16 cancer types. Please note, we evaluate the autophagy score in the whole gene signature,

instead of individual genes. Individual genes may have slightly difference, while combined as the signature may have bigger impact.

8) why do the authors analyse only 10 cancer-associated signalling pathways “to study the potential effects of autophagy on protein expression”? could the authors better explain the purpose of this analysis?

Response: This data is based on Reverse Phase Protein Array (RPPA) from TCGA, which focused on 10 cancer signaling pathways^{11,43–45}. We clarified this in the method section of the revised manuscript (page 6). The purpose of this analysis is to better understand the effects of autophagy on cancer signaling pathways by identifying alterations of cancer signaling pathways between the autophagy score-high group and the autophagy score-low group.

9) Fig7B-D. Inhibition of TORC1 by rapamycin is a well described procedure to increase autophagy. However, rapamycin has been demonstrated to induce a weak autophagic response in certain cell types (e.g., neurons, human neuroblastoma SH-SY5Y cells, etc). Moreover, TORC1 functions as an essential rheostat that integrates multiple signalling cues (e.g., insulin, EGF, aminoacid intracellular abundance, etc). Hence, it is likely that rapamycin ultimately also modulates other signal pathways other than autophagy. Therefore, how do the authors determine that the increased sensitization of A375 and SKMEL28 melanoma cells to etoposide and BMS536924 combined with rapamycin is derived from an actual increase in the autophagy levels? Did the authors assess the autophagy status after rapamycin exposure? Could the authors inhibit autophagy (e.g., ATG5 knockdown or knockout) to restore chemoresistance? A complementary method such as nutrient deprivation should be used to strengthen this data and avoid rapamycin side-effects.

Response: We thank the reviewer for these great suggestions. First of all, we assessed the autophagy status of SK-MEL-28 and A375 cells (Figure 7B) and tumors from xenograft model (Figure 7G) treated with etoposide after rapamycin exposure as mentioned above, and observed the autophagy activation.

We agree with the reviewer that for the rapamycin related experiments, there will be other possibilities. We appreciate the reviewer's suggestion "A complementary method such as nutrient deprivation should be used to strengthen this data and avoid rapamycin side-effects". We added another autophagy inducer starvation (instead of rapamycin) to induce autophagy, which showed that etoposide can still be sensitized as mentioned above (Figure 7D, S5B), suggesting that autophagy activation is likely to cause this effect.

Furthermore, we also performed the autophagy inhibition experiment. When the ATG5 was knocked down (autophagy inhibition) by siATG5-1 (Figure S7F), sensitivity of the etoposide was decreased in A375 and SK-MEL-28 cell lines (Figure 8G) as mentioned above.

We added these new data in the revised manuscript (page 11-13).

10) why did not the authors analyze the 37-gene signature status in their RNA sequenced/rapamycin-treated samples? Does their gene signature provide an autophagy-high validation?

Response: Our experimental results have shown that the autophagy status of the SK-MEL-28 and A375 cells or mouse tissue could be increasingly induced by rapamycin (Figure 7B, 7G) as mentioned above. The number of sequenced samples was too small, and may not be appropriately to test the status by the 37-gene signature.

11) Fig7F-G: data for all conditions should be reported.

Response: We thank this valuable comment. Fig7F-G showed the mRNA expression of DDIT4 by transcriptome data (rapamycin + etoposide vs etoposide in SK-MEL and A375 cell lines), which have been deposited in GSE185814. In this manuscript, we aim to understand the effect of autophagy activation (induced by rapamycin) on the etoposide, so we only compared the difference in the transcriptome of melanoma cell lines with the treatment of etoposide + rapamycin (autophagy-inducer) and etoposide + DMSO. Moreover, rapamycin is widely used to induce autophagy, and there are

many public data released (e.g., GSE27784). A re-analysis of GSE27784 showed that rapamycin can significantly decrease the expression of DDIT4 (Figure S7A). In the revised manuscript, we clarified it on page 12, “*RNA-seq showed that DDIT4 was highly expressed and significantly downregulated in A375 (fold change = -1.60, adjusted p-value = 1.32×10^{-13}) and SK-MEL-28 (fold change = -2.34, adjusted p-value = 2.60×10^{-39}) when treated with etoposide and rapamycin compared to etoposide alone (Figure 8A), which was confirmed in A375 and SK-MEL-28 with real-time PCR when treated with etoposide + rapamycin (Figure 8B). Moreover, we also observed that the expression of DDIT4 could be decreased by rapamycin in the public data (Figure S7A)*”.

Figure S7

Figure S7: Expression of DDIT4 and the efficiency and specificity of DDIT4 siRNAs or DDIT4-OE lentivirus and ATG5 siRNAs *in vitro*. (A) The expression difference of DDIT4 between the DMSO group and the rapamycin group based on public dataset, GSE27784.

12) Fig7H: qRT-PCR data for A375 cells should also be provided.

Response: In the revised Fig. 8H, we have added the qRT-PCR of DDIT4 in A375 cells (Figure 8B).

Figure 8

Figure 8: Potential mechanism through the functional characterization of DDIT4 *in vitro*.

(B) Relative mRNA expression of DDIT4 in A375 and SK-MEL-28 by RT-PCR.

13) are the effects of rapamycin on DDIT4 (DNA damage inducible transcript 4) autophagy-related? Can this decrease be abrogated when autophagy is inhibited?

Response: When we inhibit autophagy by knockdown ATG5, the expression of DDIT4 was upregulated (Figure 8H), suggesting the effects of rapamycin on DDIT4 is autophagy related.

Figure 8

H

Figure 8: Potential mechanism through the functional characterization of DDIT4 *in vitro*. (H) Western blot of DDIT4 in A375 or SK-MEL-28 cells treated with treated with si-NC or ATG5 siRNAs in combination with etoposide.

14) the discussion section does not contain any reference to the DNA damage response and autophagy, which is surprising considering the authors have focused on DDTI4 (a DNA damage-related protein) to explain the effects of rapamycin. In this regard, the authors should provide evidence that the decrease of DDIT4 by rapamycin is, indeed, causing DNA damage in SKMEL28 cells, as depicted in Fig7K.

Response: We thank this valuable comment and agreed with the reviewer's suggestions. Through the experiments, we validated the downregulation of DDIT4 and its effects on melanoma cells, which indicated that the DDIT4 gene indeed played a critical role in the mechanism of autophagy-induction sensitizing melanoma cells to etoposide. However, we do not have strong evidence that this is related to DNA damage response, which is far beyond the scope of our current manuscript. We

therefore removed DNA damage response section, and this will not impact the major conclusion of our manuscript.

Minor points:

1) the reference used to report ssGSEA may be wrong. It looks to this referee as if the reference included in the manuscript is used to report GSVA, not ssGSEA. To report the latter, Subramanian A PNAS. 2005;102(43):15545-15550 seems a better fit, according to <https://www.genepattern.org/modules/docs/ssGSEAProjection/4>.

Response: We thank this valuable comment. We revised the reference, *“In this study, we utilize single sample gene set enrichment analysis (ssGSEA)²¹, a widely used method^{22–25}, to estimate autophagy status in a large number of cancer samples, followed by comprehensive analysis to understand molecular alterations in autophagy.”* on page 4.

2) page14: “significant features for the six molecular types” should be corrected for “five molecular types”.

Response: We corrected it in the revised manuscript (page 16).

3) page7: genes are not sensitive to anticancer drugs. They can, on the other hand, be associated to sensitization/resistance.

Response: We thank and agreed with the reviewer’s comment. We corrected the related sentences on page 7, *“the majority of genes (91.1%; n = 180) are associated to the sensitization of anti-cancer drugs (Figure 3C). These pathways include some autophagy-associated pathways, such as the PI3K-AKT signaling pathway, MAPK signaling pathway, and mTOR signaling pathway (Figure 3C). For example, the PI3K signaling pathway, which was reported as the inhibitor of the chemosensitivity⁴⁵, was significantly varied in both mRNA and protein profiling. We found that 79 genes in the PI3K signaling pathway are associated to the sensitization of 11 anti-cancer drugs ($R_s < -0.3$, $FDR < 0.05$; Figure S2A), and 34 genes in the PI3K signaling pathway are associated to the resistance of 9 anti-cancer drugs ($R_s > 0.3$, $FDR < 0.05$; Figure*

S2A). *Regulating the MAPK signaling pathway enhanced the drug sensitivity of cancer cells in multiple cancer types*^{46,47}. We found that 36 genes in the MAPK signaling pathway are associated to the sensitization of 11 anti-cancer drugs ($R_s < -0.3$, $FDR < 0.05$; Figure S2B) and 4 genes in the MAPK signaling pathway are associated to the resistance of 9 anti-cancer drugs ($R_s > 0.3$, $FDR < 0.05$; Figure S2B). Previous studies also indicated that regulating the mTOR signaling pathway can sensitize cancer cells to anti-cancer drugs^{48,49}. We found that 26 genes in the mTOR signaling pathway are associated to the sensitization of 2 anti-cancer drugs ($R_s < -0.3$, $FDR < 0.05$; Figure S2C), and 10 genes in the mTOR signaling pathway are associated to the resistance of 9 anti-cancer drugs ($R_s > 0.3$, $FDR < 0.05$; Figure S2C)”.

4) page7: the PI3K/AKT pathway is a canonical inhibitor of autophagy (Klionsky DJ, et al Autophagy 2021). Since the authors suggest that this pathway may regulate autophagy in different directions, they should better clarify the foundation of this statement.

Response: We thank this valuable comment. In the revised manuscript, we deleted “For example, we found that *RPS6KA6*, a member of the *RPS6K* kinase family targeted by autophagy inducer *mTOR*⁴⁰, was highly expressed in autophagy score-high groups across multiple cancer types (Figure S2A). The PI3K-AKT signaling pathway inactivates the *mTORC1* pathway, inhibition of which will activate the autophagy process¹. *LAMA3* in PI3K-AKT, a member of the *LAMP* family that includes *LAMP-1* and *LAMP-2* as regulators in the fusion of autophagic vesicles and lysosomes, has higher expression in the high-ATG group of TGCT (Figure S2A)³³.” on page 7; “The PI3K-AKT pathway score is significantly higher in the autophagy score-high group of three cancers (GBM, head and neck squamous cell carcinoma, and lung adenocarcinoma [LUAD]), while it is significantly lower in the autophagy score-low group of breast invasive carcinoma (BRCA) and THCA. This is consistent with previous reports that PI3K-AKT may regulate autophagy in different directions in different cancer types^{1,40}, suggesting complex associations between the PI3KAKT

and autophagic machinery.” on page 7 to avoid confusion.

5) page7: the PI3K signalling pathway is a well-recognised inhibitor of chemosensitivity, not the opposite.

Response: In the revised manuscript, we corrected relevant statement as “*the PI3K signaling pathway, which was reported as the inhibitor of the chemosensitivity⁴⁰*”.

6) page11: figures S5C-F are, in fact, figures S6A-D

Response: We thank this valuable comment. We corrected these figures.

Reference:

1. Alexandrov, L. B. *et al.* The repertoire of mutational signatures in human cancer. *Nature* **578**, 94–101 (2020).
2. Calabrese, C. *et al.* Genomic basis for RNA alterations in cancer. *Nature* **578**, 129–136 (2020).
3. Yuan, Y. *et al.* Comprehensive molecular characterization of mitochondrial genomes in human cancers. *Nat Genet* **52**, 342–352 (2020).
4. Rodriguez-Martin, B. *et al.* Pan-cancer analysis of whole genomes identifies driver rearrangements promoted by LINE-1 retrotransposition. *Nat Genet* **52**, 306–319 (2020).
5. Hoadley, K. A. *et al.* Cell-of-Origin Patterns Dominate the Molecular Classification of 10,000 Tumors from 33 Types of Cancer. *Cell* **173**, 291-304.e6 (2018).
6. Chen, H. *et al.* A Pan-Cancer Analysis of Enhancer Expression in Nearly 9000 Patient Samples. *Cell* **173**, 386-399.e12 (2018).
7. Chiu, H.-S. *et al.* Pan-Cancer Analysis of lncRNA Regulation Supports Their Targeting of Cancer Genes in Each Tumor Context. *Cell Reports* **23**, 297-312.e12 (2018).
8. Berger, A. C. *et al.* A Comprehensive Pan-Cancer Molecular Study of Gynecologic and Breast Cancers. *Cancer Cell* **33**, 690-705.e9 (2018).
9. Gong, J. *et al.* A Pan-cancer Analysis of the Expression and Clinical Relevance of Small Nucleolar RNAs in Human Cancer. *Cell Rep* **21**, 1968–1981 (2017).
10. Ye, Y. *et al.* The Genomic Landscape and Pharmacogenomic Interactions of Clock Genes in Cancer Chronotherapy. *Cell Syst* **6**, 314-328.e2 (2018).
11. Zhang, Z. *et al.* Transcriptional landscape and clinical utility of enhancer RNAs for eRNA-targeted therapy in cancer. *Nat Commun* **10**, 4562 (2019).
12. Ye, Y. *et al.* Sex-associated molecular differences for cancer immunotherapy. *Nat Commun* **11**, 1779 (2020).
13. Ye, Y. *et al.* Characterization of Hypoxia-associated Molecular Features to Aid Hypoxia-Targeted Therapy. *Nat Metab* **1**, 431–444 (2019).

14. Yuan, Y. *et al.* Comprehensive Characterization of Molecular Differences in Cancer between Male and Female Patients. *Cancer Cell* **29**, 711–722 (2016).
15. Wang, Y.-H., Wintzell, V., Ludvigsson, J. F., Svanström, H. & Pasternak, B. Association Between Proton Pump Inhibitor Use and Risk of Fracture in Children. *JAMA Pediatr* **174**, 543–551 (2020).
16. Hu, H. *et al.* AnimalTFDB 3.0: a comprehensive resource for annotation and prediction of animal transcription factors. *Nucleic Acids Res.* **47**, D33–D38 (2019).
17. Zhang, Q. *et al.* hTFtarget: A Comprehensive Database for Regulations of Human Transcription Factors and Their Targets. *Genomics, Proteomics & Bioinformatics* **18**, 120–128 (2020).
18. Xie, G.-Y. *et al.* FFLtool: a web server for transcription factor and miRNA feed forward loop analysis in human. *Bioinformatics* **36**, 2605–2607 (2020).
19. Dwivedi, A. K., Mallawaarachchi, I. & Alvarado, L. A. Analysis of small sample size studies using nonparametric bootstrap test with pooled resampling method. *Stat Med* **36**, 2187–2205 (2017).
20. Cho, J. H., Collins, J. J. & Wong, W. W. Universal Chimeric Antigen Receptors for Multiplexed and Logical Control of T Cell Responses. *Cell* **173**, 1426–1438.e11 (2018).
21. Jacob, F. *et al.* A Patient-Derived Glioblastoma Organoid Model and Biobank Recapitulates Inter- and Intra-tumoral Heterogeneity. *Cell* **180**, 188–204.e22 (2020).
22. Amor, C. *et al.* Senolytic CAR T cells reverse senescence-associated pathologies. *Nature* **583**, 127–132 (2020).
23. Wu, T. *et al.* clusterProfiler 4.0: A universal enrichment tool for interpreting omics data. *The Innovation* **2**, 100141 (2021).
24. Ye, Y. *et al.* Characterization of Hypoxia-associated Molecular Features to Aid Hypoxia-Targeted Therapy. *Nat Metab* **1**, 431–444 (2019).
25. Thienpont, B. *et al.* Tumour hypoxia causes DNA hypermethylation by reducing TET activity. *Nature* **537**, 63–68 (2016).
26. Liu, C. *et al.* Immunogenomic landscape analyses of immune molecule signature-based risk panel for patients with triple-negative breast cancer. *Mol Ther Nucleic Acids* **28**, 670–684 (2022).

27. Shen, S. *et al.* Development and validation of an immune gene-set based Prognostic signature in ovarian cancer. *EBioMedicine* **40**, 318–326 (2018).
28. Miao, Y.-R. *et al.* ImmuCellAI: A Unique Method for Comprehensive T-Cell Subsets Abundance Prediction and its Application in Cancer Immunotherapy. *Advanced Science* **7**, 1902880 (2020).
29. Zhang, Q., Luo, M., Liu, C.-J. & Guo, A.-Y. CCLA: an accurate method and web server for cancer cell line authentication using gene expression profiles. *Briefings in Bioinformatics* (2020) doi:10.1093/bib/bbaa093.
30. Liu, H. *et al.* ADORA1 Inhibition Promotes Tumor Immune Evasion by Regulating the ATF3-PD-L1 Axis. *Cancer Cell* **37**, 324-339.e8 (2020).
31. Li, H. *et al.* The Beneficial Role of Sunitinib in Tumor Immune Surveillance by Regulating Tumor PD-L1. *Advanced Science* **8**, 2001596 (2021).
32. Bundscherer, A. *et al.* Antiproliferative and proapoptotic effects of rapamycin and celecoxib in malignant melanoma cell lines. *Oncol Rep* **19**, 547–553 (2008).
33. Yao, Y. *et al.* Selenium–GPX4 axis protects follicular helper T cells from ferroptosis. *Nat Immunol* **22**, 1127–1139 (2021).
34. Iniguez, A. B. *et al.* EWS/FLI confers tumor cell synthetic lethality to CDK12 inhibition in Ewing sarcoma. *Cancer Cell* **33**, 202-216.e6 (2018).
35. Ebert, R. *et al.* Probenecid as a sensitizer of bisphosphonate-mediated effects in breast cancer cells. *Mol Cancer* **13**, 265 (2014).
36. Bruch, J. *et al.* PERK activation mitigates tau pathology in vitro and in vivo. *EMBO Mol Med* **9**, 371–384 (2017).
37. Yoshii, S. R. & Mizushima, N. Monitoring and Measuring Autophagy. *Int J Mol Sci* **18**, 1865 (2017).
38. Buffa, F. M., Harris, A. L., West, C. M. & Miller, C. J. Large meta-analysis of multiple cancers reveals a common, compact and highly prognostic hypoxia metagene. *Br J Cancer* **102**, 428–435 (2010).
39. Fox, N. S., Starmans, M. H., Haider, S., Lambin, P. & Boutros, P. C. Ensemble analyses improve signatures of tumour hypoxia and reveal inter-platform differences. *BMC Bioinformatics* **15**, 170 (2014).

40. Chen, H.-T. *et al.* Crosstalk between autophagy and epithelial-mesenchymal transition and its application in cancer therapy. *Molecular Cancer* **18**, 101 (2019).
41. Nguyen, T. N. *et al.* ATG4 family proteins drive phagophore growth independently of the LC3/GABARAP lipidation system. *Molecular Cell* **81**, 2013-2030.e9 (2021).
42. Lahiri, V. & Klionsky, D. J. ATG4-family proteins drive phagophore growth independently of the LC3/GABARAP lipidation system. *null* **17**, 1293–1295 (2021).
43. Sanchez-Vega, F. *et al.* Oncogenic Signaling Pathways in The Cancer Genome Atlas. *Cell* **173**, 321-337.e10 (2018).
44. Gillette, M. A. *et al.* Proteogenomic characterization reveals therapeutic vulnerabilities in lung adenocarcinoma. *Cell* **182**, 200-225.e35 (2020).
45. Thorsson, V. *et al.* The Immune Landscape of Cancer. *Immunity* **48**, 812-830.e14 (2018).

REVIEWER COMMENTS

Reviewer #1 (Remarks to the Author):

I would like to thank the authors for addressing the majority of my comments. The manuscript is now easier to read and understand, and the quality of the figures is improved. However, it seems that in some sentences, the authors are choosing the subjects incorrectly, for example:

page 11: "Among these, etoposide and BMS536924 became significantly sensitive under rapamycin-induced autophagy conditions in both A375 and SK-MEL-28 cell lines (Figure 7C)"

page 12: "To examine the functional role of DDIT4 in sensitizing etoposide by autophagy inducer, ..."

page 14: "Notably, we found a total of 32 CAGs were significantly altered across 20 cancer types, and drugs targeting these CAGs were resistant or sensitive to autophagy in multiple cancer types"

Cell lines and tumors can be sensitive or resistant to drugs, but the drugs themselves cannot be sensitive/resistant. Therefore, I would recommend proofreading the manuscript again before its publication.

Besides that, I have no further comments on this manuscript and would like to endorse it for publication.

Reviewer #2 (Remarks to the Author):

Although the reviewer recognizes the improvements of the MS, some parts are still missing or need better description, some experiments are still not convincing, and grammar mistakes are often found in the text. Here are my main concerns:

- Some panels in Fig. 7C,D and Fig. S5B do not show high differences. It is very difficult to understand how the authors performed and statistically analysed these data to have such a low p value. How many times has each experiment been repeated? No details are found in the text or figure legend. Also, one of the plots in Fig. 7C has wrong axis. As this experiment is crucial to validate their *in silico* findings, other assays should be performed to prove that melanoma cell lines are more sensitive to etoposide and BMS536924 upon autophagy induction. Besides cell viability, cell death could also be analysed.

- The choice of etoposide was not well explained in the text, and no reference has been cited about its use in the clinics.
- The autophagy marker LC3 should be better analysed. Very often, the lipidated/unlipidated bands are not visible in their WBs. The measurement of the LC3II/I ratio in each sample should be included and, in general, the densitometry of each WB is requested (both for LC3 and p62). The authors should refer to the “Guidelines for the use and interpretation of assays for monitoring autophagy” by Klionsky et al. (Autophagy 2021) to better understand how to interpret the autophagy markers and explain them in the text.
- The statistical significance is missed in the qPCR performed in the different cell lines (Fig. S6D). There is not information in the Figure legend either.
- In general, the figure legends are still very inaccurate (e.g., it is not clear how many samples were analysed). Each value or sample should be shown as a dot on each of the bar plots.
- In Fig. S7, DDT4 is clearly not efficiently knocked-down, regardless any type of siRNA used. Therefore, the experiments showed in Fig. 8C should be repeated with more efficient silencing of DDT4.
- The addition of public data analysis concerning DDIT4 expression upon rapamycin is appreciated but needs further explanation in the text. Where does this data come from? The reference must be included.
- The known role and function of DDIT4 should be mentioned and discussed in relation to its involvement in sensitizing melanoma cells to etoposide upon autophagy induction. Potential mechanism(s) underlying the drug sensitivity, dependent on DDIT4, should be speculated in the discussion.
- Densitometry of all WBs should be done.
- English has to be carefully revised to fix all the mistakes present in the text.

Reviewer #3 (Remarks to the Author):

The authors have corrected all my points (especially major overstatements, inconsistencies and English grammar), have discussed weaknesses or limitations of their approach (e.g., proper autophagy flux monitoring across samples) and have convincingly addressed all in vivo/in vitro experiments to fully support their claims. Therefore, I congratulate the authors for their work and recommend publication without reservations.

Reviewer #1 (Remarks to the Author):

I would like to thank the authors for addressing the majority of my comments. The manuscript is now easier to read and understand, and the quality of the figures is improved. However, it seems that in some sentences, the authors are choosing the subjects incorrectly, for example:

Response: We thank the reviewer's overall positive comments. We have carefully revised all sentences and proofread the revised manuscript.

page 11: "Among these, etoposide and BMS536924 became significantly sensitive under rapamycin-induced autophagy conditions in both A375 and SK-MEL-28 cell lines (Figure 7C)"

Response: We revised this sentence as "*Among these drugs, both A375 and SK-MEL-28 cell lines became significantly sensitive to etoposide and BMS536924 under the rapamycin-induced autophagy conditions (Figure 7B)*".

page 12: "To examine the functional role of DDIT4 in sensitizing etoposide by autophagy inducer, ..."

Response: We revised this sentence as "*To examine the functional role of DDIT4 in autophagy inducer sensitizing melanoma cells to etoposide, ...*".

page 14: "Notably, we found a total of 32 CAGs were significantly altered across 20 cancer types, and drugs targeting these CAGs were resistant or sensitive to autophagy in multiple cancer types"

Response: We revised this sentence as "*Notably, we found a total of 32 CAGs were significantly altered across 20 cancer types, and multiple cancer types can be resistant or sensitive to the drugs targeting these CAGs upon autophagy induction*".

Cell lines and tumors can be sensitive or resistant to drugs, but the drugs themselves cannot be sensitive/resistant. Therefore, I would recommend

proofreading the manuscript again before its publication.

Response: We have checked relevant sentences and proofread the manuscript. For example, we have revised the sentence “*Among these drugs, both A375 and SK-MEL-28 cell lines became significantly sensitive to etoposide and BMS536924 under the rapamycin-induced autophagy conditions (Figure 7B)*”.

Besides that, I have no further comments on this manuscript and would like to endorse it for publication.

Response: We appreciate the reviewer’s endorsement.

Reviewer #2 (Remarks to the Author):

Although the reviewer recognizes the improvements of the MS, some parts are still missing or need better description, some experiments are still not convincing, and grammar mistakes are often found in the text. Here are my main concerns:

Response: We thank the reviewer's overall positive comments. We further performed additional experiments to make our conclusion more convincing. We also carefully proofread the revised manuscript.

- Some panels in Fig. 7C,D and Fig. S5B do not show high differences. It is very difficult to understand how the authors performed and statistically analysed these data to have such a low p value. How many times has each experiment been repeated? No details are found in the text or figure legend. Also, one of the plots in Fig. 7C has wrong axis. As this experiment is crucial to validate their in silico findings, other assays should be performed to prove that melanoma cell lines are more sensitive to etoposide and BMS536924 upon autophagy induction. Besides cell viability, cell death could also be analysed.

Response: In Fig 7C,D (revised Fig 7B,D) and Fig. S5B, the drug response data of different groups were fitted and compared by sigmoidal dose-response curves. Each group of experiments have 4 replicates. We added the statistical method and sample number in figure7 legends, "*The drug screen data of different groups (n = 4) were fitted and compared by sigmoidal dose-response curves.*" We also unified the y axis of Fig. 7B, ranging from 0 to 150. Please note, the drug response experiment is a well-established assay, which has been applied in our previous studies (Han et al., 2015; Ye et al., 2019), as well as many other studies (Cordo' et al., 2022; Wang et al., 2017; Chevereau et al., 2015; Shoichet, 2006). The difference presented in our drug screen assay is already very large and significant.

Rapamycin - induced autophagy

Figure 7: Characterization of drug response associated with autophagy in vitro and in vivo. (B) Dose-response curves for the mean value of cell viability of etoposide and BMS536924 in rapamycin-induced and non-induced conditions in the melanoma cell line A375 and SK-MEL-28. Cell viability was normalized to the level of cells treated with DMSO. Error bars indicate the mean \pm SD. The drug screen data of different groups ($n = 4$) were fitted and compared by sigmoidal dose-response curves.

Nevertheless, we further performed cell death assays per the reviewer's request. Our data showed that melanoma cell lines (A375, SK-MEL-28, SK-MEL-5) are more sensitive to etoposide and BMS536924 upon autophagy induction (rapamycin-induced or starvation-induced). We added these new experimental data in Figure S5. We also added relevant description in the method "*Cell death was quantified by the percentage of propidium iodide (PI; P4170, Sigma-Aldrich) positive staining cells detected by flow cytometric analysis. 2×10^5 cells per well were seeded in 6-well plates and treated with indicated drugs. After harvesting the cells, they were washed with PBS twice and then incubated with antibodies on ice for 30 minutes in the dark. Flow cytometric analysis was conducted on a FACS LSR II Fortessa (BD Biosciences) and the FACS data were analyzed with FlowJo software (Tree Star).*" and the result "*Moreover, we performed the cell death assays and observed that etoposide and BMS536924 killed more melanoma cells in rapamycin-induced/starvation-induced autophagy conditions (Figure S5C-S5G). These results suggest that autophagy induction is likely to cause the drug sensitivity.*"

Figure S5

Figure S5: Characterization of autophagy sensitizing drug response in vitro and in vivo. (D & F) Cell death was quantified by the percentage of propidium iodide (PI) positive staining cells detected by flow cytometric analysis. Representative cell death plots of melanoma cell lines (A375, SK-MEL-28, SK-MEL-5) treated with etoposide and BMS536924 in rapamycin-induced (D) and starvation-induced (F), and non-induced conditions. (E & G) Statistical data of the cell death in the (D, F) groups (n = 3) were analyzed by Student's t test. Data were presented as mean \pm SD. *P < 0.05; **P < 0.01; ***P < 0.001; ****P < 0.0001; ns is not significant.

- The choice of etoposide was not well explained in the text, and no reference has been cited about its use in the clinics.

Response: We added the reference (PMID: 1984835; The clinical pharmacology of etoposide) in the revised manuscript, "*To further confirm the effects of autophagy on the drug response in vivo, we chose etoposide, an anti-tumor drug used in the clinic⁶⁹, to create a xenograft model with the A375 cell line, which has a highly similar pattern to SK-MEL-28 in drug sensitivity (Figure 7B, 7D).*".

- The autophagy marker LC3 should be better analysed. Very often, the lipidated/unlipidated bands are not visible in their WBs. The measurement of the LC3II/I ratio in each sample should be included and, in general, the densitometry of each WB is requested (both for LC3 and p62). The authors should refer to the "Guidelines for the use and interpretation of assays for monitoring autophagy" by Klionsky et al. (Autophagy 2021) to better understand how to interpret the autophagy markers and explain them in the text.

Response: We referred to "Guidelines for the use and interpretation of assays for monitoring autophagy" by Klionsky et al. (Autophagy 2021) in the revised manuscript, and added the reference in the revised manuscript "*We examined the autophagy status of melanoma cells on different conditions by western blot of LC3A/B and p62⁶⁸, confirming the activation of autophagy by rapamycin*

treatment (Figure 7C).”. Per the guidelines, we measured the densitometry of all WBs (Figure 7C, 7G, 8C, 8E, 8H, S7B-S7F), including LC3, p62, DDIT4, and ATG5. For LC3II/I, We calculated the LC3II/I ratio of densitometry in each sample (Figure 7C, 7G, S7C, S7E). We then normalized the value for each sample to the first lanes, e.g., DMSO or Vehicles.

Figure 7

Figure 7: Characterization of drug response associated with autophagy in vitro and in vivo. (C) Western blot of autophagy markers in A375 and SK-MEL-28 cell lines treated with DMSO, rapamycin, etoposide, and rapamycin + etoposide. The ratios of the LC3-II/LC3-I and p62 band intensities normalized to DMSO are displayed below the blots. **(G)** Western blot of autophagy markers in mouse tissues with treatment of vehicle, rapamycin, etoposide, and rapamycin + etoposide. The ratios of the LC3-II/LC3-I and p62 band intensities normalized to vehicle are displayed below the blots (n = 3).

Figure 8

Figure 8: Potential mechanism through the functional characterization of DDIT4 in vitro. (C) Western blot of DDIT4 in A375 or SK-MEL-28 cells treated with si-NC or DDIT4 siRNAs in combination with etoposide. DDIT4 band intensities normalized to si-NC+etoposide are displayed below the blots. (E) Western blot of DDIT4 in A375 or SK-MEL-28 cells transfected with DDIT4-OE lentivirus or vector in combination with etoposide and rapamycin. DDIT4 band intensities normalized to Vector+Etoposide are displayed below the blots. (H) Western blot of DDIT4 in A375 or SK-MEL-28 cells treated with si-NC or ATG5 siRNAs in combination with etoposide. DDIT4 band intensities normalized to si-NC+DMSO are displayed below the blots.

Figure S7

Figure S7: Expression of DDIT4 and the efficiency and specificity of DDIT4 siRNAs or DDIT4-OE lentivirus and ATG5 siRNAs in vitro. (B) The western blot of DDIT4 in A375 and SK-MEL-28 cells transfected with si-NC or DDIT4 siRNAs. DDIT4 band intensities normalized to si-NC are displayed below the blots. (C) The autophagy status of A375 and SK-MEL-28 cells transfected with si-NC or DDIT4 siRNAs. The ratios of band intensities for the LC3-II/LC3-I normalized to si-NC are displayed below the blots. (D) The western blot of DDIT4 in A375 and SK-MEL-28 cells transfected with vector or DDIT4-OE lentivirus. DDIT4 band intensities normalized to Vector are displayed below the blots. (E) Western blot of autophagy markers in A375 and SK-MEL-28 cells transfected with vector

or DDIT4-OE lentivirus. The ratios of band intensities for the LC3-II/LC3-I normalized to Vector are displayed below the blots. **(F)** The western blot of ATG5 in A375 and SK-MEL-28 cells transfected with si-NC or ATG5 siRNAs. ATG5 band intensities normalized to si-NC are displayed below the blots.

- The statistical significance is missed in the qPCR performed in the different cell lines (Fig. S6D). There is not information in the Figure legend either.

Response: We added the statistical significance in the Figure S6D, and illustrated the statistical information in the figure legend, “*Figure S6: Analysis of autophagy sensitizing drug response. (D) Relative expression of EGF, FGF1, BINP3, PPP2R2B, SFN and DDIT4 in four groups (n = 3) of A375 and -SK-MEL-28 by RT-PCR. Data was presented as means ± SD. The difference in multiple groups was estimated by one-way ANOVA analysis. *P < 0.05; **P < 0.01; ***P < 0.001; ****P < 0.0001.*”.

Figure S6

Figure S6: Analysis of autophagy sensitizing drug response. (D) Relative expression of EGF, FGF1, BINP3, PPP2R2B, SFN and DDIT4 in four groups (n = 3) of A375 and -SK-

MEL-28 by RT-PCR. Data was presented as means \pm SD. The difference in multiple groups was estimated by one-way ANOVA analysis. *P < 0.05; **P < 0.01; ***P < 0.001; ****P < 0.0001; ns is not significant.

- In general, the figure legends are still very inaccurate (e.g., it is not clear how many samples were analysed). Each value or sample should be shown as a dot on each of the bar plots.

Response: We added dots into all bar plots (Figure 8B & 8D & 8F & 8G & S6D), and also revised the figure legends.

Figure 8

Figure 8: Potential mechanism through the functional characterization of DDIT4 in vitro. (B) Relative mRNA expression of DDIT4 in A375 and SK-MEL-28 by RT-PCR (n = 3). **(D)** The cell viability of A375 or SK-MEL-28 cells treated with si-NC or DDIT4 siRNAs in combination with etoposide (n = 4). Pink bars denote successful knockdown of si-DDIT4 (si-DDIT4-2 and siDDIT4-3), while grey bar denotes unsuccessful knockdown of si-DDIT4 (si-DDIT4-1). **(F)** The cell viability of A375 or SK-MEL-28 cells transfected with DDIT4-OE lentivirus or vector in combination with etoposide and rapamycin (n = 4). **(G)** The cell

viability of A375 or SK-MEL-28 cells treated with si-NC or ATG5 siRNA in combination with etoposide (n = 4). (B, D, F and G) Data was presented as means \pm SD. (B, D, and F) The difference in multiple groups was estimated by one-way ANOVA analysis. (G) Student's t test was used for estimation of difference in two groups *P < 0.05; **P < 0.01; ***P < 0.001; ****P < 0.0001; ns is not significant.

Figure S6

Figure S6: Analysis of autophagy sensitizing drug response. (D) Relative expression of EGF, FGF1, BINP3, PPP2R2B, SFN and DDIT4 in four groups (n = 3) of A375 and -SK-MEL-28 by RT-PCR. Data was presented as means \pm SD. The difference in multiple groups was estimated by one-way ANOVA analysis. *P < 0.05; **P < 0.01; ***P < 0.001; ****P < 0.0001; ns is not significant.

- In Fig. S7, DDT4 is clearly not efficiently knocked-down, regardless any type of siRNA used. Therefore, the experiments showed in Fig. 8C should be repeated with more efficient silencing of DDT4.

Response: We apologized for providing wrong Figure S7 in the previous

revision. We have corrected the Figure S7, which showed that si-DDIT4-2 and si-DDIT4-3 were efficiently knocked-down.

Figure S7

Figure S7: Expression of DDIT4 and the efficiency and specificity of DDIT4 siRNAs or DDIT4-OE lentivirus and ATG5 siRNAs in vitro. (B) Western blot of DDIT4 in A375 and SK-MEL-28 cells transfected with si-NC or DDIT4 siRNAs. DDIT4 band intensities normalized to si-NC are displayed below the blots.

- The addition of public data analysis concerning DDIT4 expression upon rapamycin is appreciated but needs further explanation in the text. Where does this data come from? The reference must be included.

Response: The data comes from GSE27784 dataset (PMID: 21876130), and we have included the reference in the revised manuscript, “*Moreover, we also observed that the expression of DDIT4 could be decreased by rapamycin in a public data from GSE27784⁷⁰ (Figure S7A)*”.

- The known role and function of DDIT4 should be mentioned and discussed in relation to its involvement in sensitizing melanoma cells to etoposide upon autophagy induction. Potential mechanism(s) underlying the drug sensitivity, dependent on DDIT4, should be speculated in the discussion.

Response: We thank the reviewer’s comments. Through the experiments, we validated the downregulation of DDIT4 and its effects on melanoma cells, which indicated that the DDIT4 gene indeed played a critical role in the mechanism of autophagy-induction sensitizing melanoma cells to etoposide. We speculated that the potential mechanism may through DNA damage

response. However, we do not have strong evidence that this is related to DNA damage response, which is far beyond the scope of our current manuscript. We added discussion about DDIT4 “*Our analysis further suggested that DDIT4 is the potential target to mediate the autophagy induction to sensitize the cancer cells to etoposide.*”.

- Densitometry of all WBs should be done.

Response: We added densitometry for all WBs in the revised manuscript. Please refer to our response for above comment.

- English has to be carefully revised to fix all the mistakes present in the text.

Response: We carefully revised all sentences and proofread the revised manuscript.

Reviewer #3 (Remarks to the Author):

The authors have corrected all my points (especially major overstatements, inconsistencies and English grammar), have discussed weaknesses or limitations of their approach (e.g., proper autophagy flux monitoring across samples) and have convincingly addressed all in vivo/in vitro experiments to fully support their claims. Therefore, I congratulate the authors for their work and recommend publication without reservations.

Response: We thank the reviewer's entirely positive comments and appreciate for all the great suggestions.

Reference

Chevereau, G., Dravecká, M., Batur, T., Guvenek, A., Ayhan, D.H., Toprak, E., and Bollenbach, T. (2015). Quantifying the Determinants of Evolutionary Dynamics Leading to Drug Resistance. *PLOS Biology* 13, e1002299. <https://doi.org/10.1371/journal.pbio.1002299>.

Cordo', V., Meijer, M.T., Hagelaar, R., de Goeij-de Haas, R.R., Poort, V.M., Henneman, A.A., Piersma, S.R., Pham, T.V., Oshima, K., Ferrando, A.A., et al. (2022). Phosphoproteomic profiling of T cell acute lymphoblastic leukemia reveals targetable kinases and combination treatment strategies. *Nat Commun* 13, 1048. <https://doi.org/10.1038/s41467-022-28682-1>.

Han, L., Diao, L., Yu, S., Xu, X., Li, J., Zhang, R., Yang, Y., Werner, H.M.J., Eterovic, A.K., Yuan, Y., et al. (2015). The Genomic Landscape and Clinical Relevance of A-to-I RNA Editing in Human Cancers. *Cancer Cell* 28, 515–528. <https://doi.org/10.1016/j.ccell.2015.08.013>.

Shoichet, B.K. (2006). Interpreting Steep Dose-Response Curves in Early Inhibitor Discovery. *J. Med. Chem.* 49, 7274–7277. <https://doi.org/10.1021/jm061103g>.

Wang, D., Lin, Z., Wang, T., Ding, X., and Liu, Y. (2017). An analogous wood barrel theory to explain the occurrence of hormesis: A case study of sulfonamides and erythromycin on *Escherichia coli* growth. *PLOS ONE* 12, e0181321. <https://doi.org/10.1371/journal.pone.0181321>.

Ye, Y., Hu, Q., Chen, H., Liang, K., Yuan, Y., Xiang, Y., Ruan, H., Zhang, Z., Song, A., Zhang, H., et al. (2019). Characterization of Hypoxia-associated Molecular Features to Aid Hypoxia-Targeted Therapy. *Nat Metab* 1, 431–444. <https://doi.org/10.1038/s42255-019-0045-8>.

REVIEWERS' COMMENTS

Reviewer #2 (Remarks to the Author):

I would like to congratulate the authors who have addressed most of the points raised by the reviewers. The revised manuscript is notably improved. My only concern is about some WB of LC3, where they calculated the LC3II/I ratio (Fig. 7C right panel, G, S7C, E) and the two bands are not clearly distinguishable. So, I'm wondering if they managed to correctly calculate the intensity of these bands, and it would be highly recommended to replace these with better images.

Reviewer #2 (Remarks to the Author):

I would like to congratulate the authors who have addressed most of the points raised by the reviewers. The revised manuscript is notably improved. My only concern is about some WB of LC3, where they calculated the LC3II/I ratio (Fig. 7C right panel,G, S7C,E) and the two bands are not clearly distinguishable. So, I'm wondering if they managed to correctly calculate the intensity of these bands, and it would be highly recommended to replace these with better images.

Response: We thank the reviewer's positive comments. We read some other literatures that measured the LC3II/I ratio, for example, HeLa cells (Fig. 1a; PMID: 27929117; please see below example), Vero-E6 cells (Fig. 1b; PMID: 33966045), ovarian cancer stem cells (Fig. 1a; PMID: 28726781). We noticed that this is quite usual for the indistinguishable bands. However, these bands can still be measured correctly by Image J and/or other software. We have repeated the experiments multiple times during the review process, and unfortunately, the images we provided in the manuscript are the best we can get.

Fig.1 CCT knockdown reduces LC3 turnover and autophagosome formation (figure from PMID: 27929117). (a) LC3-II levels in HeLa cells transfected with siRNA against scrambled control or CCT subunits treated with vehicle (DMSO) or Bafilomycin A1 (400 nM) for 5 h. GAPDH is loading control.